# An atypical atherogenic chemokine that promotes advanced atherosclerosis and hepatic lipogenesis

Atherosclerosis is the underlying cause of myocardial infarction and ischemic stroke. It is a lipid-triggered and cytokine/chemokine-driven arterial inflammatory condition. We identify D-dopachrome tautomerase/macrophage migration-inhibitory factor-2 (MIF-2), a paralog of the cytokine MIF, as an atypical chemokine promoting both atherosclerosis and hepatic lipid accumulation. In hyperlipidemic $Apoe^{-/-}$ mice, $Mif$-$2$-deficiency and pharmacological MIF-2-blockade protect against lesion formation and vascular inflammation in early and advanced atherogenesis. MIF-2 promotes leukocyte migration, endothelial arrest, and foam-cell formation, and we identify CXCR4 as a receptor for MIF-2. $Mif$-$2$-deficiency in $Apoe^{-/-}$ mice leads to decreased plasma lipid levels and suppressed hepatic lipid accumulation, characterized by reductions in lipogenesis-related pathways, tri-/diacylglycerides, and cholesterol-esters, as revealed by hepatic transcriptomics/lipidomics. Hepatocyte cultures and FLIM-FRET-microscopy suggest that MIF-2 activates SREBP-driven lipogenic genes, mechanistically involving MIF-2-inducible CD74/CXCR4 complexes and PI3K/AKT but not AMPK signaling. MIF-2 is upregulated in unstable carotid plaques from atherosclerotic patients and its plasma concentration correlates with disease severity in patients with coronary artery disease. These findings establish MIF-2 as an atypical chemokine linking vascular inflammation to metabolic dysfunction in atherosclerosis.

Atherosclerotic cardiovascular diseases (ASCVDs) such as myocardial infarction and ischemic stroke remain the leading cause of mortality globally. ASCVDs are promoted by risk and lifestyle factors such as hypertension, high-fat Western-type diets, and lack of physical activity; disease pathology is associated with co-morbidities such as diabetes or metabolic dysfunction-associated steatotic liver disease (MASLD)[1]. Atherosclerosis is a lipid-triggered chronic inflammatory condition of the arterial vascular wall driven by immune and inflammatory pathways[2–5]. The atherogenic process involves leukocyte infiltration and lesional inflammation, processes orchestrated by dysregulated activities of cytokines and chemokines[5,6].

Macrophage migration-inhibitory factor (MIF) is a multifunctional inflammatory mediator with pathogenic roles in inflammatory diseases[7–10]. Plasma MIF levels are associated with coronary artery disease (CAD), and we and others have previously detected enhanced MIF expression in atherosclerotic plaques[11–14] and demonstrated that MIF promotes atherogenic leukocyte recruitment through non-cognate interaction with the chemokine receptors CXCR2 and CXCR4[15]. Accordingly, MIF is an atypical chemokine (ACK), an emerging family of proteins that bind to classical chemokine receptors, while lacking classifying chemokine features such as N-terminal cysteines and a chemokine-fold[9,16]. MIF also binds to CD74, the cognate receptor of MIF, also known as the plasma-membrane form of class II invariant-chain Ii[17]. MIF engages CD74 or CXCR4 by direct high-affinity binding[15,17], but depending on context, MIF signaling through CD74 requires co-receptors such as CD44[18,19] or chemokine

✉ e-mail: Omar.El_Bounkari@med.uni-muenchen.de; juergen.bernhagen@med.uni-muenchen.de

receptors[15,20–22]. MIF's pro-atherogenic activities are mainly mediated through its chemokine receptor pathways, with contributions from CD74[23]. The MIF/CD74 axis also leads to activation of adenosine-monophosphate kinase (AMPK), a pathway associated with MIF-mediated tissue protection in cardiac ischemia and hepatosteatosis/metabolic dysfunction-associated steatohepatitis (MASH)[24,25].

D-dopachrome tautomerase (D-DT) is a member of the MIF family that was originally identified as an enzyme converting the non-natural substrate D-dopachrome to 5,6-dihydroxyindole[26,27]. It shares remarkable three-dimensional structural similarity, but only 34% sequence homology, with MIF[28]. A role for D-DT in human immunity has only emerged recently, when it was characterized as a functional cytokine homolog of MIF[29] and termed MIF-2[29–31]. MIF-2 interacts with CD74 to promote MAPK activation[29] and MIF-2/ACKR3 interactions have been implicated in epithelial repair in chronic obstructive pulmonary disease (COPD)[32,33]. MIF-2 lacks the pseudo(E)LR motif required for interaction with CXCR2[15,34] and does not bind CXCR2[35]. Functional studies in carcinoma and endotoxemia models suggest that, depending on tissue or disease context, MIF and MIF-2 exhibit overlapping or different effects[29,35–37]. In adipose tissue inflammation, MIF and MIF-2 display opposite activities[38–40]. In the ischemic heart, MIF and MIF-2 activate protective AMPK signaling in cardiomyocytes[25,41–45], but the role of MIF-2 in atherosclerosis has not been studied.

Applying a model of Apolipoprotein e-deficient (*Apoe*[-/-]) mice combined with either *Mif-2* deficiency or pharmacological MIF-2 blockade, we investigated the role of MIF-2 in atherosclerosis. *Mif-2*[-/-]*Apoe*[-/-] mice on high-fat diet (HFD) exhibited reduced atherosclerotic lesions in both early and advanced stages of atherogenesis compared to *Apoe*[-/-] mice. Biochemical binding, flow arrest, chemotaxis, and in vivo recruitment experiments indicated that MIF-2 is a CXCR4-engaging chemokine for monocytes and B cells. *Mif-2*[-/-]*Apoe*[-/-] mice showed reduced plasma lipid levels and were protected from hepatosteatosis. Sterol-response element binding proteins (SREBPs) are transcription factors that control the expression of genes involved in hepatic lipogenesis and lipid homeostasis[46,47]. Mechanistic experiments employing transcriptomics, lipidomics, hepatocyte cultures, SREBP and target gene analysis, receptor blocking, proximity ligation assay (PLA), and FLIM/FRET analysis suggested that MIF-2 promotes hepatic lipogenesis through activation of the CXCR4/CD74-SREBP axis and numerous lipogenic enzymes. Together with an observed upregulation of MIF-2 in unstable atherosclerotic plaques from CEA patients and a correlation of MIF-2 levels with the severity of CAD, our data suggest a role for MIF-2 in advanced atherosclerosis involving a dual lipid/hepatic and vascular phenotype.

## Results
### Genetic deletion and pharmacological blockade of MIF-2 mitigate atherosclerotic lesion formation in early atherogenesis
We first assessed the impact of genetic deletion of *Mif-2* in a model of early atherogenesis. *Mif-2*[-/-]*Apoe*[-/-] mice were generated by cross-breeding *Mif-2*[+/-] and *Apoe*[-/-] mice (Supplementary Fig. 1a–d; Supplementary Table 1). Groups of female *Mif-2*[-/-]*Apoe*[-/-] versus *Apoe*[-/-] littermates were fed a Western-type cholesterol-rich HFD for 4.5 weeks to initiate plaque formation (Fig. 1a). *Mif-2*-deficient *Apoe*[-/-] mice displayed no gross phenotype differences compared to *Apoe*[-/-] littermates, but exhibited a decreased blood monocyte and slightly increased blood T-cell count (Supplementary Table 2).

Quantification of atherosclerotic lesions following oil-red O (ORO) and hematoxylin-eosin (HE) staining revealed reduced lesion areas in aortic root and arch in *Mif-2*[-/-]*Apoe*[-/-] compared to *Apoe*[-/-] mice (Fig. 1b–d). Reduced plaque formation was accompanied by a decreased number of lesional macrophages as determined by CD68 staining (Fig. 1e). As vascular inflammation typically correlates with circulating markers, we evaluated the plasma levels of inflammatory cytokines and chemokines by protein array (Fig. 1f–h, Supplementary

Fig. 2). We determined a significant reduction of IFN-γ, IL-2, IL-16, IL-17, CXCL13, and IL-1α concentrations in *Mif-2*[-/-]*Apoe*[-/-] mice compared with controls (Fig. 1f–h, Supplementary Fig. 2). In addition, heatmap analysis shows a decrease for cytokines such as IL-23, IL-1ra, and IL-27, or chemokines such as CXCL9, CXCL10, and CXCL11 (Fig. 1f). A similar phenotype was observed in male *Mif-2*[-/-]*Apoe*[-/-] mice, with smaller plaques and decreased inflammation compared to *Apoe*[-/-] controls (Supplementary Fig. 3a–d). Collectively, these results indicated that global *Mif-2* deficiency attenuates early atherogenesis and vascular inflammation in hyperlipidemic mice.

To confirm the role of MIF-2 in atherogenesis, we employed a pharmacological approach, using 4-CPPC, a small molecule inhibitor of MIF-2 that exhibits selectivity for MIF-2 over MIF[31,48]. Male *Apoe*[-/-] mice were administered 4-CPPC (i.p.; 5 mg kg⁻¹) versus vehicle control three times per week parallel to the HFD (Fig. 1I). Atherosclerotic lesion size in aortic root was markedly decreased in 4-CPPC-treated mice compared with controls (Fig. 1j, k). Similarly, there was a reduction in lesional macrophages (Fig. 1l) and circulating cytokines and chemokines as determined by Luminex multiplex analysis, with the heatmap indicating reductions in IL-1α, IL-6, IL-15, or IFN-γ (Fig. 1m). The relative reductions observed in the cytokine/chemokine heatmaps of male mice on HFD were similar, although not identical, when comparing genetic *Mif-2* deficiency (*Mif-2*[-/-]*Apoe*[-/-] versus *Apoe*[-/-]) with that of pharmacological blockade of Mif-2 (*Apoe*[-/-] + 4-CPPC versus vehicle). Overlapping reduced cytokines/chemokines included IL-1α, IFN-γ, IL-5, CXCL5, IL-15, TNF-α, IL-4, IL-23, IL-1β, CCL4, and IL-27. Taken together, genetic deletion and pharmacological blockade suggested that MIF-2 is a pro-atherogenic player in early stages of atherosclerosis.

### MIF-2 promotes leukocyte adhesion and chemotactic migration
The observed phenotype suggested that, similar to MIF, MIF-2 may enhance atherogenicity of leukocytes by influencing their adhesion and migration. To address this possibility, we assessed the adhesion of MonoMac6 monocytes to confluent monolayers of human aortic endothelial cells (HAoECs) under flow stress. MIF-2 dose-dependently promoted MonoMac6 arrest. The peak effect of 1.6 nM MIF-2 was higher than that of MIF (16 nM) (Fig. 2a). Next, we interrogated the migratory capacity of primary human monocytes in response to MIF-2 employing a Transwell chemotaxis setup. MIF-2 elicited the chemotactic migration of monocytes in a dose-dependent manner (Fig. 2b), exhibiting a typical bell-shaped dose curve, well in line with the concentration optimum of MIF-2 in the arrest assay. To verify this effect, we examined the impact of 4-CPPC on monocyte migration triggered by the optimal MIF-2 dose. 4-CPPC dose-dependently inhibited monocyte migration in response to MIF-2, with complete ablation seen at a 5-fold molar excess (Fig. 2c). Additionally, we investigated MIF-2-mediated monocyte migration in a 3D-collagen matrix-based model applying time-lapse microscopy and single-cell tracking. MIF-2 concentration-dependently enhanced the directional migration of monocytes as illustrated by the single-cell migration tracks (Fig. 2d), and quantitation of the forward migration index (Fig. 2e). Together, these data provided evidence that MIF-2 is a chemokine-like mediator that promotes monocyte adhesion and chemotaxis.

We next asked whether the chemotactic capacity of MIF-2 would extend to other leukocyte populations that are implicated in atherogenesis[20,49]. MIF-2 elicited the migration of primary murine splenic B lymphocytes in a concentration-dependent manner, with a maximal chemotactic index again obtained at 4 nM (Fig. 2f). This effect was confirmed by applying the small molecule compound 4-IPP, an inhibitor of both MIF and MIF-2[50,51] (Fig. 2g). To evaluate the in vivo relevance of MIF-2-triggered B-lymphocyte trafficking, we studied homing of splenic B lymphocytes in C57BL/6 mice. To account for effects of MIF, wild-type (WT) versus *Mif* gene-deficient (*Mif*[-/-]) mice were examined (Fig. 2h). 4-IPP was administered to both groups to block MIF and MIF-2. Splenic B cells from WT mice stained with the

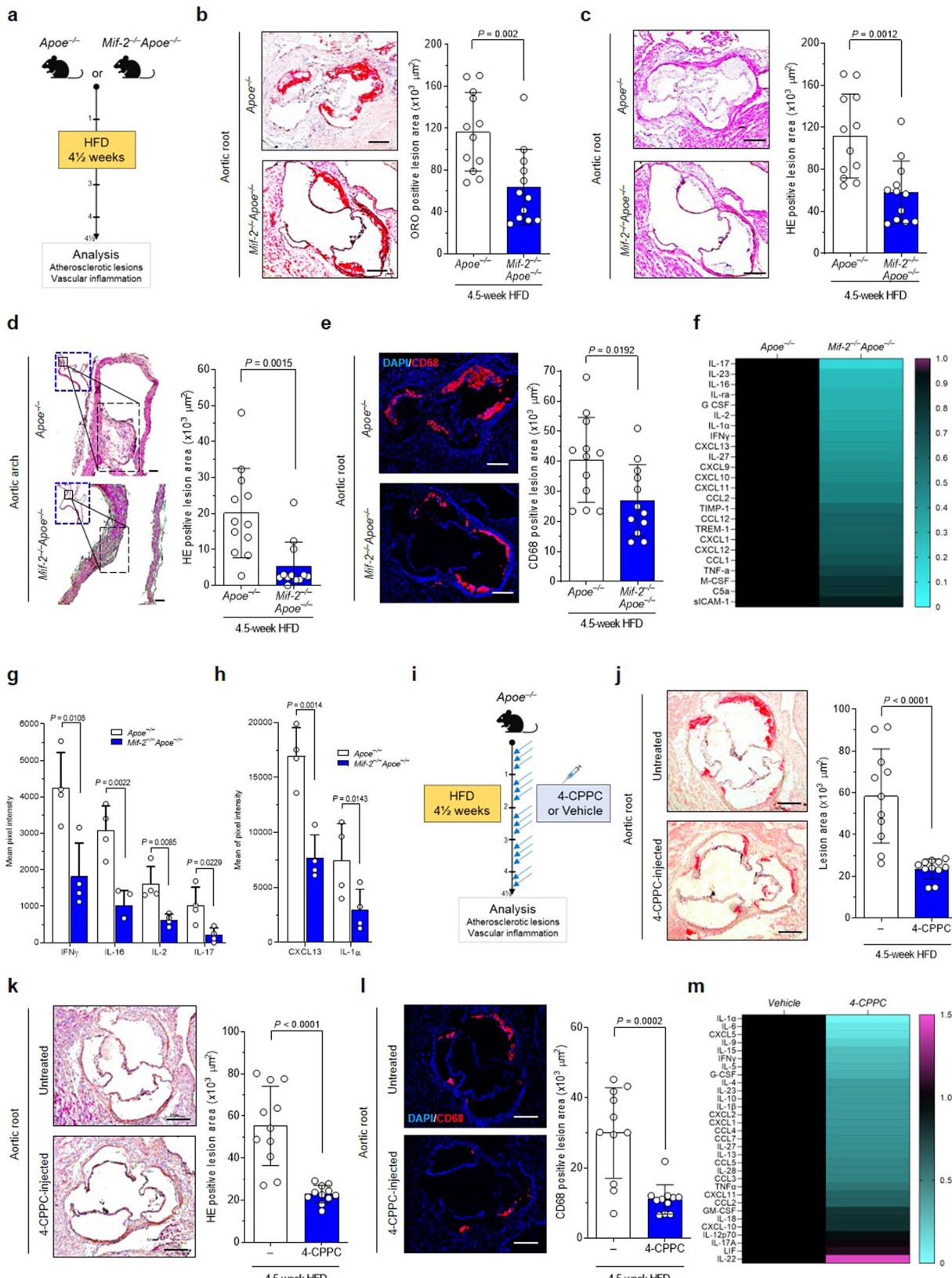

Cell-Tracker dye CMFDA were adoptively transferred into WT or *Mif*[−/−] recipient mice injected with 4-IPP (2.5 mg/kg) or vehicle control. Two-hours post-injection, CMFDA[+] cells were quantified by flow cytometry from spleen, bone marrow (BM), and lymph node (LN) (Fig. 2h). Untreated WT and *Mif*[−/−] recipient mice showed a comparable number of CMFDA[+] B cells homed into spleen (Fig. 2i). This suggested that MIF-2 or other B-cell chemokines mediated the homing effect. Treatment of mice with 4-IPP led to a marked reduction of B-cell homing into the spleen in *Mif*-deleted mice, arguing for a role of MIF-2 in the trafficking response (Fig. 2i). A similar trend was seen in BM, while no changes were observed for circulating B-cell counts or LN (Supplementary Fig. 4a–c). Collectively, these results suggested that MIF-2 promotes

**Fig. 1 | Genetic deletion or pharmacological inhibition of MIF−2 attenuates early atherosclerotic lesions and vascular inflammation in vivo. a** Experimental outline: Female *Apoe*[−/−] and *Mif-2*[−/−]*Apoe*[−/−] mice fed a cholesterol-rich HFD for 4.5 weeks. **b** ORO staining of aortic root sections and corresponding quantification (12 serial sections per mouse; *n* = 12 mice; each data-point represents one mouse; scale bar: 250 μm). **c** Same as (**b**) but HE staining. **d** HE staining and quantification of aortic arch sections (8 sections per mouse; *n* = 12 mice; scale bar: 200 μm). **e** Same as (**b**, **c**), except that sections were stained for CD68[+] macrophages (red) and DAPI (blue); scale bar: 200 μm. **f–h** Quantification of 40 inflammatory/atherogenic cytokines/chemokines in plasma samples by mouse cytokine array (*n* = 4 mice per group, analyzed in duplicate each). **f** Heatmap illustrating altered levels of cytokines/chemokines. Signals from *Apoe*[−/−] were normalized to 1 and compared with those from *Mif-2*[−/−]*Apoe*[−/−] mice (upregulated cytokines in magenta, downregulated in cyan). **g**, **h** Bar graphs of significantly altered cytokines/chemokines with

relatively lower (**h**) or higher (**g**) plasma abundance. **i–m** Inhibition of MIF-2 by 4-CPPC in a 4.5-week HFD male *Apoe*[−/−] model. **i** Scheme of the experimental outline. Mice were administered 4-CPPC (5 mg/kg) or vehicle 3× per week (Syringe icon was created with BioRender.com). **j** ORO staining of aortic root sections and quantification (12 sections per mouse; *n* = 11 mice per group; scale bar: 200 μm). **k** Same as (**j**) but HE staining. **l** Same as (**j**), except that sections were stained for CD68[+] macrophages (red) and DAPI (blue). **m** Quantitation of plasma cytokines/chemokines. Heatmap illustration of the 36 ProcartaPlex mouse cytokine/chemokine array (*n* = 4 mice per group analyzed in duplicate). Signals from vehicle-treated mice normalized to 1 and compared with those from 4-CPPC-treated mice (upregulated cytokines in magenta, downregulated in cyan). **a** and **i** were created in BioRender. Bernhagen, L. 2025 https://BioRender.com/c59v181. Values are expressed as means ± SD and statistically analyzed using an unpaired two-tailed *t*-test (**b**–**e** and **j**–**l**) or multiple unpaired *t*-tests (**g**, **h**).

## MIF-2 interacts with CXCR4 and the MIF-2/CXCR4 axis controls atherogenic activities

Given that MIF-2 promoted the migration of leukocytes, we wished to directly compare the migratory capacity of MIF and MIF-2. Primary B lymphocytes in a 3D-chemotaxis assay setup were exposed to optimal concentrations of MIF-2 (4 nM) and MIF (8 nM), so that cells were exposed to competing gradients of MIF-2 and MIF. B cells migrated toward MIF-2, irrespective of whether MIF or buffer was placed in the opposite chamber (Fig. 3a), suggesting that MIF-2 is the more potent chemokine.

Sequence alignment of MIF and MIF-2 (Fig. 3b) illustrates the homology between both MIF proteins, including the conserved Pro-2 residue, but also indicates differences in putative receptor binding motifs. Most notably, the CXCR2 binding signature motif of MIF is not present in MIF-2, in line with recent findings that MIF, but not MIF-2, recruit macrophages via CXCR2 in a polymicrobial sepsis model[35]. In contrast, residues required for binding to CXCR4 are conserved (Fig. 3b), letting us hypothesize that MIF-2 is a ligand of CXCR4.

In addition to CD74, mouse B lymphocytes express high surface levels of CXCR4[20,22]. To study the role of these receptors, we tested the effect of receptor-specific pharmacological blockade on MIF-2-mediated B-cell chemotaxis. A neutralizing antibody against CD74, but not an isotype control immunoglobulin, blocked MIF-2-elicited B-cell chemotaxis (Fig. 3c). Furthermore, co-incubation of B cells with the CXCR4-specific inhibitor AMD3100/plerixafor and the Giα-inhibitor pertussis toxin (PTX) abrogated MIF-2-mediated B-cell migration, indicating the involvement of CXCR4 as well (Fig. 3d). Thus, both CD74 and CXCR4 are involved in MIF-2-mediated B-cell chemotaxis.

To further test if CXCR4 is a MIF-2 receptor, we performed flow cytometry-based internalization experiments measuring CXCR4 surface levels on primary mouse B cells upon MIF-2 exposure. MIF-2-treatment dose-dependently triggered the internalization of CXCR4 with a maximal effect at 4 nM, a potency comparable to that of the cognate ligand CXCL12 (Fig. 3e). Kinetic studies revealed rapid MIF-2-mediated CXCR4 internalization within 15 min (Fig. 3f). The effect was more potent than that of MIF, for which maximal internalization was achieved at 32 nM within 20 min (Supplementary Fig. 5a, b).

We next took advantage of a yeast receptor assay, previously established to monitor CXCR4 binding and signaling of CXCL12, MIF[52] (Fig. 3g), and plant MIF orthologs[53]. MIF-2 not only concentration-dependently triggered CXCR4-mediated signaling in this system (Fig. 3h), but also was the more potent agonist compared to MIF (Fig. 3i). In silico protein-protein docking simulation using HADDOCK suggested that the interaction between MIF-2 and CXCR4 was overall comparable to the MIF/CXCR4 interaction (Fig. 3j, Supplementary Fig. 6a), also according to the docking parameters HADDOCK score, van der Waals (VDW) energy, electrostatic energy, and buried surface area

(Supplementary Fig. 6b). The most pronounced difference was noted for electrostatic interactions (Supplementary Fig. 6b). Lastly, we employed an in vitro assay to biochemically measure binding of MIF-2 to CXCR4. Relying on the CXCR4 surrogate peptide msR4M-L1, which binds MIF with high affinity[54], we performed fluorescence-spectroscopic titrations to determine the binding affinity between MIF-2 and msR4M-L1. Titration of fluos-msR4M-L1 with MIF-2 provided a dose-dependent increase of fluorescence intensity and suggested a reasonably affine interaction (app. $K_D$ (fluos-msR4M-L1/MIF-2) = 180.82 ± 63.07 nM) (Fig. 3k, l). We also subjected the MIF-2/msR4M-L1 interaction to HADDOCK docking. While HADDOCK score and VDW energy were lower compared to MIF-2/CXCR4 and MIF/CXCR4 and mixed outcomes were seen for electrostatic energy, MIF-2/msR4M-L1 displayed an enlarged buried surface area. Supplementary Fig. 7a illustrates the predicted binding interface between MIF-2 and msR4M-L1 and a superimposition of the MIF-2/CXCR4 and MIF-2/msR4M-L1 complexes (Supplementary Fig. 7b).

Together, the B-cell chemotaxis experiments involving CXCR4 inhibition and internalization experiments, the yeast-CXCR4 signaling results, protein-protein docking, and biochemical data for binding of MIF-2 to the CXCR4 surrogate peptide provided evidence that MIF-2 serves as a non-cognate ligand for CXCR4.

Since MIF, but not CXCL12, was shown to promote LDL receptor (LDLR)-mediated foam-cell formation in macrophages in a CXCR4-dependent manner[54,55], we used this assay to further study the role of the MIF-2/CXCR4 axis. Primary human monocyte-derived macrophages were stimulated with MIF-2 in the presence or absence of 4-IPP, 4-CPPC, or AMD3100, and exposed to fluorescently-labeled LDL (DiI-LDL). MIF-2 stimulation led to an enhancement of LDL uptake, and this effect was inhibited by pretreatment of cells with all inhibitors (Fig. 3m, n), confirming MIF-2- and CXCR4-specificity of the foam-cell-formation effect. Thus, MIF-2 is a ligand of CXCR4 and this axis mediates atherogenesis-relevant activities such as leukocyte migration and foam-cell formation.

## *Mif-2* deficiency attenuates advanced atherosclerosis, lowers body weight, and plasma lipid levels in vivo

We next studied the impact of *Mif-2*-deficiency on advanced atherosclerosis. Eight-week-old female *Mif-2*[−/−]*Apoe*[−/−] versus *Apoe*[−/−] littermates were challenged with a HFD for 12 weeks and atherosclerotic plaques analyzed (Fig. 4a). *Apoe*[−/−] mice developed pronounced lesions in aortic root with a 3-fold increase of lesion areas compared to those measured in the early-stage model (Fig. 4b; compare with Fig. 1b). Confirming the protective effect in the early atherogenesis model, global *Mif-2*[−/−] deletion attenuated lesion formation by ~40% in aortic root (Fig. 4b, c), and ~80% in aortic arch (Fig. 4d). *Mif-2*-deficiency also mitigated vascular inflammation as indicated by reduced lesional macrophage content in *Mif-2*[−/−]*Apoe*[−/−] (Fig. 4e). As expected for advanced atherosclerosis, a necrotic core formed, and it was markedly reduced in *Mif-2*-deficient mice (Fig. 4f), whereas plaque collagen content (Supplementary Fig. 8a, b), fibrous cap thickness (Supplementary Fig. 8c, d), and smooth

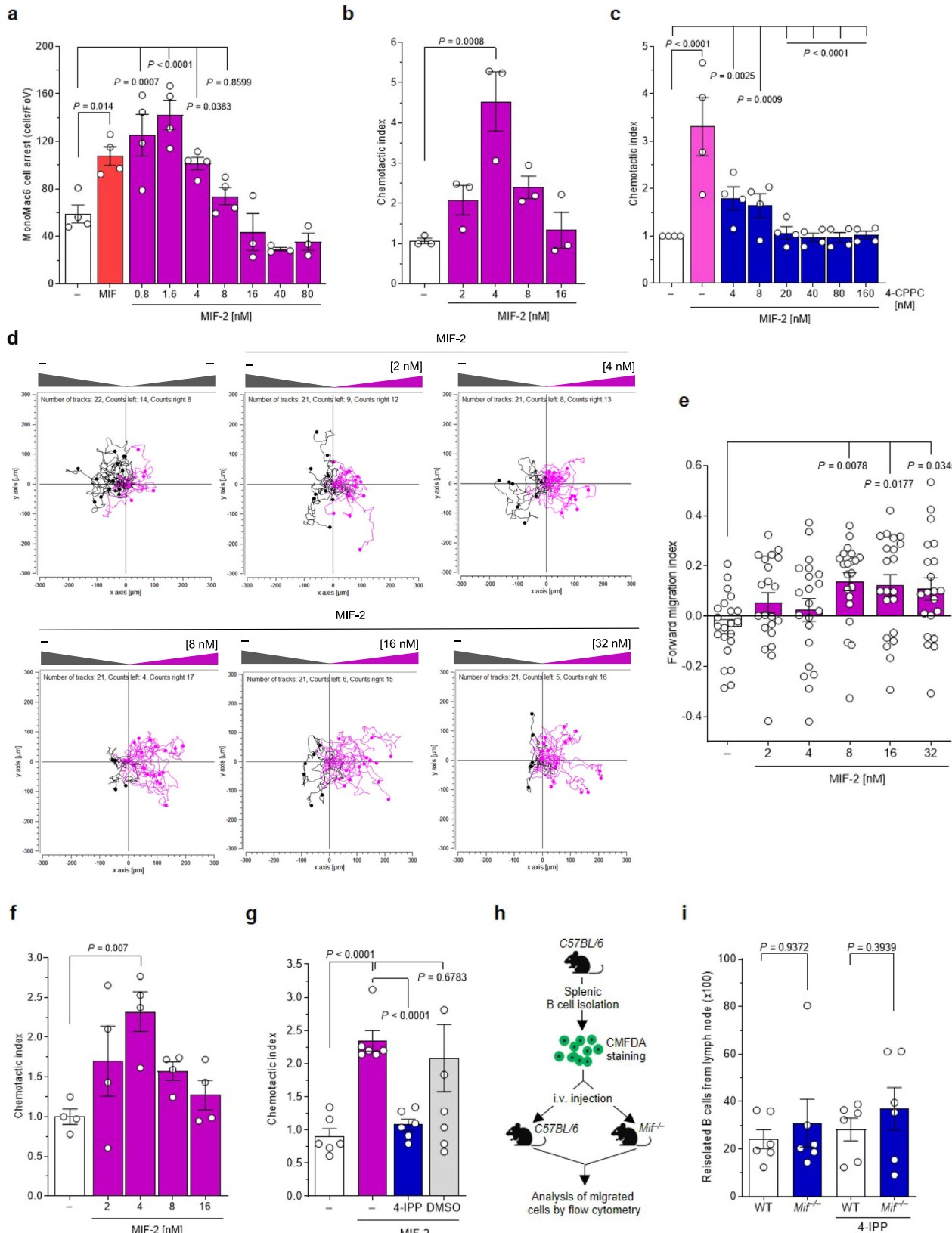

muscle cell counts as read out by α-smooth muscle actin (SMA)-positive lesion area (Supplementary Fig. 9a, b) remained unaffected. Thus, vascular smooth muscle cells (VSMCs) and VSMC-dependent parameters such as collagen content and FC thickness were unchanged between *Mif-2*-deficient *Apoe*⁻/⁻ mice and *Mif-2*-expressing *Apoe*⁻/⁻ controls and likely did not contribute to *Mif-2*-driven foam-cell formation within the

studied 12-week HFD model. The most likely explanation for this notion is that the transdifferentiation of VSMCs into plaque macrophages or CD68+ macrophage-like SMCs in Western-type HFD-fed *Apoe*⁻/⁻ mice does not occur until 12 weeks[56].

On the other hand, the pro-atherogenic effects of MIF-2 might be co-dependent on plaque VSMC-derived MIF-2 as a source of this

**Fig. 2 | MIF-2 promotes atherogenic leukocyte arrest and chemotaxis.**
**a** MonoMac6 adhesion on HAoECs under flow conditions after 2 h of MIF-2 (0.8–80 nM) stimulation compared to 16 nM MIF. For MIF-2 concentrations of 0.8–8 nM, $n = 4$ biological replicates; for 16–80 nM concentrations, $n = 3$ biological replicates. **b** Human peripheral blood monocytes were subjected to different concentrations of MIF-2 for 4 h and chemotaxis (Transwell migration) depicted as chemotactic index ($n = 3$ biological replicates). **c** Effect of 4-CPPC (1 h preincubation) on the migratory effect of 4 nM MIF-2 ($n = 4$ biological replicates each). **d**, **e** MIF-2-elicited (2–32 nM) monocyte chemotaxis in a 3D setup using single-cell tracking in x/y direction in μm. Representative experiments (MIF-2-elicited migration tracks in magenta), unstimulated control ['random motility'] in gray. Tracks of 28–30 randomly selected cells per group recorded every 2 min for 2 h (**d**) and forward migration index plotted (**e**). For MIF-2 concentrations of 2–32 nM, $n = 21$; for the untreated group (–), $n = 22$. The experiment shown is one of three

independent experiments with monocytes from different donors. **f** MIF-2-induced chemotaxis of primary B cells measured by Transwell migration ($n = 4$ biological replicates, 4 h). **g** Effect of the MIF and MIF-2 inhibitor 4-IPP on MIF-2-elicited primary B-cell chemotaxis ($n = 6$ biological replicates). Solvent control: 0.1% DMSO. **h**, **i** Effect of MIF-2 on B-cell homing in vivo. Scheme illustrating the homing experiment. Fluorescently-labeled primary splenic B cells from wild-type (WT) C57BL/6 mice were i.v.-injected into WT or $Mif^{-/-}$ recipients and 'homed' B cells isolated from target organs and quantified (**h**). Quantification of B lymphocytes homed into spleen (**i**). **h** was created in BioRender. Bernhagen, L. 2025 https://BioRender.com/c59v181. Values are represented as means ± SD with individual data points shown; statistical analysis was performed by one-way ANOVA with Dunnett's multiple comparisons (**a**–**f**), two-tailed Student's $t$-test for multiple unpaired comparison (**g**), or Mann–Whitney test (**i**).

mediator. To begin to explore this possibility, we reanalyzed comprehensive single-cell RNA sequencing (scRNAseq) data sets from the studies of Wirka et al.[57] and Pan et al.[58] for plaque VSMC expression of MIF-2. Reanalysis of the scRNAseq data from aortic root of $Apoe^{-/-}$ mice following 8 or 16 weeks of HFD revealed a prominent expression pattern of $Mif$-2 in plaque VSMCs in all three detected SMC sub-types, i.e. SMC1, SMC2, and modulated SMC (Supplementary Fig. 10a–d). Mif-2 expression in plaque SMCs was by far more pronounced than that seen in plaque fibroblasts, macrophages, or endothelial cells. This expression pattern was confirmed by reanalysis of scRNAseq data from aortic root of $Ldlr^{-/-}$ mice following 8, 16, or 26 weeks of HFD. The Mif-2 expression profile in this data set from the other key atherogenic mouse model similarly showed prominent expression of Mif-2 in all three determined SMC sub-types, i.e. SMC1, SMC2, and the so-called ICSs (SMC-derived intermediate cell state; also termed SEM (stem cell, endothelial cell, monocyte)) (Supplementary Fig. 11a–c).

In addition, multiplex cytokine analysis from plasma specimens showed a reduction in numerous inflammatory cytokines including IL-27, IL-13, IL-1β, IL-9, or TNF-α, and the chemokines CCL3 and CCL5 in $Mif$-$2^{-/-}Apoe^{-/-}$ mice (Fig. 4g). Of note, these mice showed a reduction in body size (Fig. 4h) as well as reduced body weight (Fig. 4i; Supplementary Table 2), an observation also made after 4.5-week HFD (Supplementary Fig. 12a). Similar results were obtained for the male cohort (Supplementary Fig. 12b).

Plasma lipid analysis revealed a drop of total cholesterol and triglyceride levels by 10%-20% in the $Mif$-$2^{-/-}Apoe^{-/-}$ mice after both 4.5- and 12-week-HFD, respectively (Fig. 4j, k; Supplementary Fig. 13a, b; Supplementary Table 2). Importantly, when we analyzed the lipoprotein fractions by FPLC, marked reductions of VLDL and LDL were observed in $Mif$-2-deficient $Apoe^{-/-}$ mice compared with control mice, as well as in $Apoe^{-/-}$ mice, in which MIF-2 was pharmacologically inhibited (Fig. 4l; Supplementary Fig. 13c, d). These data suggested that the pro-atherogenic role of MIF-$2$ is not restricted to early stages of atherogenesis but is important in advanced stages of atherosclerosis and influences plasma lipid and lipoprotein levels.

## MIF-2 promotes hepatosteatosis and SREBP activity to regulate lipogenesis in hepatocytes in a CXCR4- and CD74-dependent manner

Considering that circulating lipid levels are associated with liver lipid metabolism, we next analyzed liver parameters. The livers of female $Mif$-$2^{-/-}Apoe^{-/-}$ mice after 12 weeks of HFD were smaller and lighter compared to those from $Apoe^{-/-}$ mice (Fig. 5a, b). A similar difference was noticed in male mice (Supplementary Fig. 12c, d). We then determined the hepatic lipid content following staining of liver sections with HE and ORO. A reduction of neutral lipid deposition was detected in $Mif$-$2^{-/-}Apoe^{-/-}$ mice compared to $Apoe^{-/-}$ mice (Fig. 5c, Supplementary Fig. 14), together suggesting that MIF-2 promotes hepatic lipid accumulation under atherogenic conditions.

To address the possibility that MIF-2 may directly regulate lipid levels in the liver, we determined the effect of MIF-2 on the expression of lipogenic genes in hepatocytes, applying the human hepatocyte cell line Huh-7. Based on previous data suggesting a causal link between hepatic SREBP proteins, a family of key lipogenic transcription factors, and metabolic disease and atherosclerosis[59,60], we hypothesized that SREBPs could play a role in the observed liver lipid phenotype in $Mif$-$2^{-/-}Apoe^{-/-}$ mice. Using RT-qPCR, we assessed the gene expression of $SREBP$-$1$ and $SREBP$-$2$ following MIF-2 stimulation. MIF-2 increased mRNA levels of $SREBP$-$1$ and $SREBP$-$2$ in a dose-dependent manner (Supplementary Fig. 15a, b). SREBP-1 and SREBP-2 promote the transcription of sterol-regulated genes involved in lipogenesis including fatty acid synthase $(FASN)$ and $LDLR$, respectively[61,62]. Accordingly, MIF-2 led to an increase in mRNA levels of both $FASN$ and $LDLR$ (Supplementary Fig. 15a, b). Since SREBPs are activated by proteolytic cleavage, we next checked whether MIF-2 affected SREBP processing. As shown in Fig. 5d, e and 5f, g, incubation of Huh-7 cells with increasing concentrations of MIF-2 resulted in a reduction of the SREBP-1 and -2 precursors pSREBP-1 and pSREBP-2 (-125 kDa) and in an increase in the processed form (nSREBP: -65 kDa). In accord, the SREBP-1 and -2-regulated genes/proteins FASN and LDLR, respectively, were enhanced by 8–16 nM of MIF-2 (Fig. 5d–g). Moreover, applying immunofluorescence staining, we analyzed changes in the subcellular localization of processed SREBP following MIF-2 stimulation, using SREBP-2 as an example. Cells exposed to MIF-2 showed enhanced nSREBP-2 accumulation in the nucleus, whereas untreated cells exhibited broad cytoplasmic SREBP-2 distribution (Fig. 5h). We next validated the effect of MIF-2 on SREBP activation in primary hepatocytes. Hepatocytes isolated from the livers of WT mice were treated with different concentrations of MIF-2. MIF-2 increased the proteolytic activation of SREBP-1 and SREBP-2, respectively, in a concentration-dependent manner (Supplementary Fig. 16a, b). Importantly, processed nSREBP-1 and nSREBP-2 were appreciably expressed in the livers of $Apoe^{-/-}$ mice after 12 weeks of HFD, whereas their levels were strongly reduced in liver tissue from $Mif$-$2^{-/-}Apoe^{-/-}$ mice (Fig. 5i). Evidence that the SREBP-regulatory activity of MIF-2 in hepatocytes is associated with lipid accumulation/lipogenesis in these cells was obtained by an in vitro steatosis assay using an ORO staining readout. The previously identified peak concentrations (4–8 nM) of MIF-2-enhanced SREBP activity promoted lipid formation in Huh-7 cells similar to supplementation with 1 mM oleic acid (OA), while no additive effect was observed when cells were co-stimulated with MIF-2 and OA (Supplementary Fig. 17a, b). In addition to modulating lipid accumulation, MIF-2 also enhanced triglyceride synthesis in Huh-7 cells, as demonstrated by a triglyceride assay (Supplementary Fig. 17c). As ORO primarily stains neutral lipids and triglycerides and as triglycerides were directly measured by the triglyceride-Glo assay, this together suggested that MIF-2 is a regulator of SREBP-1/-2 activation, lipogenesis and lipid esterification in hepatocytes. To deeper characterize the impact of MIF-2 on hepatic lipid pathways, we performed bulk RNA sequencing

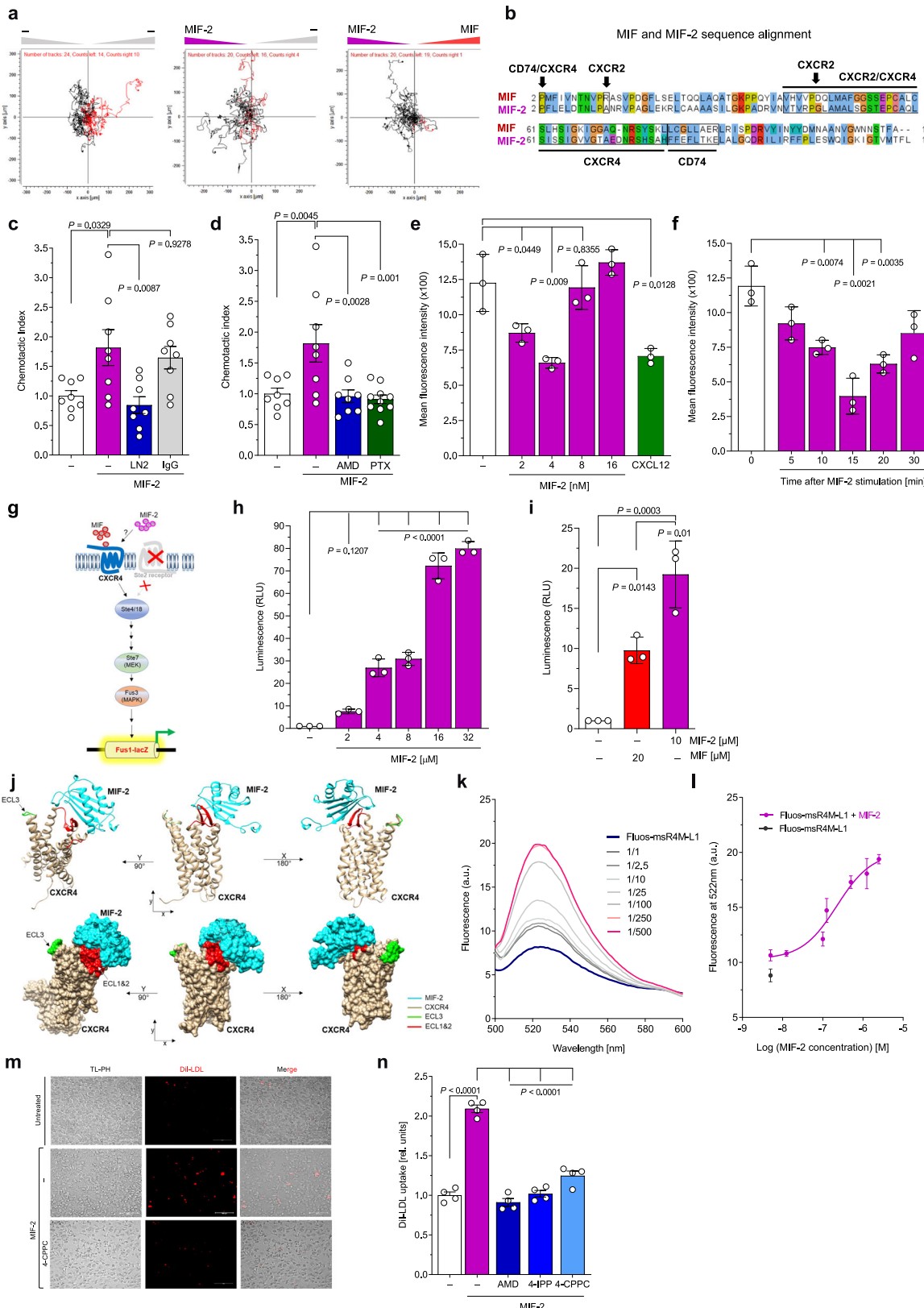

(RNAseq) from liver sections and liver lysates, as well as a comprehensive lipidomic analysis from liver tissue comparing *Mif-2⁻/⁻Apoe⁻/⁻* mice with *Apoe⁻/⁻* controls (Fig. 5j–m, Supplementary Fig. 18, Fig. 6, respectively). Using a recently established procedure for RNAseq from PFA-fixed OCT-embedded frozen tissue sections[63], we first extracted mRNA from 4 liver sections of each group and performed RNAseq.

Volcano plot and heat map analysis showed numerous differentially expressed genes (DEGs) associated with hepatic lipogenesis that were significantly enriched in the *Apoe⁻/⁻* group or downregulated in *Mif-2⁻/⁻Apoe⁻/⁻* mice (Fig. 5k, l). Prominent examples are *Apolipoprotein* A4 (*Apoa4*), *Acetyl-coenzyme A acyltransferase 2* (*Acat2, Acaa2*), *Proprotein convertase subtilisin/kexin type 9 (Pcsk9), Acyl-CoA thioesterase 1 (Acot1),*

**Fig. 3 | MIF-2 is a potent chemokine and CXCR4 ligand, and promotes foam-cell formation. a** Direct comparison of the chemotactic potency of MIF and MIF-2 on primary B lymphocytes by 3D chemotaxis (recorded at 2 min intervals for 2 h). Left, buffer in both chambers (control); middle, MIF-2 *versus* buffer; right, MIF-2 *versus* MIF competition. Experiment shown is one of three independent experiments with B cells from different mice. **b** Alignment of human MIF-2 and MIF amino acid sequences with residues implicated in receptor binding (aligned by ClustalW) indicated. Identical/homologous residues are highlighted by thesame colors. **c**, **d** Effect of MIF-2 on B-cell migration (Transwell setup) and dependence on CD74 and CXCR4. Anti-CD74 antibody (LN2) (**c**), CXCR4 antagonist AMD3100 (AMD), and pertussis toxin (PTX) (**d**) were applied to probe receptor specificity. Untreated (−), MIF-2 and AMD3100, $n = 8$; PTX, $n = 10$. **e**, **f** Effect of MIF-2 on CXCR4 internalization in primary mouse B cells. Concentration-dependency at 15 min with CXCL12 as positive control (**e**). Time-dependent effect at 4 nM MIF-2 (**f**). **g**–**i** Effect of MIF-2 on CXCR4 signaling in *S. cerevisiae* reporter-cell system. Scheme of MIF-2/CXCL12-induced signaling in yeast-CXCR4 reporter assay (**g**). Concentration-dependent reporter activity ($n = 3$ biological replicates); **h** and MIF-2 vs MIF comparison at 10 *versus* 20 μM ($n = 3$ biological replicates; **i**). **j** In situ molecular-docking simulation of monomeric MIF-2 (blue) and CXCR4 (yellow-gold) using HADDOCK, shown in ribbon (top) and surface area (bottom) views. CXCR4 extracellular loops: ECL1, ECL2 (red), ECL3 (green). **k**, **l** Fluorescence-spectroscopic titration of CXCR4 surrogate peptide Fluos-msR4M-L1 (5 nM) with increasing MIF-2 concentrations. Emission spectra (**k**), binding curve at 522 nm ($n = 3$ independent experiments; **l**). **m**, **n** Effect of MIF-2 on DiI-LDL uptake in primary human monocyte-derived macrophages and dependence on CXCR4. MIF-2 (80 nM), AMD3100, 4-IPP, and 4-CPPC (10 μM each). Representative images (**m**) and quantification ($n = 4$ independent experiments; 9 fields-of-view each) (**n**). Scale bar: 100 μm. Values are expressed as means ± SD. Statistical analysis: two-tailed unpaired Student's *t*-test (**e**, **f**) and one-way ANOVA with Tukey's multiple comparisons (**c**), with Dunnett's multiple comparisons (**d**, **h**, **i**), or with Šídák's multiple comparisons (**n**).

*and Carboxylesterase 2E (Ces2)*. As expected, *Srepb1* was downregulated in *Mif-2⁻/⁻Apoe⁻/⁻* mice as well ($P = 0.08$). Moreover, GO pathway enrichment analysis revealed that the key enriched pathway was 'fatty acid metabolic processes' ($-\log_{10}(P_{adj}) < 10^{-11}$). Other terms related to hepatic lipogenesis were also highly significantly enriched in the *Apoe⁻/⁻* group, e.g. 'generation of precursor metabolites' and 'purine (ribo)nucleotide metabolics' (Fig. 5m). This signature was overall confirmed by a more comprehensive RNAseq from whole liver lysates ($n = 7$ per group) with again 'fatty acid metabolism' and 'regulation of lipid metabolic processes' representing the key enriched terms of the GO pathway analysis that were significantly downregulated in *Mif-2⁻/⁻Apoe⁻/⁻* mice (Supplementary Fig. 18). Additional genes identified to be downregulated in *Mif-2⁻/⁻Apoe⁻/⁻* livers by this transcriptome analysis encompassed for example *17β-hydroxysteroid dehydrogenase type 13 (Hsd17b13)*, a gene that is significantly upregulated in the liver of patients with MASLD and that enhances lipogenesis; the transcript for Oxysterol-binding protein-related protein 3 (*Osbpl3*), an intracellular hepatocyte lipid receptor; or the cytochrome P450 monooxygenase *Cyp4a12a*, which is involved in fatty acid metabolism and generation of oxylipins including omega-oxidized fatty acids (Supplementary Fig. 18).

We next, performed RT-qPCR to validate selected DEGs and assess the expression of other potentially relevant ones (Supplementary Fig. 19). Significantly downregulated genes in liver of *Mif-2⁻/⁻Apoe⁻/⁻* mice with relevance for SREBP-regulated pathways, fatty acid synthesis, and triglyceride generation included *Srebp cleavage-activating protein (Scap)*, *Fatty acid synthase (Fasn)*, *Srebp-2*, and *Monoacylglycerol O-acyltransferase 1 (Mogat1)*. Additionally, genes related to lipid uptake and cholesterol biosynthesis such as *Peroxisome proliferator-activated receptor gamma (Ppar-γ, Pparg)*, the scavenger receptor *Cluster of differentiation 36 (Cd36)*, *Scavenger receptor class B type 1 (Sr-b1)*, *3-hydroxymethylglutaryl-coenzyme A synthase-1 (Hmgcs1)*, *3-hydroxy-3-methyl-glutaryl coenzyme A reductase (Hmgcr)*, or *Acetyl-CoA acetyltransferase 1 (Acat-1)* were found to be significantly reduced in liver tissue of *Mif-2⁻/⁻Apoe⁻/⁻* mice (Supplementary Fig. 19). The observed reduction in *Srebp-2* also has a significance for cholesterol metabolism. Notably, genes involved in β-oxidation and bile acid synthesis, including *Carnitine palmitoyltransferase 1 (Cpt1)*, *Acyl-CoA oxidase 1 (Acox-1)*, *Cholesterol-7-alpha-hydroxylase (Cyp7a1)*, or *Sterol-12-α-hydroxylase (Cyp8b1)* were likewise significantly downregulated in the liver of these mice (Supplementary Fig. 19).

Most strikingly, the lipidomic analysis revealed a substantial impact of *Mif-2* deficiency on hepatic lipid contents. To investigate potential lipidomic differences between the livers of atherogenic *Apoe⁻/⁻* and *Mif-2⁻/⁻Apoe⁻/⁻* mice after 12 weeks HFD, we used shotgun lipidomics applying a differential ion mobility-based methodology (termed DMS-SLA), a well-established method for comprehensive lipid analysis from plasma, tissue lysates, or cell cultures (Supplementary

Fig. 20a). First, principal component analysis (PCA) was performed on the entire lipidomic dataset and showed a pronounced separation between liver extracts from *Apoe⁻/⁻* and *Mif-2⁻/⁻Apoe⁻/⁻* mice (Supplementary Fig. 20b, c). After data quality control, we were able to reliably quantify 708 lipid species from 20 different lipid classes. In line with the HE and ORO liver staining results and the transcriptomic data, the three superordinate classes triacylglycerides (TG), diacylglycerides (DG), and cholesterol esters (CE), were markedly and significantly downregulated in the liver extracts of *Mif-2⁻/⁻Apoe⁻/⁻* mice (Fig. 6a–c). The in-depth lipidomic analysis then revealed that the reduction in TGs was based on a reduction of almost all TG species across the entire spectrum from 42 to 60 carbon atoms. Significantly reduced DG species contained FA_16:1, FA_20:3, and FA_22:5; CE species reduced in *Mif-2⁻/⁻Apoe⁻/⁻* mice compared to *Apoe⁻/⁻* mice were CE_24:0 and CE_22:2 (Fig. 6d). While hexosylceramides (HexCER) showed a significant increase of mainly very-long-chain fatty acids in liver tissue of *Mif-2⁻/⁻Apoe⁻/⁻* mice, no significant changes were observed for ceramides (Cer d18:1), dihydroceramides (Cer d18:0), free fatty acids (FA), lactosylceramides (LacCER), lysophosphatidylcholines (LPC), lysophosphatidylethanoamines (LPE), lysophosphatidylserines (LPS), phosphatidic acids (PA), phosphatidylcholines (PC), phosphatidylethanolamines (PE), phosphatidylglycerols (PG), phosphatidylinositols (PI), phosphatidylserines (PS), or sphingomyelins (SM) (Fig. 6d, Supplementary Fig. 21). Together, the substantial reductions of TG, DG, and CE species confirmed the notion that *Mif-2* deficiency reduces hepatic lipogenesis and protects from hepatosteatosis.

Some of the identified metabolic genes have been correlated with upstream pathways such as the AMP kinase (AMPK) and PI3 kinase (PI3K)/AKT pathway. AMPK negatively regulates the transcriptional activity of SREBPs to mitigate hepatosteatosis and atherogenesis[59,64] and MIF activates hepatic AMPK, an effect contributing to its protective effect in fatty liver disease[24]. The PI3K/AKT pathway 'activates' SREBPs to promote lipogenesis[65,66]. We therefore tested the specific effect of MIF-2 on these upstream regulators. Huh-7 cells were stimulated with MIF-2 and phosphorylated AMPK (pAMPK) levels analyzed by immunoblotting. MIF-2 inhibited pAMPK in a concentration-dependent manner with peak ~20–30% reduction compared to buffer-treated cells (Supplementary Fig. 22a, b). Thus, MIF-2 had an opposite effect on hepatocyte AMPK as MIF. In contrast, AKT phosphorylation as a readout for PI3K/AKT activation was enhanced by MIF-2 (Supplementary Fig. 22c, d). Of note, the latter conclusion was confirmed by comparing pAkt and total Akt levels in liver lysates of *Mif-2⁻/⁻Apoe⁻/⁻* mice with those of *Apoe⁻/⁻* mice. Supplementary Fig. 22e shows a substantial reduction of pAkt/Akt ratios in *Mif-2*-deficient liver tissue. As Western diet feeding, atherogenesis, and reduced AKT activity have been associated with insulin resistance[67], it is tempting to speculate whether MIF-2 would impact insulin resistance in hepatocytes. Such experiments would need to be done in a mouse model of

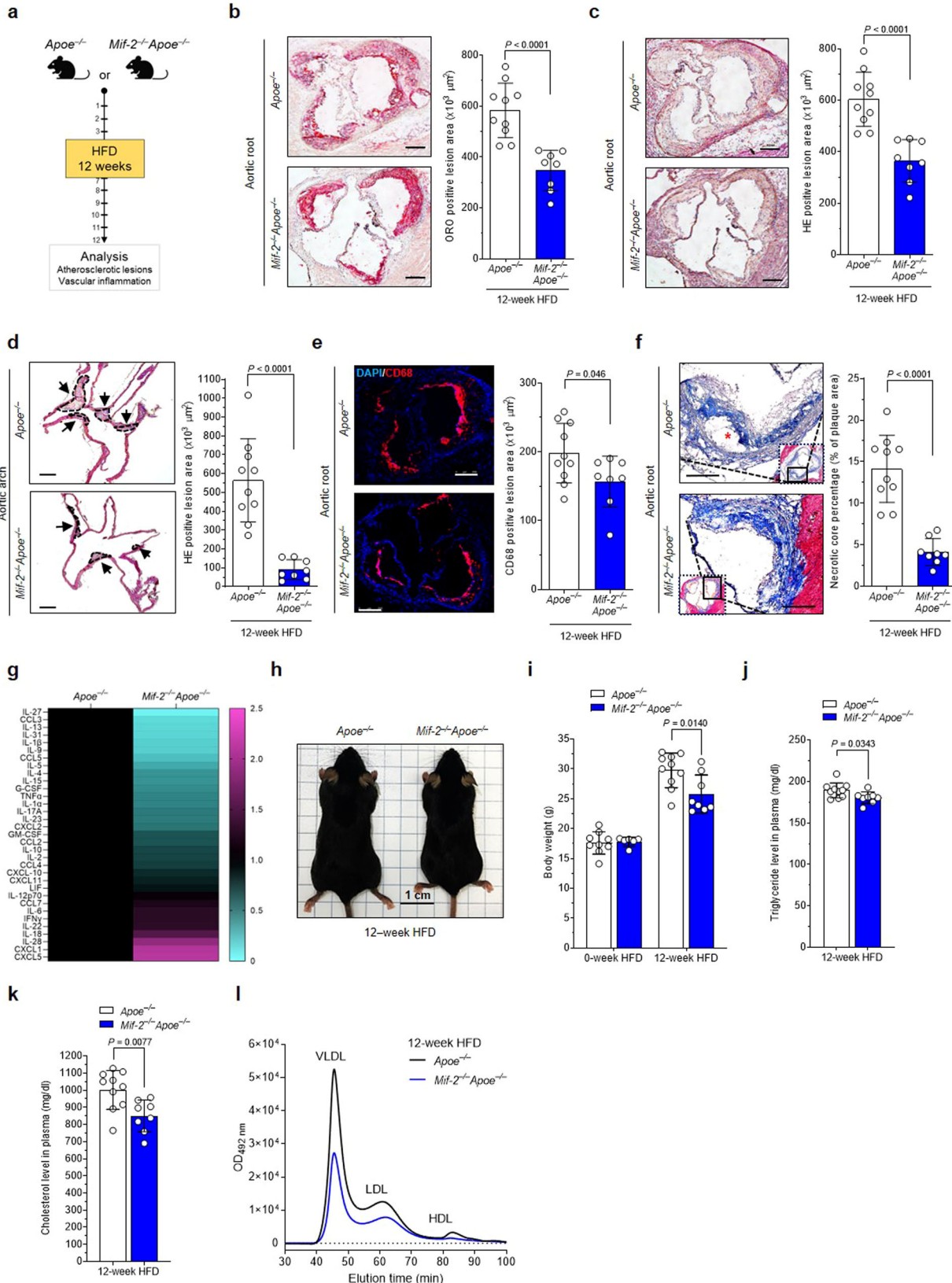

atherogenesis, as the only previous study, in which effects of MIF-2/D-DT on glucose tolerance were assessed, was done in a classical *db/db* mouse model of diabetes/obesity[40]. Together, MIF-2-driven SREBP activation and lipogenesis in hepatocytes appeared to be mediated by upstream PI3K/AKT signaling, while the inhibitory AMPK pathway is attenuated by MIF-2.

**MIF-2-mediated SREBP activation and lipogenic gene expression in hepatocytes is dependent on the CXCR4/CD74 axis**

We next aimed to clarify which MIF-2 receptor would mediate the observed signaling effects and MIF-2-triggered hepatic lipogenesis. Both CD74 and CXCR4 are candidate receptors, because we found them to be expressed on Huh-7 cells as revealed by

**Fig. 4 | *Mif-2* deficiency ameliorates advanced atherosclerotic lesion progression in hyperlipidemic *Apoe*$^{-/-}$ mice. a** Experimental outline: female *Apoe*$^{-/-}$ and *Mif-2*$^{-/-}$*Apoe*$^{-/-}$ mice fed a cholesterol-rich HFD for 12 weeks. **b–d** Effect of *Mif-2* deficiency on lesion formation in aortic root. ORO staining (**b**) and HE staining (**c**) of aortic root and corresponding quantification. HE staining of aortic arch and quantification (**d**). Data points in (**b–d**) represent *n* = 10 for *Apoe*$^{-/-}$ and *n* = 8 for *Mif-2*$^{-/-}$*Apoe*$^{-/-}$; 12 serial sections per mouse; scale bar: 200 μm. **e** Same as (**b, c**) except that the sections were stained for CD68+ macrophages (red) and DAPI (blue) (*Apoe*$^{-/-}$: *n* = 10, *Mif-2*$^{-/-}$*Apoe*$^{-/-}$: *n* = 8). **f** Effect of *Mif-2* deficiency on necrotic core formation (Masson staining). Typical necrotic core marked by red asterisk (*Apoe*$^{-/-}$: *n* = 10, *Mif-2*$^{-/-}$*Apoe*$^{-/-}$: *n* = 8; scale bar: 200 μm). **g** Quantitation of plasma cytokines/chemokines from 4 mice per group. Heatmap of 36 ProcartaPlex mouse cytokine/chemokine array result. Signals from *Apoe*$^{-/-}$ mice normalized to 1 and compared with those from *Mif-2*$^{-/-}$*Apoe*$^{-/-}$ mice (upregulated cytokines in magenta, downregulated in cyan). **h, i** Comparison of body weights between *Mif-2*$^{-/-}$*Apoe*$^{-/-}$ and *Apoe*$^{-/-}$ mice at 0 and 12 weeks of HFD. Representative *Apoe*$^{-/-}$ mouse compared to *Mif-2*$^{-/-}$*Apoe*$^{-/-}$ mouse after 12 weeks HFD (**h**) and corresponding body weights at 0 and 12 weeks of HFD (*Apoe*$^{-/-}$: *n* = 9; *Mif-2*$^{-/-}$*Apoe*$^{-/-}$: *n* = 6 (0-week HFD), *n* = 8 (12-week HFD)) (**i**). **j, k** Comparison of plasma lipid levels between *Mif-2*$^{-/-}$*Apoe*$^{-/-}$ and *Apoe*$^{-/-}$ mice fed a HFD for 12 weeks; triglycerides (**j**), total cholesterol (**k**) (*Apoe*$^{-/-}$: *n* = 10, *Mif-2*$^{-/-}$*Apoe*$^{-/-}$: *n* = 8). **l** Lipoprotein profiles of *Apoe*$^{-/-}$ *versus Mif-2*$^{-/-}$*Apoe*$^{-/-}$ mice after 12-week HFD. Representative FPLC chromatograms of lipoprotein fractions with peaks for VLDL, LDL, and HDL. **a** was created in BioRender. Bernhagen, L. 2025 https://BioRender.com/c59v181. All values are represented as means ± SD and were analyzed using unpaired two-tailed Student's *t*-test.

immunofluorescence (Supplementary Fig. 23a) and flow cytometry (Supplementary Fig. 23b, c). To investigate their functional involvement, we tested MIF-2-elicited SREBP activation in the presence *versus* absence of receptor inhibitors. Blockade of CD74 with the neutralizing antibody LN2 resulted in a pronounced reduction of processed nSREBP-1/2 levels, while inhibition of CXCR4 with AMD3100 showed a partial decrease (Fig. 7a, b). Dual inhibition of CXCR4 and CD74 with both inhibitors fully abrogated the effect of MIF-2 on SREBP-1/2 cleavage and the downstream targets FASN and LDLR (Fig. 7a, b), providing plausible evidence that the MIF-2-mediated lipogenic effects in Huh-7 are dependent on the CXCR4/CD74 axis.

We next measured the effect of MIF-2 on hepatic LDL uptake and the contribution of CXCR4 and CD74 to this process. Huh-7 cells were pretreated with AMD3100, LN2 antibody, or control IgG, alone or in combination, stimulated with MIF-2 and exposed to LDL. ORO staining revealed that MIF-2 stimulation led to enhanced LDL uptake, an effect that was attenuated by CXCR4 or CD74 inhibition and fully blocked by AMD3100/LN2 dual inhibition (Fig. 7c). Thus, MIF-2-driven LDL uptake may contribute to hepatic lipid accumulation and this effect is mediated by CD74 and CXCR4.

Considering that MIF-2 interacts with CXCR4, we next determined whether *Cd74* deficiency would affect MIF-2-induced CXCR4 internalization using the primary mouse B-cell model again. In contrast to B cells from WT mice (see Fig. 3e, f for comparison), treatment of B cells from *Cd74*$^{-/-}$ mice with different concentrations of MIF-2 (or CXCL12 as control) for 20 min did not result in internalization of CXCR4 (Fig. 7d), supporting the notion that the effects of MIF-2 involve both CXCR4 and CD74. Interestingly, the kinetics of CXCL12-mediated CXCR4 internalization in *Cd74*-deficient B cells showed a significant but delayed effect, with a minimum after 20–30 min (Supplementary Fig. 24), lending further support to a functional involvement of CD74 in agonist-induced activation of CXCR4. One mechanistic possibility underlying the joint involvement of CD74 and CXCR4 is receptor complex formation. To begin to study such complexes, we investigated CXCR4/CD74 heterocomplex formation using the in situ proximity ligation assay (PLA), a method previously used to detect interactions between MIF receptors[22]. NIH/3T3 fibroblasts were transfected with c-Myc-tagged CXCR4 and FLAG-tagged CD74 plasmids, and PLA signals visualized using anti-c-Myc and anti-FLAG antibodies. Figure 7e shows the presence of positive PLA signals, indicating interaction between CXCR4 and CD74.

To further verify complex formation, we applied Förster resonance energy-transfer (FRET) in combination with fluorescence-lifetime imaging microscopy (FLIM), i.e. FLIM-FRET on a two/(multi)-photon laser-scanning microscope (TPLSM), a powerful method to detect dynamic associations of proteins at cellular level[68,69]. CXCR4/CD74 interaction was first examined in HEK293 cells[70]. Cells were transfected with ECFP-tagged CXCR4 and EYFP-CD74 fusion proteins, serving as donor and acceptor, respectively. After fixation, colocalization analysis and FLIM-FRET measurements using time-correlated single photon counting (TCSPC)-FLIM were performed (Supplementary Fig. 25a). TPLSM showed a pronounced colocalization of the receptors on the cell membrane (Supplementary Fig. 25b) with a robust FRET efficiency of approximately 50%, as determined with SP8 Fast Lifetime Contrast (FALCON) (Supplementary Fig. 25c, d), indicating complex formation. To determine if complex formation also occurs in hepatocytes, we used the same experimental setup in Huh-7 cells (Fig. 7f). Indeed, TPLSM images showed colocalization of CXCR4 and CD74 (Fig. 7g). Next, we determined the average fluorescence lifetime of the donor ECFP-CXCR4 in the absence of the acceptor EYFP-CD74 to be 2.52 ns (Supplementary Fig. 26a–d), consistent with values published previously[71,72]. The determined value was used for FLIM-FRET evaluation with FALCON and revealed a FRET efficiency of 17–20% in Huh-7 (Fig. 7h, i). Although this value was lower than that obtained in HEK293 cells, it suggested a molecular interaction between CXCR4 and CD74.

We next asked whether complex formation is inducible by MIF-2. Again, we initially performed these experiments in HEK293 cells. The resulting fluorescence lifetime of ECFP-CXCR4 (donor) and FLIM-FRET efficiency were live-monitored at one-minute intervals during exposure to MIF-2 (Fig. 7j). The colocalization pattern of CXCR4/CD74 heteromers was equally well detected in this live cell setup (Fig. 7k). Of note, stimulation of ECFP-CXCR4/EYFP-CD74-double-transfected HEK293 cells with MIF-2 triggered a time-dependent decrease in ECFP-CXCR4 lifetime from -2.47 to -2.00 ns and a progressive increase in FLIM-FRET efficiency as shown by the decay curves (Fig. 7l) and FLIM-FRET efficiency images (Fig. 7m, n), respectively. Treatment of co-transfected cells with MIF also reduced ECFP-CXCR4 lifetime, but to a lesser extent (Supplementary Fig. 27a, b). Lastly, we switched back to Huh-7 hepatocytes again and tested for inducibility of receptor complex formation by MIF-2. MIF-2 led to receptor colocalization in the cell membrane and an increase in FLIM-FRET efficiency compared to control cells (Fig. 7o–r), indicating that MIF-2 also triggers formation of the CXCR4/CD74 receptor complex in hepatocytes.

## MIF-2 expression is upregulated in unstable human atherosclerotic plaques and correlates with clinical parameters in CAD

Our data suggested that MIF-2 is a chemokine that promotes vascular inflammation and atherogenesis. Moreover, MIF-2 enhanced foam-cell formation, and increased hepatic lipogenic gene expression and hepatosteatosis. Atherosclerotic lesions of *Mif-2*$^{-/-}$*Apoe*$^{-/-}$ mice also displayed reduced necrotic cores, altogether suggesting that MIF-2 has a role in advanced atherosclerosis.

To explore the translational significance of this notion, we analyzed the expression of MIF-2 in stable and unstable carotid artery plaques from patients undergoing endarterectomy. The plaque phenotype (stable/unstable) was assigned based on the American Heart Association (AHA) classification[73] and fibrous cap thickness (below/above 200 μm) according to Redgrave et al.[74]. Scouting data obtained by immunohistochemical staining of MIF-2 with an established antibody[29] in plaque sections suggested that MIF-2 is abundantly present in unstable plaque tissue, whereas less MIF-2 immunopositivity

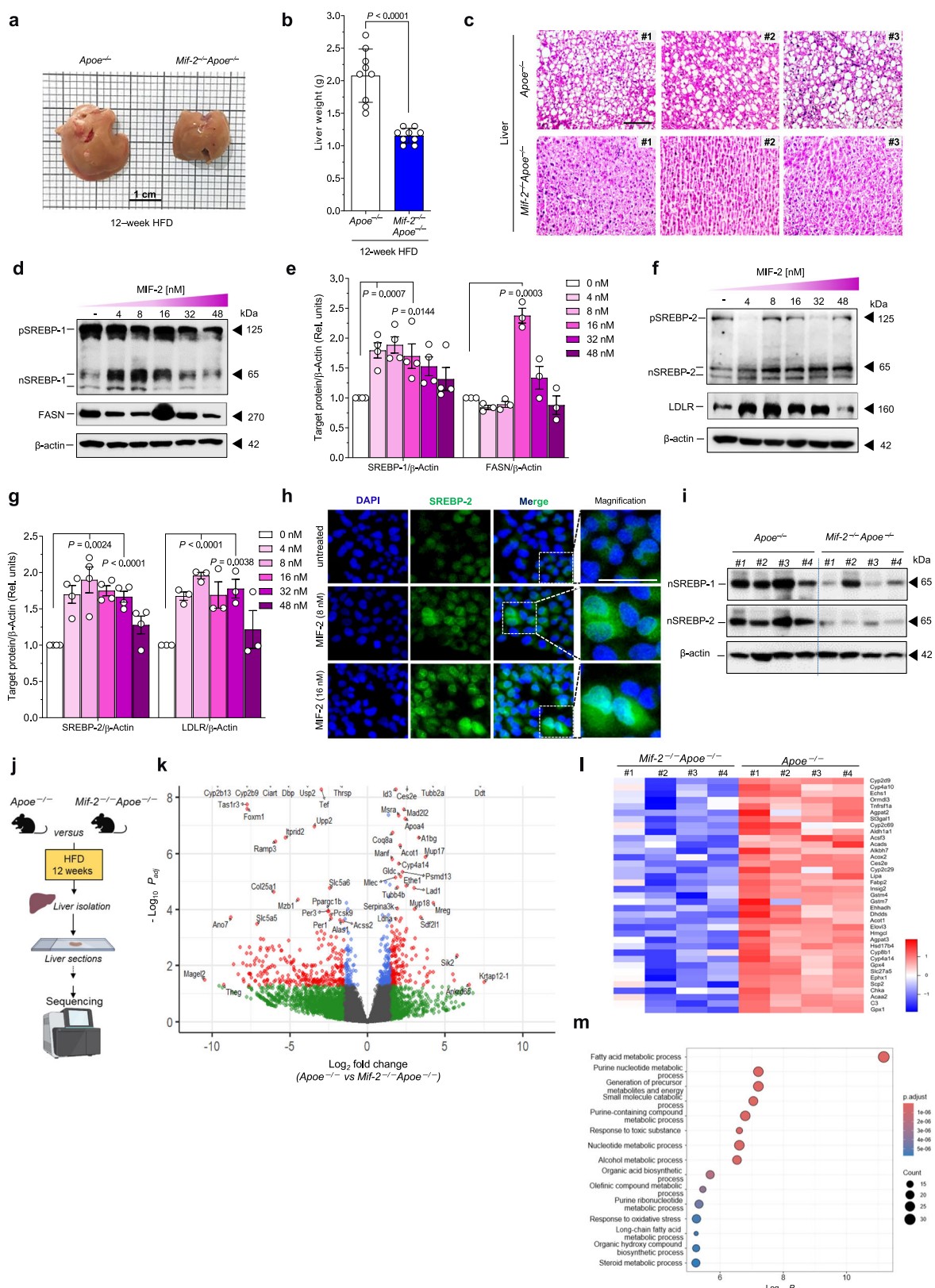

was detected in stable plaques (Fig. 8a). For control, sections from healthy vessel tissue were stained with MIF-2 positivity confined to the endothelial lining. This finding encouraged us to quantify *MIF-2* expression in plaque tissue by RT-qPCR. Comparing specimens from stable *versus* unstable plaques (15 patients each[54]), *MIF-2* expression was significantly higher in unstable plaques (Fig. 8b). This notion was confirmed, when we reanalyzed a publicly available RNAseq data set of

six stable *versus* five unstable (ruptured) CEA plaques, as classified by macrophage-rich regions[75]. *MIF-2*, but not *MIF*, expression was significantly elevated in unstable compared to stable sections (Supplementary Fig. 28). Reanalysis of a single-cell RNAseq dataset (GSE159677) of plaques from CEA patients according to Alsaigh et al.[76] showed that MIF-2, as well as MIF, is markedly expressed in inflammatory cell populations, particularly in macrophages, monocytes, and

**Fig. 5 | MIF-2 increases hepatosteatosis and elicits SREBP-mediated lipogenesis in hepatocytes. a, b** Representative liver images from *Apoe*$^{-/-}$ and *Mif-2*$^{-/-}$*Apoe*$^{-/-}$ mice after 12 weeks on cholesterol-rich HFD (**a**) and corresponding liver weights (*n* = 9 mice per group) (**b**). **c** HE-stained liver sections from *Apoe*$^{-/-}$ and *Mif-2*$^{-/-}$*Apoe*$^{-/-}$ mice after 12 weeks HFD (*n* = 3 mice; scale bar: 200 μm). White areas indicate lipid accumulation. **d–g** Effect of MIF-2 on SREBP activation and lipogenic gene expression in Huh-7 cells. Protein levels of SREBP-1 precursor (pSREBP-1), nuclear SREBP-1 (nSREBP-1), and FASN analyzed via Western blot (**d**) and corresponding densitometric quantification relative to β-actin (**e**) (SREBP-1/β-actin: *n* = 4; FASN/β-actin *n* = 3). **f, g** Same as (**d, e**) except that nSREBP-2 and LDLR were analyzed (SREBP-2/β-actin: *n* = 4; LDLR/β-actin: *n* = 3). **h** MIF-2 stimulation enhances nuclear translocation of nSREBP-2 in Huh-7 cells compared to buffer. Immunofluorescent images show SREBP-2 (green), DAPI (blue) and magnified images in right panel (representative of two separate experiments, scale bar: 40 μm). **i** Protein levels of processed SREBP (nSREBP)-1/-2 in liver lysates from *Apoe*$^{-/-}$ and *Mif-2*$^{-/-}$*Apoe*$^{-/-}$ mice (*n* = 4) analyzed by Western blot. **j–m** Bulk RNAseq analysis of liver sections from *Apoe*$^{-/-}$ and *Mif-2*$^{-/-}$*Apoe*$^{-/-}$ mice after 12 weeks of cholesterol-rich HFD. **j** Schematic of workflow (Liver, liver section, and RNA sequencing equipment icons were created with BioRender.com). **k** Volcano plot of differential gene expression. Red dots: significant genes ($P_{adj}$ < 0.05; log$_2$-fold change>1.5; up in *Apoe*$^{-/-}$, down in *Mif-2*$^{-/-}$*Apoe*$^{-/-}$ mice); dark-gray: non-significant (log$_2$-fold change<1.5); green dots: log$_2$-fold change > 1.5 and $P_{adj}$ > 0.05; blue: log$_2$-fold change < 1.5 and $P_{adj}$ < 0.05. **l** Heatmap of significantly changed genes (*P* < 0.05) showing many linked to lipid metabolism (*n* = 4 mice per group). **m** GO pathway analysis (dot plot representation) showing terms significantly enriched in *Apoe*$^{-/-}$ compared to *Mif-2*$^{-/-}$*Apoe*$^{-/-}$ mice. Dot size (counts) represents number of genes populating a term; color code indicates significance ($-\log_{10}(P_{adj})$). **j** was created in BioRender. Bernhagen, L. 2025 https://BioRender.com/c59v181. All values are means ± SD with individual data points shown; statistical analysis by two-tailed Student's *t*-test.

dendritic cells (Supplementary Fig. 29). Together with the RT-qPCR analysis of our CEA cohort, this is line with the notion that MIF-2 expression is increased in unstable carotid plaques and may contribute to plaque destabilization and rupture.

To further study the role of MIF-2 in atherosclerotic diseases, we determined the concentration of MIF-2 in plasma specimens of 149 patients with CAD by ELISA. MIF-2 plasma levels in healthy individuals are typically in the range of 5–20 ng/mL and have been found to be similar to those of MIF[29,30,77,78]. We found that MIF-2 was significantly increased in CAD patients compared with MIF (Fig. 8c; median MIF-2 levels: 24.27 ± 16.02 ng/mL; MIF: 19.03 ± 8.13 ng/mL; *P* < 0.0001). This was accompanied by a correlation between the plasma concentrations of both proteins (Fig. 8d; *R* = 0.5130, *P* < 0.0001). We next determined the correlation between plasma MIF-2 and CAD severity as determined by coronary angiography, differentiating CAD patients into sub-cohorts of acute coronary syndrome (ACS) and chronic coronary syndrome (CCS). The baseline characteristics of patients classified into CCS (*n* = 85) and ACS (*n* = 47) sub-groups are in Supplementary Table 3. Volcano plot analysis illustrating parameters significantly changed in ACS *versus* CCS patients demonstrates that an increase of MIF and MIF-2 correlated with clinical parameters of ACS, i.e. leukocyte count, left ventricular ejection fraction (LVEF), creatine kinase (CK), and C-reactive protein (CRP) (Supplementary Fig. 30). MIF-2 concentrations were significantly higher in patients with ACS than in those with CCS (Fig. 8e; 26.64 ± 8.70 *versus* 23.11 ± 7.34; *P* < 0.05). Similarly, MIF-2 levels were higher in patients with a high vessel disease score (Fig. 8f; 25.28 ± 8.12 *versus* 21.28 ± 6.58; *P* < 0.05). We next analyzed associations between the determined MIF-2 levels (and MIF for comparison), with available clinical CAD baseline parameters, which in addition to age, gender, and cardiovascular risk factors, included CVD laboratory parameters and medication (Fig. 8g; *n* = 132 CAD patients). In addition to the association with MIF, elevated MIF-2 showed significant (*P* < 0.05) coefficients of Pearson correlation ($r_p$) with smoking status, leukocyte counts, and CK levels, whereas an inverse correlation was observed for LVEF, in line with the determined association with CVD risk factors and cardiac function. The result was similar but not identical for MIF, which showed a positive correlation with mitral regurgitation, leukocyte counts, and CK, and a negative one for LVEF and the glomerular filtration rate (GFR). A similar picture was obtained, when the analysis was separately performed with the ACS and CCS sub-groups (Supplementary Fig. 31a, b). Next, we performed an orthogonal partial least square discriminant analysis (OPLS-DA) to determine differences in metabolic and inflammatory profiles (cholesterol, CK, creatinine, platelets, leukocytes, and CRP) including MIF and MIF-2 between the ACS and CCS sub-group (Fig. 8h), which was in line with the differential correlation patterns of MIF and MIF-2 between the ACS and CCS sub-groups. Lastly, linear regression analysis with forward variable selection after adjustment for arterial hypertension, active smoking, leukocytes, CRP, troponin I, CK, acetylsalicylic acid (ASA),

adenosine diphosphate receptor P2Y$_{12}$ inhibitors, beta-blockers, and statins indicated that MIF-2 qualifies as a risk factor for CAD (odds ratio [OR]: 0.27; 95% confidence interval [CI]: 0.07-0.47; *P* = 0.01) (Supplementary Table 4). Together, these analyses suggested that MIF-2 levels are associated with the severity of CAD.

## Discussion

To our knowledge, this study is the first to identify the protein mediator MIF-2/D-DT as an atypical chemokine and non-cognate ligand of CXCR4 that promotes atherogenesis and vascular inflammation. MIF-2 not just shares pro-atherogenic properties with MIF, but surprisingly, and contrary to MIF, enhances circulating atherogenic lipids, as well as hepatosteatosis, associated with a promoting effect on liver lipid content (TG, DG, and CE) and lipogenic pathways in hepatocytes. In line with this - dual - vascular and hepatic/lipid phenotype, MIF-2 was found to be markedly upregulated in unstable carotid plaques and in plasma of CAD patients, where higher MIF-2 levels were measured in patients with ACS compared to CCS, correlating with clinical parameters. Thus, MIF-2 is an atherogenic mediator that regulates lipogenesis and vascular inflammation in cardiovascular diseases and may be a risk marker for CAD (Supplementary Fig. 32).

Using mouse models of early and advanced atherosclerosis, we found that global deletion of *Mif-2* in *Apoe*$^{-/-}$ mice protected against HFD-induced atherogenesis and led to a reduction in lesion formation, vascular inflammation, and circulating inflammatory cytokines. Of note, atherosclerotic lesions of *Mif-2*-deficient mice in the more advanced 12-week HFD mouse model also displayed a reduced necrotic core area compared to the *Mif-2*-expressing controls. However, while this was accompanied by a decrease in CD68+ macrophage lesion area and is in accord with the observed promoting effect of recombinant MIF-2 on macrophage foam-cell formation in vitro, plaque collagen content, fibrous cap thickness and α-SMA+ lesion area were unchanged, suggesting that VSMCs are not chiefly involved. This is unexpected, but could be explained as follows: VSMCs can substantially contribute to atheroma foam-cell formation in *Apoe*$^{-/-}$ mice[79], but the transdifferentiation of VSMCs into plaque macrophages or CD68+ macrophage-like SMCs in Western-type HFD-fed *Apoe*$^{-/-}$ mice does typically not occur until 12–16 weeks after start of the HFD, whereas non-SMC-derived macrophages dominate the lesional macrophage pool between 5 and 12 weeks of HFD[56]. This time window was not covered in our study. Moreover, *Mif-2* deficiency might directly or indirectly impede the phenotypic switch of VSMCs into macrophage-like cells. Overall, it appears that the observed decrease in CD68+ macrophages seen in *Mif-2*-deficient *Apoe*$^{-/-}$ mice upon 12-week HFD is thus likely mostly a result of decreased monocyte infiltration and/or macrophage foam-cell formation. Yet, our reanalysis of scRNAseq data sets[57,58] from two different atherogenic mouse models in an advanced disease stage (*Apoe*$^{-/-}$ and *Ldlr*$^{-/-}$ mice after 8, 16, or 26 weeks HFD) revealed a prominent expression of *Mif-2* in all three detected plaque

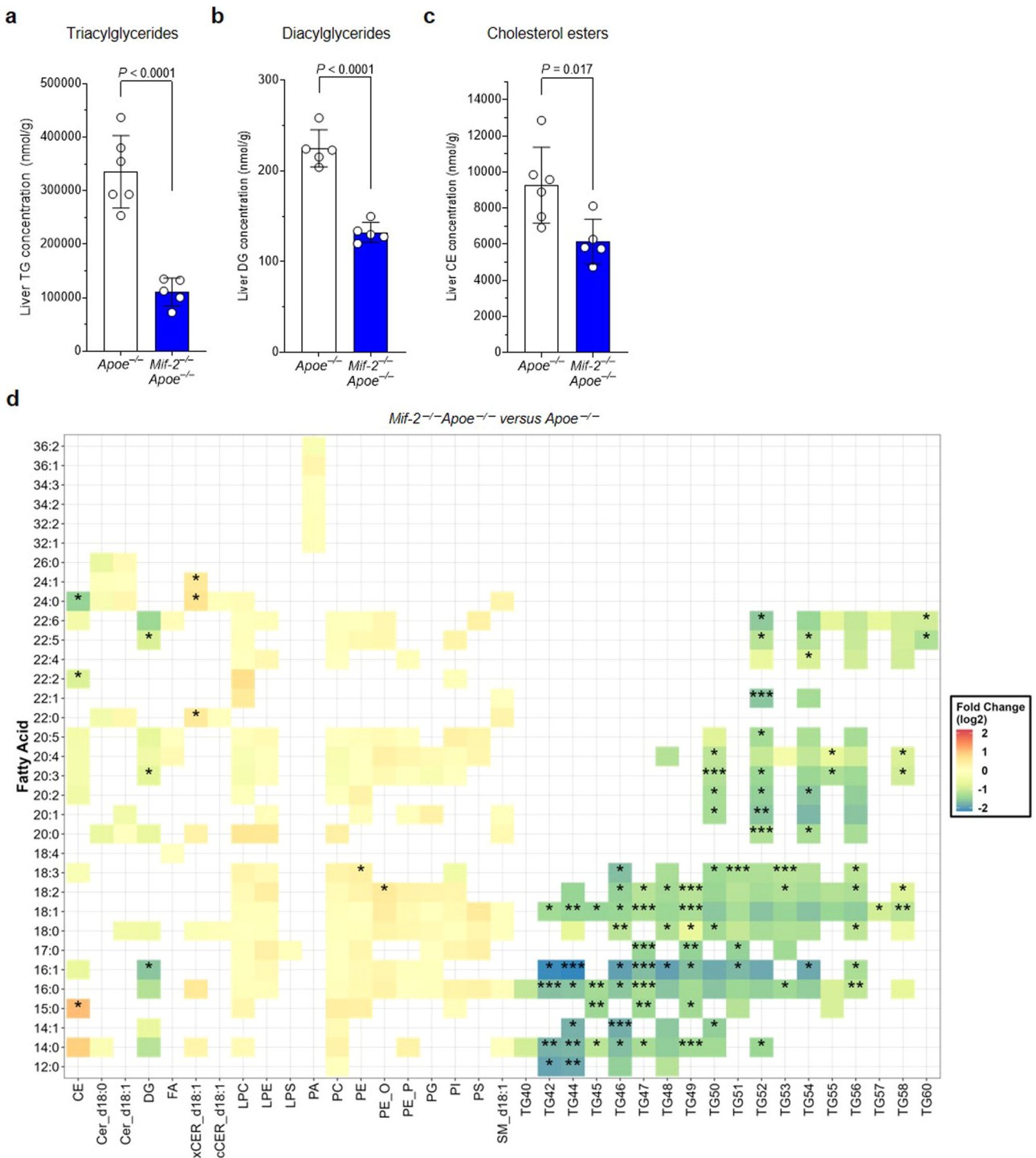

**Fig. 6 | MIF-2 deficiency reduces lipid accumulation and alters lipid composition in the livers of atherogenic *Apoe*⁻/⁻ mice. a–c** Comparison of liver lipid concentrations of triacylglycerides (**a**), diacylglycerides (**b**), and cholesterol esters (**c**) between *Mif-2*⁻/⁻ *Apoe*⁻/⁻ and *Apoe*⁻/⁻ mice after 12 weeks on cholesterol-rich high-fat diet (HFD). Lipid concentrations (nmol/g) were determined using differential mobility separation (DMS)-driven shotgun lipidomics. *Apoe*⁻/⁻ mice (*n* = 6 mice), *Mif-2*⁻/⁻ *Apoe*⁻/⁻ (*n* = 5 mice). **d** Heatmap displaying fold changes of fatty acid concentrations across various lipid classes in *Mif-2*⁻/⁻ *Apoe*⁻/⁻ versus *Apoe*⁻/⁻ mice including triacylglycerides (TG), diacylglycerides (DG), and cholesteryl esters (CE). Colors represent fold change, with green indicating a reduction and red indicating an increase in respective lipid concentration. Data are presented as means ± SEM and were analyzed using an unpaired two-tailed Student's *t*-test. Statistically significant differences are indicated by asterisks based on Benjamini–Hochberg adjusted *P*-values (**d**). HexCER hexosylceramides, Cer d18:1 C18 ceramides (d18:1), Cer d18:0 C18 ceramides (d18:0), FA free fatty acids, LacCER lactosylceramides, LPE lysophosphatidylethanolamines, LPS lysophosphatidylserines, PA phosphatidic acids, PC phosphatidylcholines, PE phosphatidylethanolamines, LPC lysophosphatidylcholines, PG phosphatidylglycerols, PI phosphatidylinositols, PS phosphatidylserines, SM sphingomyelins.

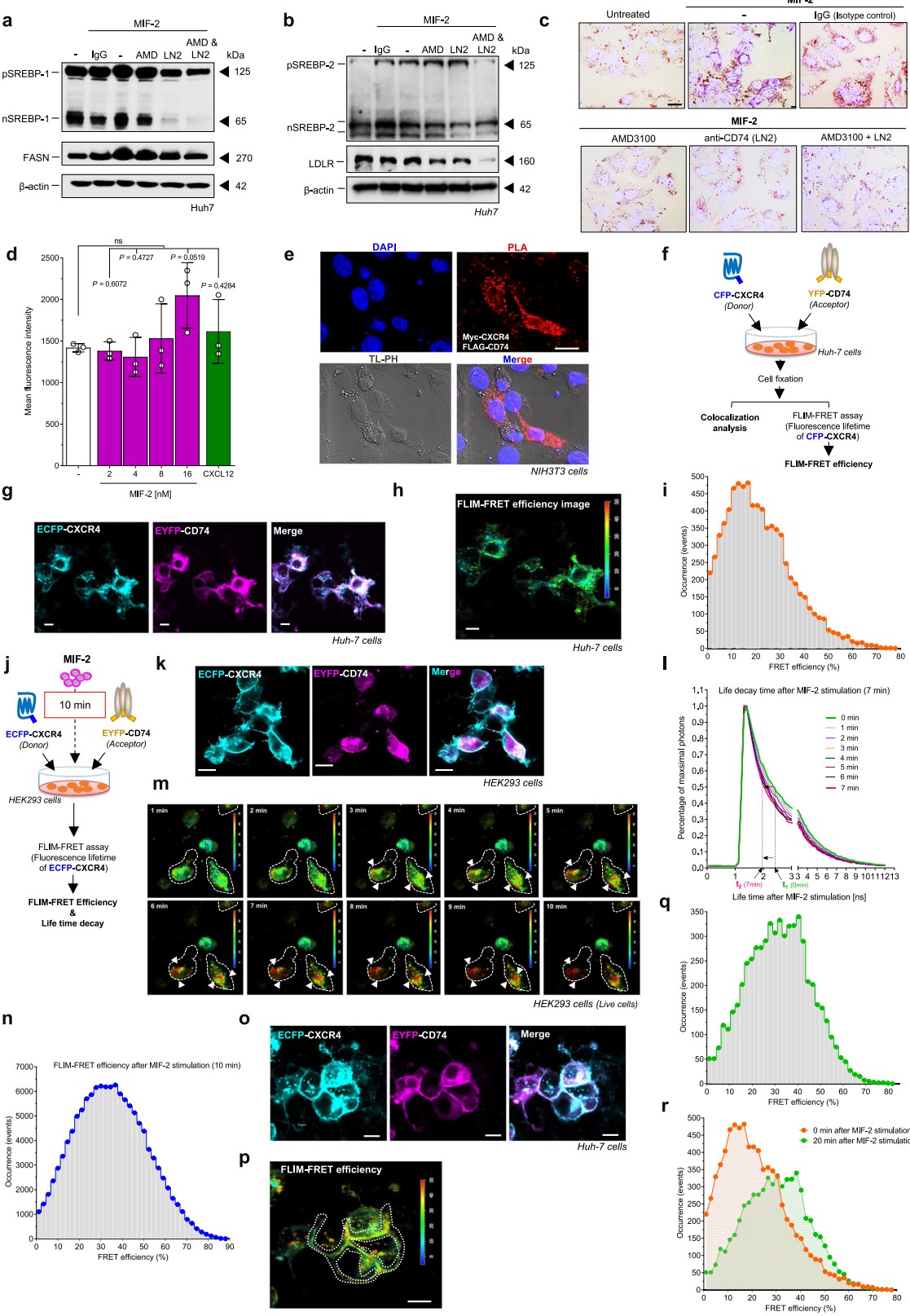

SMC cell populations. While the potential mechanisms of Mif-2 induction and secretion from VSMCs are unknown, this may suggest that VSMCs could be a relevant source of MIF-2 in the atherosclerotic microenvironment, thereby contributing to plaque inflammation and foam-cell formation.

Interestingly, the overall phenotype of *Mif-2* deficiency in atherogenic *Apoe*[−/−] mice differed from that of its homolog MIF. We

previously showed that hyperlipidemic *Mif*[−/−]*Apoe*[−/−] mice displayed a regio-specific phenotype, showing plaque size reduction in brachiocephalic artery and abdominal aorta compared with *Apoe*[−/−] mice, but not aortic root, arch, and thoracic aorta[49,80]. In contrast, *Mif-2*[−/−]*Apoe*[−/−] mice had a marked decrease of lesions in the aortic root and arch. The protective effect of *Mif-2* gene deletion was confirmed in a pharmacological model, applying the MIF-2-selective small molecule inhibitor

**Fig. 7 | MIF-2 triggers CD74/CXCR4 complex formation in HEK293 cells and hepatocytes. a, b** Role CD74 and CXCR4 in MIF-2-elicited SREBP activation in Huh-7 hepatocytes. Cells were stimulated with 16 nM MIF-2 in the presence of AMD3100 (AMD), CD74-blocking antibody (LN2), or both; control: isotype IgG. Representative Western blots for SREBP−1 and FASN (**a**), and SREBP-2 and LDLR (**b**) from two independent experiments. **c** Effect of MIF-2 on LDL uptake in Huh-7 hepatocytes and blockade by CXCR4 and CD74 inhibitors. ORO staining to visualize lipids (scale bar: 25 μm). Images representative of two separate experiments. **d** MIF-2 does not affect CXCR4 internalization without CD74. Splenic B cells from *Cd74*[−/−] mice were stimulated with MIF-2 or CXCL12 for 20 min, and CXCR4 expression monitored by flow cytometry. Statistical analysis: unpaired two-tailed Student's *t*-test; ns, not significant. **e** Proximity ligation assay (PLA) showing CXCR4/CD74 interaction in NIH/3T3 cells. Transfected cells with CXCR4-Myc and FLAG-CD74 plasmids were probed with anti-Myc and anti-FLAG antibodies. PLA signals (red); nuclei stained with DAPI (blue) (scale bar: 20 μm). Images representative of two separate experiments. **f–i** FLIM-FRET microscopy analysis of CXCR4/CD74 interaction in Huh-7 cells. **f** Schematic of the experiment. **g** Two/multi-photon microscopy of ECFP-CXCR4 (cyan) and EYFP-CD74 (magenta) colocalization in fixed Huh-7 cells. **h, i** FLIM analysis of FRET efficiency between ECFP-CXCR4 and EYFP-CD74 (FLIM image (**h**) and histogram (**i**)). Scale bar: 10 μm. **j–n** MIF-2 promotes CXCR4/CD74 complex formation in HEK293 cells. **j** Experimental schematic. **k** Co-colocalization of ECFP-CXCR4 (cyan) and EYFP-CD74 (magenta) in unstimulated, live HEK293 cells. Scale bar: 10 μm. **l** Normalized fluorescence-lifetime decay curves of ECFP-CXCR4 over time (decrease from $-t_1 = 2.47$ ns to $-t_2 = 2.00$ ns). **n** FRET efficiency images at one-minute intervals (**m**); Scale bar: 10 μm. Histogram of FRET efficiency after 10 min of MIF-2 stimulation. **o–r** MIF-2 promotes CXCR4/CD74 complex formation in Huh-7 cells. Experimental scheme as in (**j**), but Huh-7 cells were fixed after 20 min of treatment. **o** Co-colocalization of ECFP-CXCR4 (cyan) and EYFP-CD74 (magenta) in unstimulated Huh-7 cells. Scale bar: 10 μm. **p** FRET efficiency image, Scale bar: 10 μm, **q** histogram, and comparison between FLIM-FRET efficiencies of treated and untreated cells (**r**).

---

4-CPPC[31,48]. This finding also ruled out any gene compensation effect that may be seen in global gene knockout models. Moreover, the atherogenic effects of MIF-2 were underpinned by various in vitro studies. Similar to MIF[20,54], MIF-2 not only promoted macrophage foam-cell formation, but also dose-dependently enhanced the chemotactic migration and endothelial arrest of monocytes as well as B-cell recruitment. Direct comparison with MIF even suggested that MIF-2 was the more potent chemokine.

The cytokine array data provided insight into how MIF-2 may skew the systemic inflammatory response in atherogenic mice. Cytokines/chemokines significantly downregulated in *Mif-2*[−/−]*Apoe*[−/−] mice were predominantly associated with T-cell activation, including IFN-γ, IL-1α, IL-2, IL-16, IL-17, and CXCL13[81], suggesting that MIF-2 may be involved in the activation and recruitment of CD4+ and/or Th17 T cells. How this relates to the observed plaque phenotype will be subject of future scrutiny. Interestingly, when comparing the overall cytokine/chemokine profiles between genetic *Mif-2*-deficient *Apoe*[−/−] mice with *Apoe*[−/−] mice, in which Mif-2 was blocked pharmacologically by 4-CPPC, a number of similarities were noted. Cytokines/chemokines such as IL-1α, IFN-γ, IL-5, CXCL5, IL-15, TNF-α, IL-4, IL-23, IL-1β, CCL5, CCL2, and IL-27 were found to be reduced in a comparable manner.

Similar to MIF, MIF-2 has been described as a high-affinity ligand for CD74[29], which requires a co-receptor such as CD44[29]. However, Ishimoto et al. also reported that the MIF-2/CD74 axis can drive the expression of IL-6 independently of CD44 in preadipocytes[82], suggesting a role for alternative pathways. Interestingly, CD74 was found to form heteromeric complexes with the MIF chemokine receptor CXCR4[15,21,22], while Tilstam et al. demonstrated that MIF but not MIF-2 recruits inflammatory macrophages in a polymicrobial sepsis model via CXCR2[35]. The MIF/CXCR4 axis is relevant in atherosclerosis, as it has been associated with atherogenic activities of not only monocytes and T cells, but also neutrophils, platelets, and B cells[15,20,83,84]. Applying various biochemical and immunological methods, we here identify CXCR4 as a receptor for MIF-2 and provide evidence that the MIF-2/CXCR4 axis is critical in conveying atherogenic activities of MIF-2. We noted the presence of conserved motifs that may be important for the interaction between MIF or MIF-2 and CXCR4. In contrast, MIF-2 lacks the pseudo(*E*)LR motif shown to contribute to the MIF/CXCR2 interaction[35]. Together, our receptor internalization, yeast-CXCR4 transformant, and titration experiments, provided evidence that MIF-2 interacts with CXCR4 with nanomolar affinity to elicit responses in model cells and leukocytes. These findings were supported by studies applying pharmacological blockade of CXCR4. Binding of MIF-2 to CXCR4 is further supported by HADDOCK docking simulation and experimental binding to the CXCR4 surrogate peptide msR4M-L1[54].

MIF-2 is highly expressed in liver[29,85], but its role in this organ has not been explored. We show that *Mif-2*[−/−]*Apoe*[−/−] mice on HFD exhibit a pronounced liver phenotype compared to Mif-2-proficient *Apoe*[−/−] mice that is associated with a decrease in plasma cholesterol, TG, and VLDL levels, a substantial decrease in liver TG, DG, and CE levels, a less inflammatory and less lipogenic hepatic transcriptomic signature, and finally reduced hepatosteatosis. The evidence comes from conventional plasma lipid analytics, hepatic lipid stainings, an in vitro steatosis assay, an in vitro triglyceride synthesis assay, comprehensive lipidomics of liver tissues, as well as two independent bulk RNAseq analyses from both OCT-embedded frozen liver sections and liver lysates.

Since the biosynthesis of cholesterol, FA, and TG is regulated by SREBPs[46], we studied the role of MIF-2 in SREBP activation in hepatocytes. Previous studies reported that genetic depletion of *Srebp-1* in vivo was associated with a reduction in hepatic fatty acid production, whereas overexpression of *Srebp-1* or *Srebp-2* in mice promoted steatosis with an increase in plasma lipids[86,87]. Previous data suggesting a link between hepatic SREBP proteins and both metabolic disease and atherosclerosis[59,60] further added to our hypothesis that SREBPs could play a role in the observed liver lipid phenotype in *Mif-2*[−/−]*Apoe*[−/−] mice. The activation of SREBPs is initiated by proteolytic processing that is regulated by various factors including sterols. Indeed, we here provide evidence that MIF-2 acts as a regulator of SREBP activation in hepatocytes. Our results in the hepatocyte cell line Huh-7 indicate that MIF-2 dose-dependently promotes the activation of SREBP-1/-2, the expression of their lipogenic target genes *FASN* and *ACC*, along with an enhancement of lipogenesis and lipid esterification. *FASN* and *ACC* encode fatty acid synthase (FAS) and acetyl-CoA carboxylase (ACC), the two key enzymes controlling fatty acid biosynthesis. ACC catalyzes the irreversible carboxylation of acetyl-CoA to produce malonyl-CoA, thereby providing malonyl-CoA substrate for the biosynthesis of fatty acids. FAS is a multi-enzyme complex, whose main function is to catalyze the synthesis of the C16 fatty acid palmitate. Regulation of SREBPs, FAS, and ACC by MIF-2 in hepatocytes thus suggests that MIF-2 has a direct stimulatory effect on hepatic lipid synthesis, in line with the results from the triglyceride synthesis assay performed on Huh-7. This notion is supported by comparing nuclear SREBP levels in liver lysates from *Apoe*[−/−] *versus* *Mif-2*[−/−]*Apoe*[−/−] mice, and RT-qPCR results revealing that in the livers of *Mif-2*[−/−]*Apoe*[−/−] mice, *Srebp-2* and *Fasn* mRNA levels were significantly downregulated, while *Acc* expression was also reduced, though the decrease was not statistically significant.

Of note, a role for MIF-2 in controlling SREBP-related hepatic pathways and, more generally, in driving hepatic lipogenic pathways was confirmed by unbiased transcriptomic analysis from both microscopic liver sections and whole liver tissue lysates, comparing the hepatic transcriptomes from *Apoe*[−/−] *versus* *Mif-2*[−/−]*Apoe*[−/−] mice. The identified DEGs encode for lipid and phospholipid synthesis-regulating enzymes such as *Acat2/Acaa2*, *Acot1*, *Acyl-CoA synthase family member 3* (*Acsf3*), *Hsd17b4*, *Hsd17b13*, *Agpat2*, *Agpat3*, *Ces1*, *Ces2c*, *Ces2e*, *Aldh3a2*, and many others. Strikingly, many cytochrome P450 genes including *Cyp4a12a*, *Cyp2a12*, *Cyp4a10*, *Cyp2a4*, *Cyp2d9*,

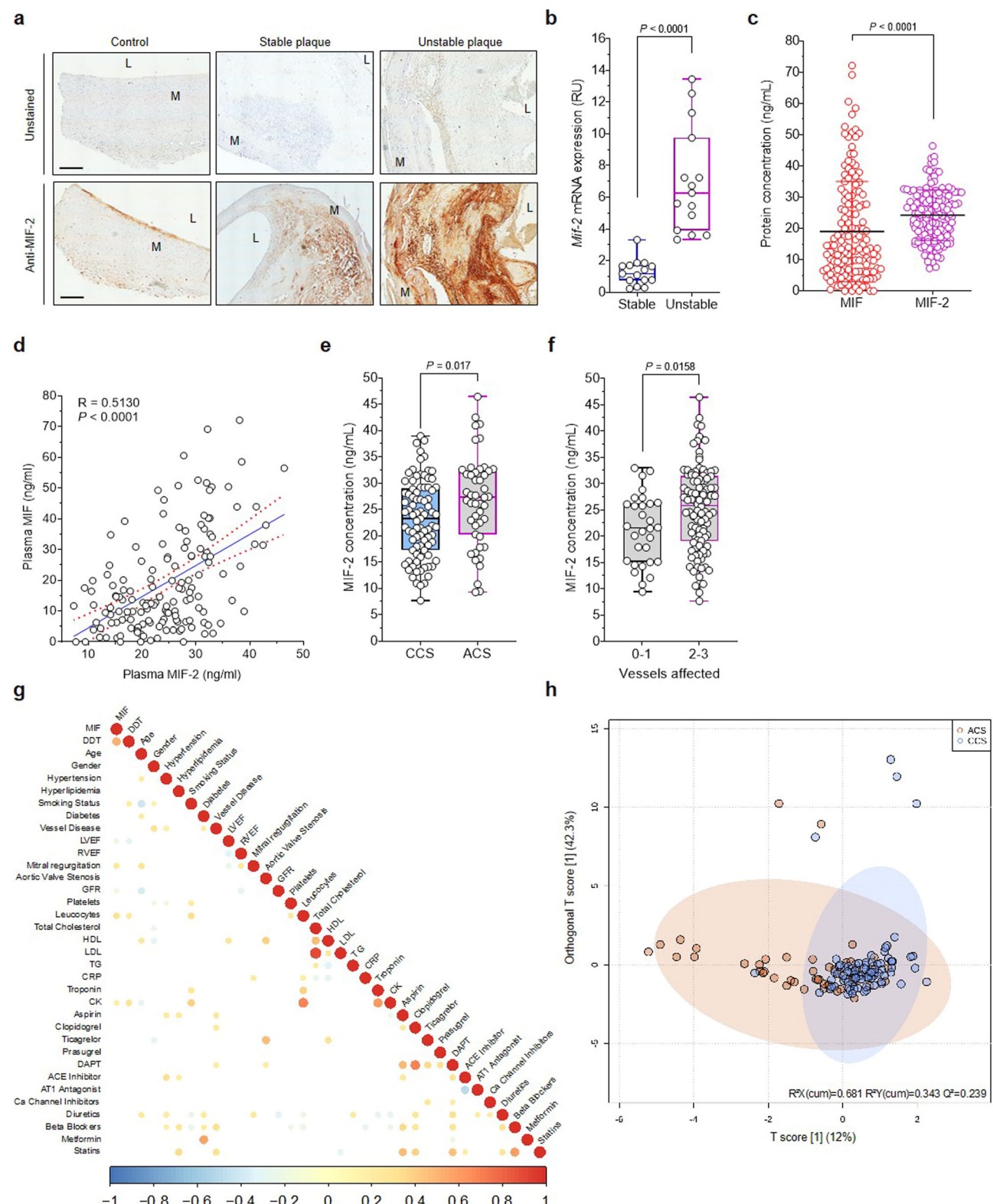

important – among other functions - for the omega-1 oxidation of fatty acids and monooxygenase activity, were strongly downregulated in the *Mif-2*-deficient livers. Moreover, genes related to lipid binding, uptake, and lipoprotein formation such as *Apoa4, Fatty acid-binding protein 2 (Fabp2)*, or *Lipa* were enhanced in the *Mif-2*-expressing livers. In accord, pathway analysis strongly predicted an enrichment of the terms 'fatty acid metabolism' and 'regulation of lipid metabolic processes'. The significance of the *Mif-2*-dependent enrichment of

pathways like 'purine nucleotide metabolic processing' is less clear at first sight, but purines are also known to serve as cofactors such as NADH and coenzyme A, which likely connects this pathway to Mif-2-driven fatty acid metabolism. Surprisingly, we noted a significant increase in the expression of *Pcsk9* in the livers of *Mif-2^(−/−)Apoe^(−/−)* mice. This upregulation corresponds with the elevated transcript levels of *Pcsk9* identified in the *Mif-2^(−/−)Apoe^(−/−)* mice through RNAseq data. PCSK9 is known to play a critical role in cholesterol uptake and

**Fig. 8 | Upregulation of MIF-2 expression in unstable human carotid plaques and correlation with acute CAD. a, b** MIF-2 expression in plaques from patients undergoing CEA. **a** MIF-2 immunopositivity in stable and unstable plaques and comparison to healthy vessels. Representative DAB staining from FFPE plaque sections developed with an anti-MIF-2 antibody (right). Unstained (control without primary antibody, left). Images are representative of 4 vessels per group. Scale bar: 500 μm. **b** MIF-2 mRNA expression in stable *versus* unstable plaques measured by RT-qPCR. *n* = 15 patients per group. Values are means ± SD and were analyzed using an unpaired two-tailed Student's *t*-test. **c–h** MIF-2 levels in plasma of CAD patients and correlations with clinical parameters. **c** Comparison of MIF-2 and MIF concentrations determined by ELISA (*n* = 149 patients). Statistics: unpaired two-tailed Student's *t*-test. **d** Correlation between MIF-2 and MIF (Spearman's rank correlation

*R* = 0.5130, *P* < 0.0001). **e** MIF-2 plasma levels in sub-groups of patients with chronic coronary syndrome (CCS; *n* = 85) and acute coronary syndrome (ACS; *n* = 47). Statistics: unpaired two-tailed Student's *t*-test and Mann–Whitney test. **f** MIF-2 levels according to vessel disease score (scale 0–3). Statistics: unpaired two-tailed Student's *t*-test and Mann–Whitney test. **g** Correlation analysis of MIF and MIF-2 plasma levels with baseline clinical parameters, including cardiovascular risk factors, laboratory parameters, and medication. Significant Pearson coefficients (*rp*) (*P* < 0.05) highlighted and colored accordingly. **h** Orthogonal partial least squares discriminant analysis (OPLS-DA) score plots between ACS and CCS patients including MIF, MIF-2, and clinical/laboratory parameters ($R^2X$ = 0.681; $R^2Y$ = 0.343; $Q^2$ = 0.239). Dots represent individual patients colored according to disease severity (ACS = orange; CCS = blue).

metabolism in the liver by modulating the LDL receptor, which leads to increased circulating cholesterol levels. However, our study demonstrated that *Mif-2$^{-/-}$Apoe$^{-/-}$* mice exhibited lower plasma cholesterol levels. On the other hand, the combination of RNAseq data and RT-qPCR results revealed a significant decrease in the expression of key enzymes and factors involved in cholesterol biosynthesis and lipid uptake including *Ppar-γ* (*Pparg*), *Cd36*, *Sr-b1*, *Hmgcs1*, *Hmgcr*, and *Acat-1*. Therefore, in addition to the role of PCSK9 in regulating circulating cholesterol, the downregulation of these crucial genes in the liver likely contributes significantly to the observed decrease in circulating cholesterol levels. The unexpected increase of PCSK9 expression may suggest the presence of a compensatory mechanism.

Of note, the *Mif-2*-expressing liver tissue also showed an enhancement of inflammatory genes such as *Serpina12*, *Serpina3k*, *Tumor necrosis factor receptor 1 (Tnfrsf1a)*, *Complement factor 3 (C3)*, or *S100a10*, including the MIF/MIF-2 receptor *Cd74*, altogether suggesting an enrichment of lipid synthesis and inflammation transcripts in *Mif-2*-expressing compared to *Mif-2*-deficient liver.

Importantly, a comprehensive lipidomic phenotyping of liver tissue from both genotypes confirmed the predicted role of MIF-2 in promoting hepatic lipid accumulation. Liver tissue from *Apoe$^{-/-}$* mice contained markedly elevated levels of TG with different fatty acid chain length and degree of desaturation, as well as enhanced DG and CE levels compared to liver tissue from *Mif-2$^{-/-}$Apoe$^{-/-}$* mice. This further underscored a striking lipogenic role of MIF-2 in the liver and is in line with the liver tissue stainings, the transcriptomic profiles, the in vitro steatosis assay and SREBP activity measurements in hepatocytes, as well as the plasma lipid analysis. The observation that one class of lipids – hexosylceramides (HexCer) – was found to be elevated in *Mif-2*-deficient liver is interesting, but its significance and relationship to the athero- and lipoprotective phenotype upon *Mif-2* deficiency is currently unclear. However, plasma levels of circulating HexCer and their ceramide precursors have been shown to be associated with CVD/CAD in several human trials[88–90].

Some of the identified genes in the RNAseq analyses have been correlated with upstream pathways such as AMPK or PI3K/AKT. AMPK negatively regulates the transcriptional activity of SREBPs to mitigate hepatosteatosis and atherogenesis[59,64] while PI3K/AKT activates SREBPs to promote lipogenesis[65,66]. In accord, we observed that MIF-2 increased AKT activation in Huh-7, while it led to an attenuation of AMPK phosphorylation. Thus, MIF-2 could be a regulator of hepatic lipid homeostasis that affects SREBP activity through activating PI3K/AKT signaling while suppressing the AMPK pathway. This notion is in line with previous studies suggesting a link between MIF-2 and AMPK signaling in adipocytes[40].

The cleavage and subsequent translocation of SREBPs is mediated by the well-conserved SCAP-S1P-S2P axis[47,91], but the upstream signaling components involved in the activation of this axis remain unexplored. One important observation in our study is that the blockade of either CXCR4 or CD74 abolished MIF-2-mediated SREBP activation, suggesting that these receptors and their ligand MIF-2 are upstream of the proteolytic process. We speculate that CXCR4 and CD74 either

operate via receptor complex formation or signaling crosstalk. Our PLA and FLIM-FRET imaging data suggest close proximity and a direct interaction between CXCR4 and CD74. Moreover, FLIM-FRET showed that MIF-2 stimulation promoted the formation of the CXCR4/CD74 receptor complex in HEK293 cells and hepatocytes. Therefore, and as MIF-2 is highly expressed in liver, it seems reasonable to speculate that signaling elicited by MIF-2/CXCR4/CD74 activates SREBPs in hepatocytes. In line with this notion are earlier findings from Kim et al. in cancer cells, demonstrating that the CXCL12/CXCR4 axis is involved in SREBP-1-mediated FASN expression by enhancing the nuclear translocation of SREBP-1[92].

The roles of MIF and MIF-2 in hepatic steatosis and liver lipid metabolism appear to be oppositional. Previous data indicated that MIF has a hepatoprotective effect in steatosis and fibrosis by promoting the CD74/AMPK pathway in hepatocytes and *Mif$^{-/-}$* mice on a non-Western-type HFD showed enhanced lipid accumulation in the liver, which was associated with upregulated *Srebp-1* and *Fasn* levels[24,93]. In contrast, while the specific HFD was different (triglyceride-rich *versus* Western), our results suggest that MIF-2 promotes hepatic steatosis.

Since our data pointed to a role of MIF-2 in advanced atherogenesis, we analyzed MIF-2 expression levels in human atherosclerotic plaque tissue, comparing stable and unstable plaques from CEA patients, which typically suffer from metabolic co-morbidities. MIF-2 was not only elevated in plaque compared to healthy vessel, its expression was upregulated in unstable compared to stable plaques, a finding that was confirmed, when we reanalyzed publicly available bulk and single-cell RNAseq datasets from carotid plaques of atherosclerotic patients[75,76] for MIF-2 and MIF expression. This observation is interesting, as MIF expression levels, while also generally upregulated in CEA, do not differ between stable and unstable plaque tissue[54]. Thus, the two homologs may have differential roles in different stages of atheroprogression.

To further assess the significance of our findings for ischemic heart disease, we quantified the plasma concentrations of MIF-2 in a large cohort of CAD patients and also compared ACS and CCS subgroups as an indicator of disease severity. MIF-2 levels in CAD were elevated compared to concentrations known in healthy individuals. This is in accord with a previous study, in which serum MIF-2 was found to be increased in the reperfusion phase of cardiac surgery patients compared to its pre-operative concentrations[78]. Of note, MIF-2 levels were significantly higher in patients with ACS compared to CCS. While elevated MIF plasma levels have previously been noted in ACS patients[13,94], we found that MIF-2 concentrations were even higher than those of MIF. Both MIF and MIF-2 showed correlations with CAD risk parameters such as LVEF, CK, leukocytes and CRP. To our knowledge, this is the first study to examine MIF-2 levels in CAD patients[45]. Together, MIF-2 levels may be associated with high-risk atherosclerotic plaques in CAD patients[95].

In summary, our study identifies MIF-2/D-DT as a pro-atherogenic atypical chemokine and CXCR4 as a receptor. Unlike MIF, MIF-2 contributes to lipid accumulation in the liver and elevated MIF-2 levels correlate with unstable CEA and symptomatic coronary plaques,

pointing toward a role for MIF-2 in advanced atherosclerotic diseases. This work also gives insight into the underlying mechanism of how MIF-2 promotes lipogenesis in hepatocytes, identifying the MIF-2/CD74/CXCR4 axis as an upstream pathway triggering SREBPs. While the causal impact of the potential cellular sources of MIF-2 expression (immune cells, VSMCs, hepatocytes) is currently unclear and should be an important aim of future studies, our data overall imply that MIF-2 may be a promising therapeutic target in atherosclerosis, vascular inflammation, and associated metabolic conditions. Targeting could entail joint targeting together with MIF, as both MIF and MIF-2 are clear-cut pro-atherogenic mediators that exacerbate vascular inflammation and lesion formation. Both monoclonal antibodies and dual-targeting small molecule inhibitors such as ISO-1 or 4-IPP are available to jointly target MIF and MIF-2 and some of these agents are being developed further for clinical applications (e.g. reviewed in refs. [11,96]). MIF-2-selective strategies could be of interest, when an emphasis is laid on combined hepatic conditions. In fact, recent work has discovered promising MIF-2-selective inhibitors. In addition to MIF-2-selective mAbs[29,35], three small molecule compounds with high selectivity for MIF-2 have recently been developed. 4-CPPC[48,97] shows 13-fold selectivity over MIF, was used in our study and its administration in a model of early atherogenesis in vivo recapitulated the phenotype seen in *Mif-2*-gene-deficient atherogenic mice. Two other compounds, thieno[2,3-d]pyrimidine-2,4(1H,3H)-dione and 2,5-pyridinedicarboxylic acid exhibit even higher selectivities for MIF-2 over MIF, but so far have only been tested in enzymatic tautomerase and cell culture assays[98,99]. Another option to target MIF-2 could be the CXCR4 surrogate peptide msR4M-L1, applied in this study in our biochemical experiments. This peptide was originally designed to block the MIF/CXCR4 axis and binds MIF with high nanomolar affinity, but, as shown here, also binds to MIF-2 with six-fold lower affinity. msR4M-L1 would therefore, in part, represent a dual anti-MIF/MIF-2 agent, but might be amenable to structure-activity optimization to generate a MIF-2-selective variant. Along the same lines, the peptide construct MHCII DRα1-MOG-35-55 is a promising pathway-specific blocker, as it inhibits MIF and MIF-2 signaling through CD74 as well as T-cell activation[100]. Thus, potential therapeutic concepts for MIF-2-based targeting are already available in the field.

Overall, the study contributes to our understanding of how vascular inflammation and adverse hepatic phenotypes are linked in atherosclerotic diseases and identifies pathways for targeting.

## Methods

### Atherosclerotic mouse models and treatment

All mouse experiments were approved by the Animal Care and Use Committee of the local authorities and were performed in accordance with the animal welfare officer of the Center for Stroke and Dementia Research (CSD). *Apoe*−/− mice with C57BL/6 background were originally obtained from Charles River Laboratories (Sulzfeld, Germany) and backcrossed before use in the CSD animal facility. Although statistical methods were not used to predetermine sample size, G-power analysis was applied in this study to validate sufficient numbers of mice in all cohorts. To generate *Mif-2*−/−*Apoe*−/− mice, *Mif-2*+/− and atherogenesis-prone *Apoe*−/− mice were crossed. Genetic mouse experiments were conducted on eight-week-old female and male *Mif-2*−/−*Apoe*−/− mice and *Apoe*−/− mice. For pharmacological blockade of MIF-2 in vivo, *Apoe*−/− mice were administered 4-CPPC (5 mg kg−1). For this purpose, eight-week-old male *Apoe*−/− mice were randomly divided into two groups of 11 mice each. The experimental group was administered with 100 μg of 4-CPPC dissolved in physiological saline (0.9% NaCl) intraperitoneally every other day for 4.5 weeks; while the control group received saline at the same time intervals. The 4-CPPC injection showed no toxicity or other side effects in the mice.

All mice were housed under a 12 h light/dark cycle and had *ad libitum* access to food and water. At the age of 8 weeks, mice were fed a Western-type high-fat diet (HFD) containing 0.21% cholesterol (ssniff Spezialdiäten GmbH, Soest, Germany) for 4.5 or 12 weeks. Early to moderate atherosclerotic lesions could be developed in these mouse models[101]. For the prospective end point of the proposed experiment, mice were subjected to isoflurane and midazolam/medetomidine/fentanyl (MMF) anesthesia, body weight was measured, and blood was collected by cardiac puncture for routine immune cell counts and lipid analysis. Mice were then perfused transcardially with saline, hearts and proximal aortas were dissected and fixed for quantitative plaque analyses and vessel morphometry. The other organs such as spleens, livers, and adipose tissues were stored at −80 °C.

### Plasmids

The cloning of pECFP-N1-CXCR4 and pEYFP-C1-CD74minRTS was achieved by inserting cDNA encoding human CXCR4 and human CD74minRTS into the pECFP-N1 and pEYFP-C1 vectors, respectively[21]. To generate C-terminal Myc-tagged CXCR4 (CXCR4-Myc) and N-terminal FLAG-tagged CD74minRTS (FLAG-CD74minRTS), cDNA encoding CXCR4 or CD74minRTS was amplified from pECFP-N1-CXCR4 and pEYFP-C1-CD74minRTS using primers: 5′-GCCTCGAGGC-CACCATGTACCCATACGAT-3′ and 5′-GCAAGCT TCCGCTGGAGTGAA AACTTGAAG-3′ for CXCR4; 5′-GCGGATCCATGATGACCAGCGCGAC-3′ and 5′-AGCGAATTCCGTCACATGGACTGGCC-3′ for CD74, and then inserted into pcDNA3.1-Myc-HisB and pcDNA3.1-FLAG plasmids kindly provided by Bernard Lüscher (RWTH Aachen University, Germany) and Teruko Tamura-Niemann (Hannover Medical School, Germany).

### Expression and purification of MIF and MIF-2 proteins

MIF was expressed in *E. coli* BL21/DE3-pET11b and purified essentially as described previously[54,102]. MIF-2 was cloned into a pET22b expression vector, expressed in *E. coli* BL21-CodonPlus, and purified following a slight modification of a previously reported procedure[29]. Briefly, bacterial extracts were filtered, purified by Q Sepharose chromatography (Cytiva) applying fast protein liquid chromatography (FPLC, Cytiva Europe GmbH, Freiburg, Germany) and a subsequent high-performance liquid chromatography step using C18 reverse-phase separation (RP-HPLC). MIF-2 was refolded using the protocol established for MIF renaturation[29]. SDS-PAGE in combination with Coomassie or silver staining was utilized to evaluate protein purity and mass spectroscopy applied to verify protein identity. The resulting proteins contained <10 pg LPS/μg protein, as quantified by the PyroGene Recombinant Factor C assay (Cambrex, East Rutherford, NJ, USA).

### Cell culture, transfection, and treatment

MonoMac6 cells (a human monocytic cell line derived from a patient with relapsed acute monocytic leukemia[103]) were cultured in RPMI 1640 GlutaMAX medium (GIBCO, Karlsruhe, Germany) supplemented with 10% fetal calf serum (FCS) (GIBCO), 1% non-essential amino acids (NEAA), 1 mM sodium pyruvate and 1% penicillin/streptomycin (P/S) (GIBCO). Human aortic endothelial cells (HAoECs; initially isolated from healthy human aorta) were acquired from PromoCell GmbH. After thawing, cells were cultured in endothelial cell growth medium (ECGM) from the same company. Cells were plated on collagen-coated (Biochrom AG, Berlin, Germany) cell culture plates/dishes. The mouse fibroblast cells line NIH/3T3, the human hepatocellular carcinoma cell line Huh-7, and human embryonic kidney cells (HEK293) were initially purchased from the German Society for Microorganisms and Cell Cultures (DSMZ), sub-passaged, and cultured in DMEM-GlutaMAX (GIBCO) containing 4.5 g/L D-glucose, 1 mM sodium pyruvate, 10% FCS and 1% P/S. All cells were maintained at 37 °C in a humidified $CO_2$ incubator. For cell subculture, cells were divided 2 to 3 times per week at a ratio of 1:3−1:5. HAoECs were used at passage 5−8, passages for the cell lines HEK293, Huh-7, and NIH/3T3 were below 25.

## Primary cells

**Isolation of human peripheral blood mononuclear cells (PBMCs) and monocytes.** The isolation of PBMCs and monocytes was carried out in accordance with the principles outlined in the Declaration of Helsinki principles and was approved by the Ethics Committee of LMU Munich (ethics approval numbers 18-104 and 23-0639). These approvals cover the use of anonymized tissue and blood samples for research purposes. Written informed consent was obtained from all donors. For purification and differentiation of primary human monocytes, PBMCs were isolated following a routine protocol[15]. In brief, peripheral blood was obtained from healthy donors and carefully mixed with prewarmed phosphate buffered saline (PBS) (Invitrogen) at a ratio of 1:1. The layer of PBMCs was separated by density gradient centrifugation using Ficoll-Paque Plus density gradient media (GE Healthcare, Freiburg, Germany). After centrifugation, the cells in the interphase were carefully aspirated and subsequently washed with PBS. Red blood cells (RBC) were removed by adding RBC lysis buffer for 3 min at RT followed by washing with an excess of RMPI 1640 supplemented with 10% FCS. Isolated PBMCs were maintained in the same medium at 37 °C in a humidified atmosphere of 5% $CO_2$. Primary human monocytes were then purified by negative depletion using the human Pan Monocyte Isolation Kit (Miltenyi Biotec, Bergisch Gladbach, Germany) following the manufacturer's protocol. The purity of isolated monocytes was analyzed by flow cytometry using anti-CD14 antibody (Miltenyi Biotec). The purified monocytes were used for the Transwell migration and 3D chemotaxis assays as described below.

Differentiation of isolated monocytes into macrophages was performed in the presence of macrophage colony-stimulating factor (M-CSF). Monocytes were resuspended in RPMI 1640 GlutaMAx medium containing 10% FCS and 1% P/S, and treated with 100 ng/mL M-CSF (PeproTech, Hamburg, Germany) every other day for 6 days. After differentiation, the medium was replaced with fresh medium, and PBMC-derived macrophages were subjected to DiI-LDL uptake assay.

**Isolation of splenic B lymphocytes.** *I*solation of primary B cells from spleens of WT mice was performed using the protocol from Klasen et al.[20]. In brief, spleens were harvested and immediately placed on ice-cold PBS. Cell suspension was performed in a 40-μm cell strainer. After centrifugation, the cell pellet was washed and resuspended in 3 mL RBC buffer (Pierce) at RT for 3 min. RBC cell lysis was stopped by adding 27 mL of RPMI 1640 medium containing 10% FCS. Cells were washed thoroughly with the same medium, and splenic B lymphocytes were then purified using the mouse Pan B Cell Isolation Kit (Miltenyi Biotec) and LD MACS Separation Column (Miltenyi Biotec) according to the manufacturer's protocol. Purified B cells were resuspended in RPMI 1640 medium containing 10% FCS and 1% P/S and maintained at 37 °C in the incubator.

**Isolation of primary murine hepatocytes.** Isolation of primary hepatocytes from adult mice was performed using the protocol from Charni-Natan et al.[104]. In brief, mice were anesthetized and perfused with $Ca^{2+}$- and $Mg^{2+}$-free HBSS (pH 7.4) supplemented with 0.5 mM EDTA and 25 mM HEPES. Livers were perfused via the portal vein with $Ca^{2+}$- and $Mg^{2+}$-containing HBSS (pH 7.4) containing 25 mM HEPES, and then with the same buffer supplemented with liberase (1 mg/mL) to allow enzymatic digestion. Livers were cut into small pieces and dissociated in plating media (low glucose DMEM, 5% FBS and 1% P/S) and filtered through a 70-μm cell strainer. The cell suspension was centrifuged at $50 \times g$ for 2 min at 4 °C. The cell pellet was resuspended in 10 ml Percoll solution (90% Percoll in PBS) and centrifuged at $200 \times g$ for 10 min at 4 °C. Purified hepatocytes were washed twice and cultured in William's medium E containing 2 mM glutamine on collagen-coated 6-well plates at 37 °C. Hepatocytes were treated with various concentrations of recombinant MIF-2, and cell lysates analyzed by Western blot.

## Tissue preparation and lesion analysis

The heart tissues saved for plaque analysis were embedded in Tissue-Tek optimum cutting temperature (O.C.T.) compound and directly fresh-frozen on dry ice in preparation for sectioning. After the block was trimmed, serial eight-μm thick frozen sections were arranged for Oil-Red O (ORO), hematoxylin and eosin (HE) staining, and subsequent quantification of other plaque components, such as immune cells, collagen and necrotic core. The lipid content in the aortic root was stained with 0.5% ORO solution in propylene glycol (Sigma-Aldrich, Taufkirchen, Germany) at 37 °C for 45 min and nuclei were lightly stained with hematoxylin at RT for 1 min. The lesion area was alternatively stained with hematoxylin (Sigma-Aldrich) for 10 min and then counterstained with eosin (Sigma-Aldrich) for 30 s. The macrophage content in plaques of the aortic root was visualized by a rat anti-CD68 antibody (1:100) in combination with a Cy5-conjugated secondary antibody (1:300). Meanwhile nuclei were stained with DAPI. Previously isolated and trimmed aortic arch was fixed in 1% paraformaldehyde (PFA) overnight and transferred into PBS on the day before dehydration. After immersed completely, samples were embedded in paraffin. Molded blocks can be stored at RT or be used for direct sectioning. Four μm paraffin sections including three main branches were cut and HE-stained for plaque measurement. In addition, collagen and necrotic core were stained in consonance with the manufacturer's procedures of trichrome stain (Masson) kit. Nuclei stains black, cytoplasm and muscle fibers stain red, whereas collagen displays blue coloration. Images were acquired with a Leica DMi8 fluorescence microscope (Leica Microsystems, Wetzlar, Germany), and signals were quantified using computer-assisted image analysis software (ImageJ).

## Inflammatory cytokine array

Plasma cytokine and chemokine profiles from *Mif-2⁻/⁻Apoe⁻/⁻* mice and *Apoe⁻/⁻* mice were mapped by capitalizing on a membrane-based mouse cytokine array panel A (R&D Systems, Minnesota, USA) following the standard instructions. Samples were constituted by placing mouse plasma in a 1:10 dilution of array buffer, mixed with a detection antibody cocktail (1:100), and incubated at RT for 2 h. Then pre-blocked membranes were covered with reconstituted samples at 4 °C overnight. The second day, membranes were rinsed followed by 30 min of incubation with the diluted streptavidin-horseradish peroxidase (HRP) solution (1:2000), and then visualized with chemireagent mix and developed by an Odyssey® Fc imager (LI-COR Biosciences, Bad Homburg, Germany) for 2 min, 10 min and 1 h, respectively. All measurements in this assay were conducted in duplicates. The mean pixel density of each pair of duplicate spots was quantified by ImageJ and is represented as the relative level of the corresponding cytokine.

## Multiplex cytokine and chemokine quantification by Luminex

Plasma concentrations of cytokines and chemokines: ENA-78 (CXCL5), eotaxin (CCL11), G-CSF (CSF-3), GM-CSF, GRO-α (CXCL1), IFN-α, IFN-γ, IL-1α, IL-1β, IL-10, IL-12p70, IL-13, IL-15, IL-17A (CTLA-8), IL-18, IL-2, IL-22, IL-23, IL-27, IL-28, IL-3, IL-31, IL-4, IL-5, IL-6, IL-9, IP-10 (CXCL10), LIF, M-CSF, MCP-1 (CCL2), MCP-3 (CCL7), MIP-1α (CCL3), MIP-1β (CCL4), MIP-2α (CXCL2), RANTES (CCL5), and TNF-α were measured using a multiplex x-MAP magnetic bead-based immunoassay kit (ProcartaPlex Mouse Cytokine and Chemokine Convenience panel 1A 36-Plex, #EPXR360-26092-901, Invitrogen (Karlsruhe, Germany)). The assay was performed according to the manufacturer's instructions. Frozen plasma from different mouse cohorts was thawed, homogenized, and centrifuged at 17,000 rpm for 5 min to remove debris and excess lipid droplets. Samples and manufacturer-supplied cytokine standards (standard mix A #287083-000 and mix 1B #288534-00) were diluted 1:6 and 1:4, respectively, in 1× universal assay buffer (UAB) and run as duplicates in 96-well flat-bottom plates. Analysis was performed on a Luminex 200 system using xPONENT v. 3.1 software (Luminex

Corporation, Austin, USA). Washing steps were conducted using a hand-held magnetic plate washer (Invitrogen). The doublet discrimination (DD) gate was set at 7500-25.000, the sample volume was 50 μL, and at least 50 events per bead were recorded. The concentration of the different analytes was calculated using ProcartaPlex Analysis App software (Invitrogen). Data analysis and standard curve fitting were performed using a five-parameter logistic (5PL) fitting model.

## Gene expression analysis in Huh-7 hepatocytes and mouse heart by RT-qPCR

For all experiments involving gene expression analysis, total RNA from cultured cells or tissues was isolated, and then concentrations of RNA samples were measured by Nanodrop spectrophotometer (Thermo Fisher Scientific, Waltham, MA, USA). The real-time quantitative PCR (RT-qPCR) analysis was carried out by using the 2× SensiMix PLUS SYBR No-ROX Kit (Bioline Meridian Bioscience, Luckenwalde, Germany) in a Rotor-Gene 6000 (Qiagen, Hilden, Germany) and employing specific primers, which were purchased from Eurofins BioPharma Product Testing Munich. Raw data were acquired via the Rotor-Gene 6000 Series 1.7 software (Corbett) and relative mRNA levels were calculated by using the $\Delta\Delta C_t$ method with $\beta$-actin as a housekeeping gene. PCR primers are detailed in Table S2.

For analysis in cultured cells, $4 \times 10^5$ Huh-7 hepatocytes were seeded in the 12-well plate on the day before starting the experiment and then starved with DMEM containing 2% FCS overnight. On the third day, Huh-7 cells were stimulated with different concentrations of purified recombinant MIF-2 (i.e. 0 nM, 4 nM, 8 nM, 16 nM, 32 nM, 48 nM) for 24 h. Afterward, cells were lysed and RNA was isolated using TRIzol™ Reagent (Thermo Fisher Scientific) following the manufacturer's protocol.

For murine tissues, half of the mouse heart was collected and flash-frozen on dry ice and then transferred to a 40 μm cell strainer (Corning, Kaiserslautern, Germany). The tissue was cut into small pieces and ground thoroughly using a pipette tip. Genomic DNA was removed from the lysate and total RNA was purified through washing and elution using the RNA/Protein Purification Plus kit (Norgen Biotek, Thorold, Canada) according to the manufacturer's protocol.

## Signaling pathway experiments and Western blot analysis

For all experiments involving protein analysis using Western blot, protein samples were denatured at 95 °C for 10 min. For immunoblot analysis, 10–20 μL protein/lane (corresponding to around 10–20 μg total protein) was loaded into an SDS-PAGE gel, using gels of different acrylamide percentage according to the molecular weight of target proteins. Electrophoresis was carried out and proteins electrotransferred to a PVDF membrane. After blocking with 5% BSA-TBST, membranes were incubated with primary antibodies diluted in 5% BSA-TBST, as follows: anti-SREBP-1 (1:500), anti-SREBP-2 (1:1000), anti-FASN (1:500), anti-LDLR (1:500), anti-MIF-2 (1:1000), anti-AMPK (1:1000), anti-pAMPK (1:1000). After overnight incubation with primary antibody, the membranes were rinsed in 1× TBST for three times and incubated with goat anti-mouse or rabbit HRP-linked secondary antibody (1:10,000). Following thorough washing, the immunoblot was eventually visualized by an Odyssey® Fc imager.

To study the SREBP signaling pathway in vitro, $1 \times 10^5$ Huh-7 cells were seeded in a 24-well plate on the day before starting the experiment and then starved with DMEM including 2% FCS overnight. On the third day, Huh-7 cells were stimulated with different concentrations of MIF-2 (i.e. 0 nM, 4 nM, 8 nM, 16 nM, 32 nM, 48 nM) in 2% FCS DMEM for 24 h. Moreover, Huh-7 cells were also incubated with 8 nM MIF-2 with or without IgG, AMD3100, and LN2 for 24 h to evaluate the blocking effects of receptors. After stimulation, Huh-7 cells were lysed in 1× lysis buffer with dithiothreitol (DTT) and boiled for immunoblotting analysis. As for the AMPK signaling pathway analysis, Huh-7 cells were

stimulated with different concentrations of MIF-2 in 0.05% FCS DMEM for 30 min and samples processed as above.

## Immunofluorescence staining

$1 \times 10^4$ Huh-7 cells were seeded on coverslips in a 24-well plate and maintained overnight in DMEM medium containing 2% FCS and 1% P/S. The next day, cells were stimulated with rMIF-2 (8 and 16 nM) for 24 h at 37 °C. After stimulation, cells were fixed with 4% PFA for 20 min and then permeabilized with 0.02% TritonX-100 in PBS. After washing with PBS, cells were blocked with 1% BSA and 5% goat serum in PBS and then incubated with mouse anti-SREBP-1 (1:100) or with mouse anti-SREBP-2 (1:300) overnight at 4 °C. Cells were washed three times with PBS and incubated for 1 h at RT with goat anti-mouse Alexa Fluor 488 (1:500). Coverslips were incubated with Vectashield Antifade Mounting medium with DAPI (Vector Laboratories, Burlingame, CA, USA), placed on the slides, fixed with nail polish, and imaged using Leica DMi8 fluorescence microscope.

## Hepatic immunochemistry

ORO and HE staining of OCT/paraffin-embedded mouse liver tissue were performed on 8 μm thick frozen sections and 4 μm paraffin sections. After storage at RT for 30 min, frozen sections were immersed in propylene glycol for 2 min and stained with prewarmed ORO solution for 10 min. After staining, tissues were differentiated in 85% propylene glycol for 1 min and then counterstained with modified Mayer's hematoxylin and mounted with aqueous mounting medium (Carl Roth, Karlsruhe, Germany). Because lipids are dissolved by organic solvents during this sample preparation, ORO staining cannot be performed in paraffin sections. HE staining of liver tissues in paraffin or frozen sections was performed as described above for HE staining of vessels.

## Flow cytometry

Total peripheral blood cells isolated from $Apoe^{-/-}$ or $Mif\text{-}2^{-/-} Apoe^{-/-}$ mice fed with a HFD were analyzed by flow cytometry using a cocktail of fluorescently-labeled antibodies. In brief, $1 \times 10^6$ cells of the isolated peripheral leukocytes were washed three times with ice-cold PBS supplemented with 0.5% bovine serum albumin (BSA) and then incubated with a combination of specific antibodies containing V450-conjugated anti-CD45 (BD Biosciences, Heidelberg, Germany), FITC-conjugated anti-CD3 (Miltenyi Biotec), APC-Cy7-conjugated anti-CD19 (BioLegend, Amsterdam, The Netherlands), PE-Cy7-conjugated anti-CD11b (BioLegend), PE-conjugated anti-CD11c (BioLegend), APC-conjugated anti-Ly6C (BioLegend), and PerCP-conjugated anti-Ly6G (BioLegend) or the appropriate isotype controls (IgG) for 1 h at 4 °C. After incubation, cells were washed thoroughly and analyzed using BD FACSVerse™ (BD Biosciences). The quantification of obtained measurements was performed with FlowJo software (Tree Star Inc., Ashland, OR, USA).

Cell surface expression of CXCR4 and CD74 on Huh-7 cells was similarly performed using APC-conjugated anti-CXCR4 and FITC-conjugated anti-CD74 or the corresponding APC- or FITC-conjugated IgG isotype controls.

## Plasma lipid and lipoprotein analysis

Plasma concentrations of total triglycerides and total cholesterol in $Mif\text{-}2^{-/-}Apoe^{-/-}$ mice and $Apoe^{-/-}$ mice fed HFD for 4.5 or 12 weeks were determined using an enzymatic colorimetric assay kit (Glycerol and Cholesterol Assay Kits, Cayman Chemical) according to the manufacturer's instructions.

Plasma very low-density lipoproteins (VLDL), low-density lipoproteins (LDL), and high-density lipoproteins (HDL) were separated by fast performance liquid chromatography (FPLC)-size exclusion chromatography analysis (gel filtration on a Superose 6 column (GE Healthcare) at a flow rate of 0.5 mL/min). The different lipoprotein

fractions were collected according to their retention times as follows: VLDL between 40 and 50 min, LDL between 50 and 70 min, and HDL between 70 and 90 min. Results are presented as optical density (OD) profiles at 492 nm of the isolated lipoproteins.

## Transwell migration assay

The chemotactic migration of primary murine B lymphocytes or human peripheral blood monocytes was assessed using a Transwell device[20]. In brief, primary human monocytes or mouse splenic B cells were isolated as described above and maintained in RPMI 1640 medium with 10% FCS and 1% P/S overnight. A suspension of 100 μL containing $1 \times 10^6$ cells was loaded in the upper chamber of the Transwell culture insert. Filters were transferred into the wells ('the lower chamber') containing different doses of MIF-2 with or without inhibitors. The inhibitory effects on MIF-2-mediated cell migration were determined by prior incubation of inhibitors at 37 °C for 1 h. The Transwell device was incubated for 4 h at 37 °C in a humidified atmosphere of 5% $CO_2$. Cells migrated into the lower chamber were collected and counted by flow cytometry using the CountBright™ Absolute Counting Bead (Invitrogen-Molecular probes, Karlsruhe, Germany) method.

## 3D chemotaxis assay – live imaging

The effect of MIF-2 on the motility and migratory ability of isolated human peripheral blood monocytes and mouse splenic B cells was studied using the 3D-Chemotaxis μ-Slide (Ibidi GmbH, Munich, Germany) device according to the manufacturer's instructions. Freshly isolated cells ($4 \times 10^6$) were seeded into a prepared gel matrix containing 1 mg/mL rat tail collagen type I (Merck Millipore, Darmstadt, Germany) in DMEM. To accelerate the polymerization process, the collagen gel containing the cell suspension was incubated at 37 °C for 30 min and then exposed to different concentrations of MIF-2 gradient (i.e. 0 nM, 2 nM, 4 nM, 8 nM, 16 nM, and 32 nM). Cell motility was monitored by performing time-lapse imaging every 2 min at 37 °C for 2 h using a Leica DMi8 microscope (Leica Microsystems). Images were imported as stacks into ImageJ software and analyzed with the manual tracking and the chemotaxis and migration tools (Ibidi GmbH).

The comparison between MIF- and MIF-2-mediated B-cell migration was evaluated under the same conditions with the corresponding optimal concentration that showed a high migratory effect, namely 8 nM and 4 nM for MIF and MIF-2, respectively.

## Flow adhesion assay

The effect of MIF-2 on monocyte cell adhesion under flow conditions was studied in vitro using a microfluidic chamber (μ-slide I 0.8, ibidi) and the ibidi pump system (ibidi). Briefly, a confluent monolayer of human aortic endothelial cells (HAoECs) was prepared by seeding ~$7 \times 10^4$ cells onto the microfluidic chamber precoated with collagen type I (ibidi), followed by overnight incubation at 37 °C under humidified conditions. Cells were then exposed to various concentrations of MIF-2 (i.e., 0.8 nM, 1.6 nM, 4 nM, 8 nM, 16 nM, 40 nM, and 80 nM) or MIF (16 nM) as a positive control for 2 h at 37 °C. MonoMac6 cells were also treated with the same concentrations of MIF-2 and MIF for 5 min, resuspended in assay buffer (1× HBSS, 10 mM HEPES, 0.5% BSA) to a final concentration of $0.6 \times 10^6$ cells/mL, and perfused through the flow chamber for 10 min at 37 °C with a laminar shear stress of 1.5 dyn/$cm^2$. The adherent cells were then imaged using an inverted light microscope (Leica DMi8), and quantification of images from four different fields of view of each chamber was performed using Image J software.

## In vivo homing assay

The homing assay was performed by injecting stained B lymphocytes into mice, from which various organs were subsequently harvested and analyzed by flow cytometry. B lymphocytes were first isolated from the spleens of wild-type (WT) mice as described above. The donor mice (WT mice) had the same littermates as the recipient mice (*Mif*$^{-/-}$ or WT mice). Cells were then stained with the intracellular Cell Tracker Green dye CMFDA (5-chloromethylfluorescein diacetate) resuspended in RPMI 1640 medium at a concentration of 25 μM and incubated for 45 min at 37 °C protected from light. Cells were washed twice with prewarmed isotonic sodium chloride solution and resuspended in the same solution. Three hours before cell injection, recipient mice (*Mif*$^{-/-}$ and WT mice) were administered intraperitoneally with 1 mg 4-IPP (a dual inhibitor of both MIF and MIF-2) or DMSO-containing solvent as a control. $20 \times 10^6$ cells were then injected into the tail vein of a recipient mouse under sterile conditions. After 2 h of migration, the mice were sacrificed, blood, bone marrow, spleen, and lymph nodes were collected, and single-cell suspensions were prepared. Homed B cells were analyzed by flow cytometry on the fluorescein isothiocyanate (FITC) channel, and the number of stained cells in the cell suspension was determined using CountBright™ Absolute Counting Bead (Molecular Probes).

## Internalization assay

Freshly isolated primary splenic B cells from wild-type (C57BL/6) or *Cd74*$^{-/-}$ mice ($1 \times 10^6$ cells) were resuspended in RPMI 1640 medium supplemented with 10% FCS and 1% P/S and then treated with various concentrations of recombinant MIF-2 (0 nM, 2 nM, 4 nM, 10 nM, 20 nM) for 20 min or treated with 4 nM MIF-2 at different time points (i.e. 0 min, 5 min, 10 min, 15 min, 20 min, 30 min) under a 5% $CO_2$ humidified atmosphere at 37 °C. After incubation, the internalization process was stopped by placing the cells on ice for 10 min. Cells were then washed twice with ice-cold PBS followed by flow cytometry buffer (PBS containing 0.5% BSA) and incubated for 1 h in the dark at 4 °C with phycoerythrin (PE)-conjugated anti-CXCR4 antibody or the appropriate PE-conjugated isotype control (IgG). After incubation, cells were washed thoroughly and analyzed using the BD FACSVerse™ (BD Biosciences). Quantification of the measurements was performed using FlowJo software.

## CXCR4-specific yeast reporter signaling assay

The *S. cerevisiae* strain (CY12946) expressing functional human CXCR4 protein, which replaces the yeast STE2 receptor and is linked to a MAP kinase signaling pathway and β-galactosidase (lacZ) readout is a well-established human CXCR4-specific receptor binding and signaling cell system[52,54]. MIF was previously demonstrated to elicit a CXCR4-specific signaling in this system[52]. In short, yeast transformants stably expressing human CXCR4 were grown at 30 °C overnight in yeast nitrogen base selective medium (Formedium). Cells were diluted to an $OD_{600}$ of 0.2 in yeast extract-peptone-dextrose (YPD) medium and grown to an $OD_{600}$ of 0.3–0.6. Transformants were incubated with different concentrations of human MIF-2 (i.e. 0 μM, 2 μM, 4 μM, 8 μM, 16 μM, 32 μM) for 1.5 h. Moreover, 10 μM MIF-2 and 20 μM MIF were used to directly compare the binding of these proteins to CXCR4. β-galactosidase activity was measured by a commercial BetaGlo Kit (Promega, Mannheim, Germany) to assess the activation of CXCR4 signaling, and $OD_{600}$ was detected by a plate reader with a 600 nm filter.

## Fluorescence spectroscopy

Fluorescence-spectroscopic titrations between the soluble CXCR4 surrogate molecule msR4M-L1 and MIF-2 were performed using a JASCO FP-6500 fluorescence spectrophotometer. MIF-2 was reconstituted in 20 mM sodium phosphate buffer (pH 7.2). The N-terminal fluorescein-labeled peptide msR4M-L1 (abbreviated as Fluos-msR4M-L1) was generated by introducing Nα-fluorescein into msR4M-L1 following the method outlined in Kontos et al.[54]. Briefly, msR4M-L1 was synthesized as C-terminal amide on Rink amide MBHA resin by SPPS using Fmoc chemistry. To introduce Nα-fluorescein, 5(6)-carboxyfluorescein-N-hydroxysuccinimide ester was coupled N-terminally to side chain-

protected msR4M-L1 on solid phase, after Fmoc-deprotection. For the spectroscopic measurements, stock solutions of Fluos-msR4M-L1 in 1,1,1,3,3,3-hexafluoro-2-propanol (HFIP) (Sigma-Aldrich), prepared on ice, were used. These measurements were performed in 10 mM sodium phosphate buffer (pH 7.4) containing 1% HFIP, Fluos-msR4M-L1 (5 nM), and various amounts of MIF-2 at room temperature. The 500–600 nm spectra shown are representative of one of three independent experiments. Apparent $K_D$ values were calculated assuming a 1:1 binding model[54,105], using sigmoidal curve fitting with OriginPro 2016 (OriginLab Corporation, Northampton, MA, USA).

### Sequence alignment
The alignment between MIF and MIF-2 sequences was performed by the ClustalW algorithm (http://www.genome.jp/tools-bin/clustalw) using standard parameters in the Jalview multiple sequence alignment editor desktop application.

### Molecular-docking analysis
The molecular interactions for MIF-2/CXCR4, MIF/CXCR4, and MIF-2/msR4M-L1 complexes were analyzed using HADDOCK docking simulations (HADDOCK Webserver 2.4)[106,107]. Structures were generated using the following Protein Data Bank (PDB) files: MIF-2/D-DT (PDB ID: 7MSE), CXCR4 (PDB ID: 3OE6), and MIF (PDB ID: 1MIF). The peptide structures of msR4-L1 were predicted using the HPEPDOCK web server[108]. The four best predictions were selected and overlaid with D-DT(MIF-2)-CXCR4 docking to select the best spatially matching peptide structure compared to extracellular loop (ECL)-1 (ECL1) and ECL2 of CXCR4. The MIF-2/msR4M-L1/CXCR4 complex was generated by a spatial overlay of the MIF-2/CXCR4 and the MIF-2/msR4M-L1 docking simulation. All complexes were finally modeled using UCSF Chimera. Data were analyzed using Matlab2022a (MathWorks, Natick, MA, USA).

### DiI-LDL uptake assay in PBMC-derived macrophages
The uptake assay of DiI complex-labeled low-density lipoprotein (DiI-LDL) in primary human monocyte-derived macrophages was performed following an established protocol[54,55]. Briefly, macrophages were cultured in the full RPMI 1640 medium at 37 °C, and then changed to MEM medium containing 0.2% BSA for 2–4 h to achieve starvation. Later, cells were pre-incubated with inhibitors (10 μM AMD3100, 4-IPP, or 4-CPPC) for 30 min along with 1 μg/mL MIF-2 overnight. The second day, cells were cultured in the same medium added with 1% 2-hydroxypropyl-β-cyclodextrin (HPCD) (Sigma-Aldrich) at 37 °C for 45 min. After rinsing with PBS three times, macrophages were maintained in 25 μg/mL DiI-LDL solution at 4 °C for 30 min (the 'binding step'), and subsequently moved to 37 °C for 20 min (the 'uptake step'). Treated macrophages were washed with PBS, fixed with 4% PFA, permeabilized with 0.1% Triton X in PBS, and stained with DAPI (1:100,000). Representative images were acquired with a Leica DMi8 fluorescence microscope and the uptake effect was characterized as the index of relative corrected total cell fluorescence (CTCF) via ImageJ and Excel.

### In situ proximity ligation assay (PLA)
The interaction and formation of the heteromeric complex between CXCR4 and CD74 was studied by in situ PLA (Duolink II Detection Kit, Olink Bioscience, Uppsala, Sweden). For this purpose, NIH/3T3 fibroblast cells were transfected with N-terminal FLAG-tagged hCD74 and C-terminal Myc-tagged hCXCR4 plasmids using PolyFect (Qiagen) according to the manufacturer's instructions. After 48 h, cells were fixed with 4% PFA solution, permeabilized with PBS solution enriched with 0.2% Triton X-100 for 10 min, and blocked with 5% FCS in PBS for 1 h. Cells were washed thoroughly with PBS and incubated overnight at 4 °C with primary antibodies, i.e. mouse anti-cMyc (9E10) mAb (Santa Cruz, Dallas, TX, USA; 1:200) and rabbit anti-DYKDDK Tag (D6W5B) mAb (anti-FLAG, Cell Signaling Technologies, Heidelberg, Germany;

1:200). After three washing steps, cells were treated with PLA according to the manufacturer's protocol and first incubated for 1 h at 37 °C with PLA probes (anti-rabbit PLUS and anti-mouse MINUS), which served as secondary antibodies and were conjugated to specific oligonucleotides. After hybridization, ligation, and detection with the detection kit (OLINK Bioscience, Uppsala, Sweden), nuclei were stained with DAPI, cells were covered with embedding medium (Invitrogen), and imaged with a Zeiss LSM 710 confocal microscope.

### Two (multi)-photon microscopy and FLIM-FRET imaging
Huh-7 cells or HEK293 cells were transfected with an equimolar amount of C-terminal ECFP-tagged hCXCR4 and N-terminal ECFP-tagged hCD74 plasmids using PolyFect (Qiagen) according to the manufacturer's instructions. After 48 h, cells were fixed with 4% PFA solution and washed with PBS. To investigate the effect of MIF-2 stimulation on the formation of the heteromeric receptor complexes between CXCR4 and CD74, transfected cells were stimulated for 10 min at 37 °C with MIF-2 (4 nM) or with MIF (8 nM) as a comparative control, and the fluorescence lifetime of the donor (ECFP-CXCR4) was monitored by fluorescence-lifetime imaging (FLIM) analysis.

Colocalization and heteromeric complex formation between CXCR4 and CD74 were visualized using a two (multi)-photon (2PM/MPM) Leica TCS SP8 DIVE multispectral microscope (Leica) equipped with a tunable extended IR spectrum laser (New InSight® X3™, Spectra-Physics; 680–1300 nm and 1045 nm fixed IR laser; pulsed) and a HC PL IRAPO 25×/1.0 water objective. Images were acquired in a sequential scan mode, detected with hybrid diode reflected light detectors, (HyD-RLDs) (CFP: excitation 840 nm/emission 445–524 nm; YFP: excitation 950 nm/emission 441–560 nm), and were processed using the LAS-X software package. Deconvolution was performed using Leica LIGHT-NING deconvolution algorithms.

For FLIM/FRET imaging, up to 1000 photons per pixel were acquired in a time-correlated single photon counting (TCSPC) mode. Images were analyzed using Leica FALCON software by fitting the lifetime decay curve for each pixel of the image in a multi-exponential model. The fluorescence lifetime of the donor (ECFP-CXCR4) was similarly determined in the absence of the acceptor (EYFP-CD74) using Leica FALCON software. The FLIM-FRET efficiency was automatically calculated by the software using the following equation: FRET eff = 1 − (τDA/τD), where τDA is the lifetime of the donor in the presence of the acceptor and τD is the lifetime of the donor without the acceptor.

### Native LDL uptake assay in Huh-7 hepatocytes
Huh-7 cells (~0.5 × 10⁶ cells) were seeded in 24-well plates and maintained overnight in DMEM growth medium supplemented with 2% FCS and 1% P/S. Cells were first incubated with AMD3100 (10 μg/mL), anti-human CD74 (LN2) (10 μg/mL), or IgG control for 1 h at 37 °C and then stimulated with 8 nM MIF-2 (100 ng/mL) for 24 h. Cells were treated with serum-free medium containing 25 μg/mL native LDL for 4 h. Excess LDL was removed by washing the cells thoroughly with PBS followed by fixation with 4% PFA for 20 min at RT with gentle shaking. Cells were washed extensively and stained with Oil-Red-O (ORO, Sigma-Aldrich) for 10 min. Cell nuclei were counterstained with DAPI for 5 min at RT in the dark, and LDL uptake was analyzed and visualized using a Leica DMi8 microscope.

### In vitro steatosis assay in hepatocytes
The steatosis assay was conducted using the Steatosis Colorimetric Assay kit (Cayman Chemical Company, Ann Arbor, MI, USA) following the manufacturer's instructions. Briefly, Huh-7 cells were seeded at a density of 1 × 10⁴ cells per well in a 96-well plate, starved for 2 h in DMEM medium without FCS (Gibco), and treated with various concentrations of MIF-2, with or without 1 mM oleic acid (OA), for 24 h. Following incubation, the cells were fixed at room temperature for 15 min using a fixative solution and washed twice with a washing

solution. Lipid accumulation was detected by incubating the cells with ORO staining solution for 30 min. Excess ORO was removed through additional washing steps. Finally, the cells were washed, resuspended in PBS, and lipid accumulation was imaged using a Leica DMi8 microscope.

## In vitro triglyceride synthesis assay in hepatocytes

The effect of MIF on de novo triglyceride synthesis was evaluated using the Triglyceride-Glo™ Assay kit (Promega, Mannheim, Germany) according to the manufacturer's protocol. Briefly, Huh-7 cells were seeded at a density of $1 \times 10^4$ cells per well in a 96-well plate, starved for 2 h in FCS-free DMEM medium, and stimulated with various concentrations of MIF-2 for 24 h at 37 °C. After incubation, the cells were washed with PBS and lysed at room temperature for 5 min using 50 μL of glycerol lysis solution containing lipoprotein lipase. Next, 50 μL of the triglyceride detection reagent was added to the lysates, and the plate was incubated at room temperature for 1 h, protected from light. The de novo synthesis of triglycerides mediated by MIF-2 was determined by measuring glycerol concentrations using a luminescence readout on an EnSpire Multimode Plate Reader (Perkin Elmer). Triglyceride levels were quantified by comparing the luminescence of treated and untreated samples against a standard curve.

## Bulk RNA sequencing of liver tissue from atherosclerotic mouse models

### Bulk RNAseq from frozen liver section embedded in OCT

**Sample preparation, RNA purification, and library preparation.** Thawed frozen liver sections, embedded in OCT, were immersed in 50 μL of freshly prepared RNA isolation buffer containing 20% proteinase K (Qiagen) in buffer PKD (Thermo Fisher Scientific) for approximately 30 s to allow lysis and homogenization of the fixed tissue. Using a 200-μL tip, the tissue lysate was scraped from the glass slide under the microscope. The lysates from 3 to 4 liver sections were combined and frozen at −80 °C. RNA was purified from the lysates and washed with oligo dT25 magnetic beads (Thermo Fisher Scientific). The beads were then removed by incubating the samples at 80 °C followed by placing the samples on a magnet.

To initiate library construction, 1 ng RNA extracted from each sample was used to undergo reverse transcription and subsequent cDNA amplification through the Smart-seq2 protocol. This process was then followed by library construction, following established procedures[80]. Briefly, samples were thawed and subjected to a 3-min incubation at 72 °C, transferred to ice, and progressed through the steps of reverse transcription (RT) and pre-amplification. The obtained cDNA underwent purification using AMPure beads (Beckman-Coulter) at a 0.7:1 beads-to-PCR product ratio. Evaluation of the libraries was performed using the Bioanalyzer (Agilent, catalog no. 2100) with a High Sensitivity DNA analysis kit. Additionally, cDNA concentrations were measured fluorometrically applying Qubit's DNA HS assay kits and a Qubit 4.0 Fluorometer (Invitrogen). Subsequently, cDNA concentrations were normalized to 160 pg/μL, and 1.25 μl of these cDNA samples were employed for library construction, employing an in-house Tn5 transposase. The libraries were barcoded and pooled together before undergoing a clean-up process involving three rounds of AMPure beads (Beckman-Coulter) at a ratio of 0.8:1 beads to library. Subsequently, libraries were sequenced in a paired-end fashion with reads of $2 \times 100$ bp on a DNB-seq platform, achieving an average depth of $3 \times 10^6$ reads per sample. The sequencing procedure was carried out by BGI Tech Solutions Co., Limited (Hong Kong, China).

**Bio-computational analysis of RNAseq data.** The analysis of RNAseq data was conducted on the Galaxy platform (https://usegalaxy.org/) following a previously outlined protocol[109]. Initially, FastQC software assessed the sequence quality of the raw data files. Post-quality

control, the fragments underwent 5′ and 3′ adapter trimming and were filtered to include only reads ≤20 bp using *cutadapt* software. The resulting trimmed reads were aligned to a reference mouse genome using HISAT2 software, allowing quantification of reads per gene, specifically those mapping to gene exons. Identified genes were annotated, and DESeq2 was employed to analyze genes displaying differential expression between groups, with an assigned expressed gene $P$-value threshold of <0.05. Volcano plots were created using the EnhancedVolcano package (1.22.0) to visualize differentially expressed genes, plotting $\log_2$ fold change against $-\log_{10}$ adjusted $P$-value. Genes showing significant (threshold: $-\log_{10}(P_{adj}) > 1.5$) up- or down-regulation by $\log_2$ fold change >1.5 are represented in the Volcano plot by red dots, while non-significant genes with $\log_2$ fold change <1.5 are depicted as dark-gray dots. Green dots represent genes with $\log_2$ fold change >1.5 and $-\log_{10}(P_{adj}) < 1.5$, while blue dots are $\log_2$ fold change <1.5 and $-\log_{10}(P_{adj}) > 1.5$.

### Bulk RNAseq from liver lysates

**Sample preparation, RNA purification and quality assessment.** RNA was isolated from liver tissue using the RNeasy Mini kit (Qiagen). In brief, 20 mg of liver tissue from atherogenic *Apoe*⁻/⁻ and *Mif-2*⁻/⁻ *Apoe*⁻/⁻ mice were dissected and placed in buffer RLT containing β-mercaptoethanol. The samples were thoroughly homogenized using the TissueLyser LT (Qiagen). Following centrifugation, the supernatant was collected and mixed with an equal volume of 70% ethanol. The mixture was transferred to a RNeasy spin column for RNA isolation, which was carried out through a series of washing steps, followed by elution in RNase-free water. The concentration and purity of the isolated RNA were assessed using the Qubit RNA IQ Assay kit (Invitrogen), and measured on a Qubit Fluorometer (Invitrogen) according to the manufacturer's protocol. The resulting Qubit RNA IQ score indicated the quality and integrity of the RNA, with higher scores reflecting RNA that is suitable for RNAseq and RT-qPCR.

**Library preparation and sequencing.** RNA sequencing libraries were prepared using the Prime-Seq protocol and processed using the zUMIs pipeline following an established workflow[110,111].

**Data analysis.** Gene expression count data were imported into R 4.4.1 for downstream analysis. Gene names were mapped using a custom annotation file ("Simon_liver.gene_names.txt"). Sample metadata, including experimental condition and RIN values, were incorporated from the sample information file ("Simon_liver samples_RIN.txt"). To minimize noise from lowly expressed genes, we retained genes with counts of at least one in a minimum of three samples within either the *Mif-2*⁻/⁻ *Apoe*⁻/⁻ mice or *Apoe*⁻/⁻ groups. The top 10 most highly expressed genes were excluded to prevent skewing of downstream analyses. Mitochondrial genes, ribosomal RNA (rRNA) genes, and pseudogenes were removed by filtering out genes located on chromosome M, annotated as "rRNA" or "rRNA_pseudogene," or with names starting with "Gm" or ending with "Rik". Normalization of count data was performed using the DESeq2 package (1.44.0)[112], which calculates size factors to account for differences in sequencing depth and library composition. Differential gene expression analysis was conducted using DESeq2. A DESeqDataSet object was created with the design formula '~Condition + RIN' to adjust for potential confounding due to RNA quality differences. Variance-stabilizing transformation (VST) was applied to the normalized counts to stabilize variance across the mean expression levels using the 'varianceStabilizingTransformation' function. Statistical testing for differential expression between the *Mif-2*⁻/⁻ *Apoe*⁻/⁻ or *Apoe*⁻/⁻ mice was performed using the Wald test implemented in DESeq2. Genes with an absolute $\log_2$ fold change greater than 0.5 and an adjusted $P$-value ($P_{adj}$) less than 0.05, after Benjamini–Hochberg correction for multiple testing, were considered significantly differentially expressed.

Volcano plots were created using the EnhancedVolcano package (1.22.0) to visualize differentially expressed genes, plotting $\log_2$ fold change against $-\log_{10}$ adjusted *P*-value. Genes showing significant (threshold: $P_{adj} < 0.05$) up- or downregulation by $\log_2$ fold change >0.5 are represented in the Volcano plot by red dots, while non-significant genes with $\log_2$fold change <0.5 are depicted as dark-gray dots. Green dots represent genes with $\log_2$fold change >0.5 and $P_{adj} > 0.05$, while blue dots are $\log_2$fold change <0.5 and $P_{adj} < 0.05$.

**Gene ontology analysis.** Gene ontology enrichment analysis was conducted using the clusterProfiler package (4.12.0)[113]. Upregulated and downregulated genes ($\log_2$ fold change > |1.5| or |0.5| ; $P_{adj} < 0.05$) were analyzed separately. Mouse gene annotations were obtained from the org.Mm.eg.db package (3.19.1). Enrichment analyses focused on Biological Process (BP) terms with significance determined using the Benjamini–Hochberg adjusted *P*-values. GO terms with $P_{adj} < 0.05$ were considered significant. The top 15 enriched GO terms were visualized using dot plots generated by the 'dotplot' function.

## Lipidomics

**Sample homogenization and lipid extraction procedure.** Frozen liver samples were weighed into homogenization tubes with ceramic beads (1.4 mm). To each 1 mg of frozen liver tissue, 3 µL of a cooled mixture (4 °C) of ethanol/phosphate buffer (85/15, v/v) were added. Tissue samples were homogenized using a Precellys 24 homogenizer (PEQLAB Biotechnology GmbH, Germany) three times for 30 s at 5500 rpm and 4 °C, with 30 s pause intervals to ensure constant temperature. 15 µL (equivalent to 5 mg) of the liver homogenates were transferred into 1.5 mL glass vials together with 85 µL of MilliQ water ($H_2O$). For accurate quantification, 25 µL of a mix of 77 deuterated internal standards were then added to the samples (Ultimate SplashOne, dFA 18:1, dFA 20:4, dCer d18:0/13:0, Glu Cer(d18:1-d7/15:0), dLacCer d18:1/15:0, and 15:0-18:1-d7-PA (Avanti Polar Lipids, Alabaster, AL, USA)).

Lipid extraction was based on the protocol by Matyash et al.[114]. 160 µL of methanol (MeOH, Optigrade, Thermo Fisher) and 575 µL methyl tert-butyl ether (MTBE, LC grade) were added followed by incubation for 30 min on an orbital shaker DOS-10L (Neolabline, Heidelberg, Germany) at 300 rpm. For phase separation, 200 µL of $H_2O$ was added to each vial. The mixtures were vortexed, and the vials were centrifuged at $5000 \times g$ for 10 min at RT with a Sigma 4-5C centrifuge (Qiagen). The upper (organic) phase was transferred into new glass vials and evaporated with nitrogen gas using a TurboVap® 96 dual evaporator (Biotage, Uppsala, Sweden). The aqueous phase was again extracted with 100 µL MeOH and 300 µL MTBE. After addition of 100 µL $H_2O$, the samples were incubated for 5 min at RT at 300 rpm and then centrifuged for 10 min at $5000 \times g$. The organic phase was transferred into the respective vial from the first extraction step and evaporated to dryness with gaseous nitrogen. Samples were reconstituted in 275 µL running solvent (10 mM ammonium acetate in dichloromethane:MeOH (50:50, v/v)) and 267 µL were subsequently transferred into new vials with insert. For quality control purposes (QC-pool samples), 10 µL of each study sample were pooled in a 1.5 mL Eppendorf tube. After vortexing, 15 µL aliquots were prepared and extracted with the above-described procedure. Additionally, 3 blank samples consisting of 15 µL ethanol (EtOH)/phosphate buffer were prepared and extracted.

**Shotgun lipidomics measurements.** The DMS-SLA shotgun lipidomics assay is based on the method published by Baolong Su et al.[115]. All samples were measured with a SCIEX Exion UHPLC-system coupled to a SCIEX QTRAP 6500+ mass spectrometer equipped with a SelexION differential ion mobility interface (SCIEX, Darmstadt, Germany) operated with Analyst 1.6.3. 75 µL of the re-dissolved sample were injected using the running solvent (10 mM ammonium acetate in

dichloromethane:MeOH (50:50, v/v)) at an isocratic flow rate of 8 µL/min. After 9 min, the flow rate was ramped to 30 µL/min for 2 min to allow for washing. Each sample was analyzed using multiple reaction monitoring (MRM) in two consecutive flow injection analysis (FIA) runs. In the first run, phosphatidylcholines (PC), phosphatidylethanolamines (PE), phosphatidylglycerols (PG), phosphatidylinositols (PI), phosphatidylserines (PS), and sphingomyelins (SM) were separated with the SelexION DMS cell using field asymmetric ion mobility mass spectrometry (FAIMS) prior to analysis in the Turbo Spray IonDrive source of the mass spectrometer. To enhance the separation of the lipid classes, 1-propanol was used as a chemical modifier. In the second run, cholesteryl esters (CE), ceramides (Cer d18:1), dihydroceramides (Cer d18:0), lactosylceramides (LacCER), hexosylceramides (HexCER), phosphatidic acid (PA), lysophosphatidylcholines (LPC), lysophosphatidylethanolamines (LPE), lysophosphatidylglycerols (LPG), lysophosphatidylinositols (LPI), lysophosphatidylserines (LPS), free fatty acids (FA), diacylglycerides (DG), and triacylglycerides (TG), were measured with the DMS cell switched off. The mass spectrometer was operated with the following conditions: curtain gas 20 psi, ion source gas 1 14 psi, ion source gas 2 20 psi, collision gas medium, temperature 150 °C, separation voltage 3500 V, ion spray voltage +4200 and +4500 V in ESI+ mode and −4400 and −3300 V in ESI− mode for run 01 and 02, respectively. Prior to each batch, the DMS cell was tuned, and the stability and sensitivity of the instrument was checked with the EquiSPLASH mixture (Avanti Polar Lipids), by using the Shotgun Lipidomics Assistant software (SLA.v1.5; https://github.com/syjgino/SLA/tree/v1.5-keyV4). MRMs were used as described in spname_dict_V4_1,31.xlsx.

**Lipidomics data processing.** Sciex wiff files were converted to mzml using the msvonvertGUI tool (v3.0.22074) from proteowizard (https://proteowizard.sourceforge. io/download.html). The converted files were subsequently processed using the SLA software v1.5. Twenty MRM-scans were averaged while a maximum of 2 out of 20 scans were allowed to contain missing values. Lipids were quantified by calculating the area ratio between the analyte and the respective internal standard of known concentration. Lipid species concentrations, reported in nmol/g, were then corrected for Type-II isotopic overlap using lipid specific correction factors in ISOcorrectlistV4_1.31.

The shotgun lipidomics raw data set contained 1215 individual lipid species. Data were subsequently pre-processed using R (version 4.3.1). To assure high data quality, a multi-step procedure was applied: In the first step of this quality control (QC) procedure, lipids with missing values in more than 35% in the pool samples were discarded from the data set ($n = 154$). In the second step, the group-specific missingness was evaluated, i.e., whether a specific lipid is observed in only one of the biological groups. Lipids exhibiting a groupwise missingness of 50% in all groups were discarded from the data set ($n = 24$). Next, lipids that had a coefficient of variation (CV) > 25%, determined by the QC-pool samples, were removed from the data set ($n = 27$). The last quality control step comprised the calculation of the dispersion ratio (D-ratio) that uses the ratio of the technical variance (determined by QC-pool samples) and the total variance (variance of the biological samples) as a quality marker for each lipid[116]. We used a D-ratio cut-off of 50%, as this implies that the technical variance is higher than the biological variance ($n = 302$ lipids were removed).

After quality control, 708 lipid species remained in the liver data set, which contained 77 missing values (equivalent to 10% of the data set). Missing values were imputed using the GSimp imputation approach[117]. The algorithm was initialized using a quantile regression approach for left-censored missing data (QRILC). Missing values were then predicted with the Gibbs sampler approach using an elastic net model with alpha = 0.1 and lambda = 0.01. 50 iterations were used to optimize the value for each missing variable (iters_each) and 10 iterations for the whole data set (iters_all).

## Retrieval of human carotid artery plaques from CEA patients and analysis of MIF-2 expression by immunostaining and RT-qPCR

Human carotid artery plaques were retrieved during carotid artery endarterectomy (CEA) surgery at the Department of Vascular and Endovascular Surgery at the Klinikum rechts der Isar of the Technical University Munich (TUM). The study was approved by the local ethics committee at the Medical Faculty of the Klinikum rechts der Isar of the Technical University Munich (ethics approval 2799-10). All patients provided their written informed consent. Advanced carotid lesions with an unstable/ruptured or stable plaque phenotype were cut in ~50 mg pieces and preserved on dry ice. Plaque stability was defined as follows: An unstable lesion was characterized as a ruptured or rupture-prone plaque with a fibrous cap less than 200 μm thick, overlying the necrotic core. In contrast, stable lesions were defined as those with a thick fibrous cap (≥200 μm) or those lacking a lipid or necrotic core[118,119].

**Determination of MIF-2 expression by immunohistochemistry.** Human CEA tissue and healthy vessel control sections were fixed for 48 h in 2% zinc-paraformaldehyde at room temperature, paraffin-embedded, and cut into 5 μm-thick sections. Immunofluorescence staining of sections was performed applying the DAB+ kit (Abcam, ab64238) following a standard protocol. MIF-2 was detected with a polyclonal rabbit anti-D-DT antibody[29]. HRP-conjugated polyclonal goat anti-rabbit immunoglobulin (GE Healthcare, NA934V, 1:1000) was used as secondary antibody. Slides were counterstained with Mayer hematoxylin and staining analyzed with a Leica DMi8 fluorescence microscope.

**Determination of MIF-2 expression by RT-qPCR.** Homogenization of human carotid artery plaque tissue was performed in 700 μL Qiazol lysis reagent and total RNA was isolated using the miRNeasy Mini Kit (Qiagen) according to the manufacturer´s instructions. RNA concentration and purity were assessed using NanoDrop. RIN number was assessed using the RNA Screen Tape (Agilent, Santa Clara, CA, USA) in the Agilent TapeStation 4200, and only samples with a RIN > 5 and DV200 of >60% were included for further analysis. Next, cDNA synthesis was performed using the High-Capacity-RNA-to-cDNA Kit (Applied Biosystems, USA) according to the manufacturer's instructions. Quantitative real-time PCR was performed in 96-well plates with a QuantStudio3 Cycler (Applied Biosystems, Foster City, CA, USA), using TaqMan Gene Expression Assays (Thermo Fisher) for detection of RPLPO (housekeeping) and MIF-2/D-DT.

## Reanalysis of transcriptomics dataset from human carotid artery plaques

**Reanalysis of a gene chip array dataset from human carotid artery plaques.** The GSE41571 dataset was downloaded from the Gene Expression Omnibus (GEO)[75] and reanalyzed for the expression of MIF-2/D-DT and MIF in human stable and unstable carotid plaques. The dataset contained transcriptomics data from 5 ruptured and 6 stable human atheromatous plaques from carotid endarterectomy (CEA) specimens, from which RNA was isolated following laser-microdissection of plaque tissue[75]. Transcripts were then profiled using Affymetrix HG-U133 plus 2.0 GeneChip arrays. Normalized MIF-2/D-DT and MIF expression was calculated and compared using unpaired *t*-test.

**Reanalysis of single-cell RNA sequencing data sets of human carotid artery plaques.** Human single-cell RNAseq datasets from patients undergoing CEA previously published by Alsaigh et al.[76] in the Genome Expression Omnibus (GSE159677) were reanalyzed according to the standard Seurat vignette using R and Seurat V4.2.0. The study was based on carotid artery tissue sections from CEA patients, from which cells were sorted on a MoFLo Astrios EQ cell sorter (Beckman-Coulter) and subjected to sequencing on a 10X Genomics Chromium single-cell 3′ v3 gene expression platform[76]. Data sets from three patients were filtered (QC), merged, and corrected for batch effects using canonical correlation analysis (CCA). Duplicates were removed using DoubletFinder[120]. The resolution was set to 0.58. Clusters were annotated according to the markers used by Alsaigh et al.[76].

## ELISA quantification of MIF-2 and MIF concentrations in plasma of CAD patients

**Patients and plasma samples.** Blood samples were collected from 149 consecutive patients with symptomatic coronary artery disease (CAD) during percutaneous coronary intervention (PCI), and plasma isolated[121,122]. Patients were admitted to the Department of Cardiology of the University Hospital of Tübingen, Germany. All subjects gave their written informed consent. CAD patients were sub-grouped into patients with chronic coronary symptoms (CCS, n = 85) and individuals presenting with acute coronary syndrome (ACS, n = 47) (for details, see Supplementary Table S3). ACS was defined as acute chest patient occurring with or without persistent ST-segment elevation as well as positive, or negative in case of unstable angina, cardiac enzymes. Myocardial infarction (MI) was defined as an acute myocardial injury with clinical evidence of acute myocardial ischemia and with detection of a rise and/or fall of cardiac troponin (cTn) values with at least one value above the 99th percentile upper reference limit and at least one of the following symptoms of myocardial ischemia: new ischemic changes. The CCS group included patients with suspected CAD, who presented with stable angina symptoms or newly diagnosed heart failure/left ventricular dysfunction; asymptomatic or symptomatic patients with stabilized CAD < 1 year after ACS or patients with recent revascularization, as well as patients >1 year after initial diagnosis or revascularization; and patients with vasospastic or microvascular angina as well as asymptomatic subjects, in whom CAD was detected at screening.

**MIF and MIF-2 ELISA assays.** To correlate the expression of MIF-2 with disease severity in CAD patients and compare it with MIF, the concentrations of MIF and MIF-2 in the plasma of CAD patients were examined by ELISA. The ELISA for human MIF-2 was performed following the protocol outlined in Merk et al.[29]. In brief, 96-well microtiter plates were coated with 15 μg/mL of polyclonal anti-D-DT, followed by washing and blocking with a solution of 1% BSA and 1% sucrose. Samples were added and incubated for 2 h. Subsequently, a biotinylated anti-D-DT antibody and a streptavidin-HRP conjugate were applied. D-DT concentrations were determined by measuring the optical density (OD) at 450 nm using a microplate reader, from which the D-DT concentration was calculated. The human MIF ELISA was conducted with a kit from R&D Systems[123]. In brief, 96-well microtiter plates were first coated overnight with rabbit polyclonal human anti-MIF-2 antibody or with the human monoclonal anti-MIF antibody MAB289 (R&D Systems). After extensive washing and blocking with 1% BSA and 1% sucrose, plasma samples were added and incubated overnight at 4 °C. Detection steps were performed by incubation with biotinylated anti-MIF-2 or with biotinylated anti-MIF (BAF289, R&D Systems) and a streptavidin-HRP conjugate for signal development. Recombinant MIF-2 and MIF were used as standards.

## Schematic illustrations

Icons and schematic illustrations representing experimental procedures or overviews (Fig. 1a, i, Fig. 2h, Fig. 4a, Fig. 5j, Supplementary Fig. 1a, Supplementary Fig. 20a, and Supplementary Fig. 32) were created in BioRender. Bernhagen (2025) https://BioRender.com/c59v181 and https://BioRender.com/l02i034 (Agreement number: DJ27RWDYPM).

## Statistics

Statistical analysis was performed using GraphPad Prism version 8 software. Data are represented as means ± SD. After testing for normality using the D'Agostino–Pearson test, data were analyzed by two-tailed Student's *t*-test, Mann–Whitney *U*-test, multiple *t*-tests, one-way ANOVA with Tukey's multiple comparisons test, or Kruskal–Wallis with Dunn's multiple comparisons test was applied as appropriate. Differences with *P* < 0.05 were considered to be statistically significant.

## Reporting summary

Further information on research design is available in the Nature Portfolio Reporting Summary linked to this article.

## Data availability

All data generated or analyzed in this study are provided within the manuscript and its supplementary information files. Source data are included alongside this paper. The reanalysis of the transcriptomics dataset from human carotid artery plaques used in this study is available in the Gene Expression Omnibus (GEO) database under the following accession codes: GSE159677; GSE41571; GSE131780; GSE155514. The RNAseq data from OCT-embedded frozen liver tissue sections of *Apoe*[−/−] and *Mif-2*[−/−]*Apoe*[−/−] mice after 12 weeks on a high-fat diet (HFD) are available in the Gene Expression Omnibus (GEO) under GSE281285. Additionally, RNAseq data from whole liver lysates can be accessed under GEO accession number GSE287230. The lipidomics data are referenced in MetaboLights under REQ20250117208170 (temporary code) and can be accessed at: (https://www.ebi.ac.uk/metabolights/editor/study/REQ20250117208170). Source data are provided with this paper.

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

## Acknowledgements

This work was supported by Deutsche Forschungsgemeinschaft (DFG) grant SFB1123-A3 to J.B. and A.K., DFG INST 409/209-1 FUGG to J.B., SFB1123-A1 to C.W. and Y.D., SFB1123-Z1 to R.T.A.M., SFB1123-B5 to L.M., SFB1123-B10 to A.B., SFB1123-A7 to C.S., SFB240-B01 (374031971 – TRR 240) to M.G., and by DFG under Germany's Excellence Strategy within the framework of the Munich Cluster for Systems Neurology (EXC 2145 SyNergy—ID 390857198) to J.B. and C.W. A.B. is supported by a German Center for Cardiovascular Research (DZHK) Junior Research Group Grant. T.H. is supported by the Clinician Scientist Program of the German Cardiac Society (DGK). R.B. is supported by US NIH 1R01-AR078334. C.Z. and B.Y. acknowledge support by the China Scholarship Council (CSC) fellowship program CSC/LMU. We are thankful to Dr. Yaw Asare and Dr. Sijia Wang for valuable discussions and to Kathleen Hille for technical assistance with peptide synthesis.

## Author contributions

O.E. and J.B. conceived the project and designed the experiments with contributions from C.Z., D.S., J.M., A.H., L.M., W.K., O.G., R.B., R.T.A.M., D.R., M.G., C.W., and A.K; O.E., C.Z., J.W., S.E., B.Y., N.K., E.B., P.B., M.A., C.K., M.Z., Y.J., W.E.K., N.S., N.W., D.M., H.J., S.G., M.B., M.H., F.R., and T.H. conducted experiments and analyzed data. NK, MH, FR, MA, and YD supported the steatosis/lipogenesis and lipidomics experiments; AB, NW, and MA worked on metabolism studies; and DM and CS contributed to the BMT/cell type analyses. O.B., C.Z., and J.B. wrote the initial version of the manuscript, and, after contributions from all authors, finalized the manuscript. L.M., A.H., R.B., D.R., M.G., C.W., and A.K. contributed critical materials; and L.M., O.G., A.H., R.B., R.T.A.M., A.B., C.S., M.G., C.W., A.K., and J.B. provided funding.

## Funding

## Competing interests

J.B., R.B., and C.W. are co-inventors of patents covering anti-MIF strategies for inflammatory and cardiovascular diseases. C.Ko., A.K., O.E., and J.B. are co-inventors of a patent application covering MIF-binding CXCR4 ectodomain mimics for inflammatory and cardiovascular diseases. The remaining authors declare no competing interests.

## Additional information

**Omar El Bounkari** [1,19] ✉, **Chunfang Zan**[1,19], **Bishan Yang**[1], **Simon Ebert** [1], **Jonas Wagner**[1], **Elina Bugar**[1], **Naomi Kramer**[1], **Priscila Bourilhon**[1], **Christos Kontos**[2], **Marlies Zarwel**[1], **Dzmitry Sinitski**[1], **Jelena Milic**[1], **Yvonne Jansen**[3], **Wolfgang E. Kempf**[4], **Nadja Sachs** [4,5], **Lars Maegdefessel**[4,5], **Hao Ji**[6], **Ozgun Gokce**[6,7,8,9], **Fabien Riols**[10], **Mark Haid** [10], **Simona Gerra**[1], **Adrian Hoffmann** [1,11], **Markus Brandhofer** [1], **Maida Avdic**[1], **Richard Bucala**[12], **Remco T. A. Megens** [3,5,13], **Nienke Willemsen**[3], **Denise Messerer**[14], **Christian Schulz** [14,15], **Alexander Bartelt** [3,5,16], **Tobias Harm**[17], **Dominik Rath**[17], **Yvonne Döring** [3,18], **Meinrad Gawaz** [17], **Christian Weber**[3,5,7,13], **Aphrodite Kapurniotu**[2] & **Jürgen Bernhagen** [1,5,7] ✉

[1]Division of Vascular Biology, Institute for Stroke and Dementia Research (ISD), LMU Klinikum, Ludwig Maximilian University (LMU) Munich, Munich, Germany. [2]Division of Peptide Biochemistry, TUM School of Life Sciences, Technische Universität München (TUM), Freising, Germany. [3]Institute for Cardiovascular Prevention, LMU Klinikum, Ludwig Maximilian University (LMU) Munich, Munich, Germany. [4]Institute of Molecular Vascular Medicine, TUM Klinikum, Technische Universität München (TUM), Munich, Germany. [5]German Center for Cardiovascular Research (DZHK), partner site Munich Heart Alliance, Munich, Germany. [6]Systems Neuroscience Lab, Institute for Stroke and Dementia Research (ISD), LMU Klinikum, Ludwig Maximilian University (LMU) Munich, Munich, Germany. [7]Munich Cluster for Systems Neurology (SyNergy), Munich, Germany. [8]Department of Neurodegenerative Diseases and Geriatric Psychiatry, University Hospital Bonn Venusberg-Campus 1, Bonn, Germany. [9]German Center for Neurodegenerative Diseases (DZNE) Bonn, Munich, Germany. [10]Metabolomics and Proteomics Core, Helmholtz Zentrum, Neuherberg, Germany. [11]Department of Anaesthesiology, LMU Klinikum, Ludwig Maximilian University (LMU) Munich, Munich, Germany. [12]Yale University School of Medicine, New Haven, CT, USA. [13]Cardiovascular Research Institute Maastricht

(CARIM), Maastricht University, Maastricht, The Netherlands. [14]Department of Medicine I, LMU Klinikum, Ludwig Maximilian University (LMU) Munich, Munich, Germany. [15]Department of Immunopharmacology, Mannheim Institute for Innate Immunoscience (MI3), Medical Faculty Mannheim, University of Heidelberg, Mannheim, Germany. [16]Institute for Diabetes and Cancer (IDC), Helmholtz Center Munich, German Research Center for Environmental Health, Neuherberg, Germany. [17]Department of Cardiology and Angiology, University Hospital Tübingen, Eberhard-Karls-University Tübingen, Tübingen, Germany. [18]Division of Angiology, Swiss Cardiovascular Center, Inselspital, Bern University Hospital, University of Bern, Bern, Switzerland. [19]These authors contributed equally: Omar El Bounkari, Chunfang Zan. ✉e-mail: Omar.El_Bounkari@med.uni-muenchen.de; juergen.bernhagen@med.uni-muenchen.de

