## [Transparent Peer Review file · Nature Communications]

An atypical atherogenic chemokine that promotes advanced atherosclerosis and hepatic lipogenesis

Corresponding Author: Dr Omar El Bounkari

Version 0:

Reviewer comments:

Reviewer #1

(Remarks to the Author)

The manuscript presents substantial data demonstrating the role of MIF-2 as a novel atherogenic chemokine and CXCR4 and ligand associated with CAD. Moreover, the data supports the role of MIF2 in lesion formation and vascular inflammation, but also in hepatic lipogenesis in a CD74/CXCR4/SREBP-mediated manner. However, I have some minor comments.

1) How could it be explained that Mif-2 deficiency does not affect the collagen content in hyperlipidemic Apoe^{-/-} mice, since it was observed that there was a significant reduction in the accumulation of macrophages (CD68 cells in the lesion area and it is well documented that actually, a great proportion of the CD68 positive cells in the lesion are with VSMCs origin. Therefore it is expected that there is preservation of VSMC phenotype and subsequently of collagen accumulation, the authors should address this issue and demonstrate the percentage of Myh11 positive VSMCs (αSMA expression) in the atherosclerotic lesion as well as fibrous cap thickness .

2) MIF-2 enhanced foam cell formation, is MIF-2 expressed in VSMCs as major source of foam cells present in the atherosclerotic plaque.

Wang Y, Dubland JA, Allahverdian S, Asonye E, Sahin B, Jaw JE, Sin DD, Seidman MA, Leeper NJ, Francis GA. Smooth muscle cells contribute the majority of foam cells in ApoE (Apolipoprotein E)-deficient mouse atherosclerosis. *Arterioscler Thromb Vasc Biol.* 2019; 39:876–887. doi: 10.1161/ATVBAHA.119.312434

3) It is not clear why the authors check particularly the lipogenic genes Srebp-1 and Srebp-2 in hepatocytes?

4) In the discussion, the authors should include a paragraph on development of MIF2 -based therapeutic concepts in atherosclerosis and vascular inflammation.

Reviewer #2

(Remarks to the Author)

This is a comprehensive study providing evidence that MIF-2 is an atypical chemokine that regulates atherosclerosis and hepatic steatosis. Furthermore, they showed that MIF-2 acts in reciprocal fashion compared to MIF. Mechanistic studies suggest that MIF-2 interacts with its receptors, induces their interactions, and elicits signaling pathways that culminate in the suppression of Srebp1 activation. The main strength of the paper is about establishing the role of MIF-2 as an atypical chemokine.

The data about Srebp1 activation are convincing. However, its role in lipogenesis is mainly based on ORO that is a less sensitive assay. Authors can measure lipid levels in the liver to better define steatosis (9Fig 5C).

Authors can measure lipogenesis by incubating hepatocytes with radiolabeled precursors such as acetate, glycerol or oleic acid. Another issue that was not addressed related to reductions in plasma lipids. The DKO mice have significantly lower plasma lipids. How can authors explain that the decreases in inflammatory response are not secondary to reduced atherogenic lipoproteins?

How increased hepatic lipogenesis and reduced plasma lipids can be associated with reduced hepatic steatosis? If reduced plasma lipids are secondary reduced lipoprotein production, then increased hepatic lipogenesis and reduced lipoprotein production are expected to enhance hepatic steatosis.

Minor comments:

Do they have information about changes in body weight? And other physiological parameters?

Fig 4B: DKO mice aortic arch shows more ORO staining than Apoe KO mice. Please check whether pictures have been misplaced.

Did authors measure cholesterol and triglyceride in FPLC fractions? Such as analysis will provide better understanding about the changes in plasma lipids and lipoproteins.

Authors agreed that MIF-2 plays a role in lipogenesis. They highlighted that RNA-Seq analysis identified down regulation of apoA4, PCSK9 and Ces2. It is unclear how these proteins play a role in lipogenesis. It is also not known how purine metabolites play a role in lipogenesis.

Authors have not discussed the regulation of ACC and FASN in the DKO livers; two critical enzymes in lipogenesis.

Maybe they should de-emphasize lipid metabolism and mainly focus on chemokine studies.

Reviewer #3

(Remarks to the Author)

This article presents new findings on how MIF2, d-dopachrome tautomerase (DDT) regulates atherosclerosis and hepatic lipogenesis. Chemokines affecting both vascular and metabolic pathologies are rare, but DDT/MIF2 promotes atherosclerotic plaque formation and hepatic lipid accumulation. MIF2 deficiency or blockade of MIF2 by the pharmacological inhibitor, 4-CPPC successfully protected lesion formation and vascular inflammation in early and advanced atherogenesis. In vitro experiments demonstrate that MIF2 promotes monocyte migration and arrest, and B cell recruitment, and foam cell formation which are important steps in the development of atherogenesis. MIF2 regulates these immune functions by binding to the CXCR4 receptor. In the liver, MIF2 also binds to CXCR4 and CD74 to regulate the Akt/SREBP signaling pathway. Finally, MIF2 staining was enhanced in unstable human carotid plaques from atherosclerotic patients undergoing CEA. Plasma MIF2 levels are high in CAD patients and correlated with disease severity. All these data suggest an important role for MIF2 in mediating atherosclerosis and affecting hepatic lipogenesis. Overall, the current study is very well organized and presents for the first-time novel findings regarding the role of DDT in modulating atherosclerosis. The reviewer has some thoughts and questions about this manuscript as follows.

- 1) This study only compares Apoe^{-/-} and Mif2^{-/-}Apoe^{-/-} mice with HFD. It is unclear whether HFD upregulates plasma MIF2 compared to normal chow. If so, will MIF2 induce hepatic lipogenesis in 12-week HFD fed mice or this only happens in Apoe^{-/-} mice with HFD?
- 2) Where is MIF2 derived from? If it is majorly from immune cells, whether transplantation of Mif2^{-/-} bone marrow cells into Apoe^{-/-} could reverse the atherosclerotic phenotypes?
- 3) Was 4-CPPC also injected into Mif2^{-/-}Apoe^{-/-} as shown in Figure 11? But this experiment was not mentioned in the article.
- 4) Why do Mif2^{-/-} and MIF2 inhibition show reductions in different cytokine profiles? Does this suggest that MIF2 inhibitors may not be able to fully block the effects of MIF2?
- 5) 4.5 week HFD significantly regulates the formation of atherosclerotic plaques whether this short-term HFD feeding also affects liver lipogenesis or alteration of liver lipogenesis only occurs following 12 week HFD feeding.
- 6) Does Mif2^{-/-} affect food intake and is accompanied by weight loss as shown in Figure S9D?
- 7) In Figure 5D, F and N and Supplemental Figure S14, MIF2 treatment inhibits AMPK but activates Akt. Akt may be a key upstream regulator of SREBP in hepatocytes. However, in fact, high fat diet may induce insulin resistance, resulting in reduced Akt in the liver. Just curious if MIF2 improves insulin sensitivity in the liver in vivo?
- 8) In Figure 5N, reduction of AMPK may attenuate fatty acid oxidation, further exacerbating lipid storage. Fatty acid oxidation related genes may need to be evaluated, such as CPT1 which is also a downstream target of AMPK.
- 9) Additionally, many experimental details are missing from the manuscript. For example, what is the duration of MIF2 treatment in Figure 2A to C.

Version 1:

Reviewer comments:

Reviewer #1

(Remarks to the Author)

The authors have addressed precisely and extensively the raised by the authors issues. The manuscript represents sufficient novel data with translational relevance. I recommend Accept as is.

Reviewer #2

(Remarks to the Author)

Authors have made sincere attempts to address several of the questions.

Reviewer #3

(Remarks to the Author)

All of my questions have been addressed, I have no further questions and recommend acceptance of this manuscript for publication.

Point-by-point response to the Reviewers' critique

We thank all three Reviewers very much for taking the time and effort to review our manuscript and for their constructive and thoughtful criticism. We have addressed all comments made, most/many of them by **extensive new experimentation**. Please find our point-by-point response below. All changes and edits in the revised manuscript are highlighted in color by Word tracked changes mode; edits in the Supplemental Information file are marked in red. Clean versions of the revised manuscript file and of the Supplemental Information file are also included. Information about page and line numbers of the revisions in the revised manuscript below refers to the tracked changes version.

Reviewer #1:

General comments:

The manuscript presents substantial data demonstrating the role of MIF-2 as a novel atherogenic chemokine and CXCR4 and ligand associated with CAD. Moreover, the data supports the role of MIF2 in lesion formation and vascular inflammation, but also in hepatic lipogenesis in a CD74/CXCR4/SREBP-mediated manner. However, I have some minor comments.

Response: We thank the Reviewer for his/her concise summary, overall assessment and positive feedback.

1. *How could it be explained that Mif-2 deficiency does not affect the collagen content in hyperlipidemic Apoe^{-/-} mice, since it was observed that there was a significant reduction in the accumulation of macrophages (CD68 cells in the lesion area and it is well documented that actually, a great proportion of the CD68 positive cells in the lesion are with VSMCs origin. Therefore, it is expected that there is preservation of VSMC phenotype and subsequently of collagen accumulation, the authors should address this issue and demonstrate the percentage of Myh11 positive VSMCs (αSMA expression) in the atherosclerotic lesion as well as fibrous cap thickness.*

Response: Thank you for this important comment. We fully agree with the Reviewer that VSMC plasticity in the atherogenic vessel wall may give rise to significant phenotype switching to CD68+ macrophage-like SMCs. Following the Reviewer's advice, we have quantified the percentage of Myh11+ VSMCs in the aortic lesions employing a Cy3-conjugated anti-α-SMA antibody for staining of aortic root sections. The data indicate that lesional VSMCs remain unchanged when comparing *Mif-2*-deficient *Apoe^{-/-}* mice with *Apoe^{-/-}* controls (**new Supplementary Figure S9**). We also followed the Reviewer's suggestion and determined the fibrous cap (FC) thickness. Although a slight trend towards a reduction of the FC thickness was noticed in aortic root of *Mif-2*-deficient *Apoe^{-/-}* mice, this difference did not reach statistical significance (**new Supplementary Figure S8C,D**). This result is in line with our previous observation that collagen content in the lesion remained constant, as collagen is primarily produced by VSMCs. Thus, together with the observed lack of difference for plaque collagen content (previous Supplementary Figure S8A,B), these data provide a consistent picture that VSMCs and VSMC-dependent parameters such as collagen content and FC thickness are unchanged between *Mif-2*-deficient *Apoe^{-/-}* mice and *Mif-2*-expressing *Apoe^{-/-}* control mice in the examined atherogenic model after 12 weeks of HFD. We agree with the Reviewer that this finding is perhaps unexpected, given that plaque macrophages were reduced in the *Mif-2*-deficient *Apoe^{-/-}* mice and that VSMCs may transdifferentiate into CD68+ macrophage-like phenotypes, but have the following mechanistic explanations, which will also be incorporated as new text paragraph into the revised manuscript (**page 12 of revised manuscript Results text and page 24 of the revised manuscript Discussion text**).

The most likely explanation, in our view, is that the transdifferentiation of VSMCs into plaque macrophages or CD68+ macrophage-like SMCs in Western-type HFD-fed *Apoe^{-/-}* mice does not occur until 12 weeks of HFD feeding. In fact, fate-mapping studies by the Greif laboratory showed that the phenotypic switch of VSMCs towards macrophages and their occurrence in the intima is typically seen 12-16 weeks after start of the HFD, whereas non-SMC derived

macrophages dominate the lesional macrophage pool between 5 and 12 weeks of HFD (Misra et al *Nat Commun* 2018¹). Potential other explanations are: i) *Mif-2* deficiency might impede the phenotypic switch of VSMCs into macrophages; ii) once VSMCs have switched, *Mif-2* deficiency might impair macrophage proliferation or promote macrophage apoptosis; however, mechanistic studies to address these possibilities would be beyond the scope and revision timeline of this manuscript. In conjunction with our observation that MIF-2 promotes monocyte migration and LDL uptake *in vitro* (see Figures 2A-G and Figure 3M or the initial manuscript version), it appears overall plausible to us that the observed decrease in CD68+ macrophages seen in *Mif-2*-deficient mice in Figure 1E is probably mainly a result of decreased monocyte infiltration and/or macrophage foam cell formation. To this end, we also thank the Reviewer for alerting us to the study by Wang et al (ATVB 2019²) providing evidence that SMCs contribute the majority of atheroma foam cells (leukocyte and non-leukocyte) in *ApoE*^{-/-} mice (see also next point).

2. *MIF-2 enhanced foam cell formation, is MIF-2 expressed in VSMCs as major source of foam cells present in the atherosclerotic plaque.* Wang Y, Dubland JA, Allahverdian S, Asonye E, Sahin B, Jaw JE, Sin DD, Seidman MA, Leeper NJ, Francis GA. *Smooth muscle cells contribute the majority of foam cells in ApoE (Apolipoprotein E)-deficient mouse atherosclerosis. Arterioscler Thromb Vasc Biol* 2019; 39:876–887. doi: 10.1161/ATVBAHA.119.312434.

Response: Thank you! This is a very interesting point as well. To address this question, we have performed a re-analysis of a single-cell RNAseq data set from aorta of *ApoE*^{-/-} mice subjected to 8 or 16 weeks of HFD. This scRNAseq study had been performed by the Quertermous group (Wirka et al, *Nat Med* 2019) in the context of their comprehensive publication on the characterization of the atheroprotective roles of smooth muscle cell phenotypic modulation³. The study focused on the analysis of plaque resident cell types, and thus, in addition to immune cells, shows an excellent resolution of aortic root parenchymal cell types, including three (V)SMC clusters. Interestingly, *Mif-2/D-dt* expression was found to be prominent in all three SMC clusters, SMC1, SMC2, and the so-called modulated SMCs, whereas only little expression was noted in fibroblasts or any other resident plaque cell types. SMC1 are characterized by marker genes such as *Acta1*, *Cnn1*, *Myl9*, *Myh11*, *Tagln*, or *Tpm2*, and represent contractile SMCs. SMC2 were characterized by marker genes such as *Clec3b*, *Gpx3*, or *Serp1*. Modulated SMCs prominently expressed markers such as *Lgals3*, *Spp1*, *Fn1*, or *Timp-1*. SMC phenotypic modulation depends on transcription factor 21 (Tcf21) expression, a gene associated with CAD. The study further concluded that SMCs undergoing phenotypic modulation shifted along a continuous trajectory from a contractile SMC towards a fibroblast-like cell, termed “fibromyocyte” (**new Supplementary Figure S10**).

Broad expression of *Mif-2/D-dt* expression in SMC cell clusters is further confirmed by re-analysis of the scRNAseq data set applied in a study from Muredach Reilly’s laboratory (Pan et al, *Circulation* 2020⁴). Those authors combined SMC fate mapping and scRNAseq of atherosclerotic plaques (among other analyses) from atherogenic *Ldlr*^{-/-} mice and identified three SMC cell populations: i) SMCs (with features of contractile SMCs); ii) SMC-derived intermediate cells, termed “SEM” cells (stem cell, endothelial cell, monocyte) or “ICS” cells that can differentiate into macrophage-like and fibrochondrocyte-like cells; iii) as well as a minor SMC cell cluster. Our re-analysis of that data set shows that *Mif-2/D-dt* is highly expressed in all three SMC clusters, with some expression also seen in fibroblast clusters and one macrophage population (**new Supplementary Figure S11**).

Together, our re-analysis of those two scRNAseq data sets from two different atherogenic mouse models indeed confirm the Reviewer’s view that the *Mif-2/D-dt* gene is prominently expressed in VSMCs. Whether this translates into substantial protein expression and leads to *Mif-2/D-dt* secretion from plaque VSMCs with functional consequences for atherosclerotic plaque development is unknown and is a very relevant subject for future studies. In fact, we are currently teaming up with the Stark laboratory at LMU Munich University hospital, an expert on spatially-determined functional VSMC differences in the plaque⁵ to further and systematically explore this question. These studies are just beginning, will take 1-2 years, and we hope that the Reviewer can agree that they are beyond the scope of our revision timeline for

the current manuscript. The observed substantial expression of the *Mif-2/D-dt* gene in VSMCs as determined by the above re-analysis of aortic plaque scRNAseq data sets also would be in support of the Reviewer's hypothesis that VSMC-derived MIF-2 may contribute to foam cell formation. Thus, MIF-2, which promotes LDL uptake and foam cell formation of macrophages (our *in vitro* data in this manuscript; see Figure 3M-N), may act through autocrine or paracrine mechanisms to mediate macrophage and/or VSMC foam cell formation. We have added a corresponding paragraph in the Discussion section of the revised manuscript (**page 24-25 of revised manuscript text**).

3. It is not clear why the authors check particularly the lipogenic genes *Srebp-1* and *Srebp-2* in hepatocytes?

Response: Thank you for bringing this up and we apologize for not having explained well-enough the rationale for analyzing SREBP-1 and SREBP-2. The reason, we initially looked at SREBP transcription factors and hypothesized that the expression and/or activity of hepatocyte SREBP-1 and -2 might be (co-)regulated by MIF-2, was that these transcription factors: i) are key factors controlling lipogenesis in the liver; ii) and that previous reports had convincingly shown causal links between SREBPs and not only dyslipidemia and metabolic disease but also atherosclerosis^{6,7}. Moreover, one of those studies⁶ also suggested a potential connection between SREBPs, atherosclerosis, and hepatic AMPK, a metabolic kinase that had previously been linked to roles of MIF (MIF-1) in the liver and heart^{8,9}. To better account for this rationale, we have added the following text to the respective Results paragraph: "Based on previous data suggesting a causal link between hepatic SREBP proteins, a family of key lipogenic transcription factors, and metabolic disease and atherosclerosis^{6,7}, we hypothesized that Srebps could play a role in the observed liver lipid phenotype in *Mif-2^{-/-}Apoe^{-/-}* mice." (**page 14 of revised manuscript text**), and have added a corresponding paragraph to the Discussion (**page 27 of revised manuscript text**). Furthermore, evidence from our bulk RNAseq analysis of liver tissue (see the RNAseq analysis from liver sections shown in previous Figure 5K-L; and new bulk RNAseq analysis from mRNA preparations of liver lysates, see **new Supplementary Figure S18** and response to question #4 of Reviewer #2 below) further underscores a potential role of SREBPs and SREBP-regulated downstream genes. For example, *Srebp2* was found to be down-regulated in *Mif-2*-knockout livers compared to the control group ($P=0.08$; bulk RNAseq analysis from liver sections, previous Figure 5K) and various *Srebp*-controlled genes associated with hepatic lipogenesis were found to be down-regulated in *Mif-2*-knockout liver according to both bulk RNAseq from liver sections (previous Figure 5K-L) and bulkRNAseq from liver lysates (**new Supplementary Figure S18**). This notion was confirmed by RT-qPCR data from liver tissue of *Apoe^{-/-}* and *Mif-2^{-/-}Apoe^{-/-}* mice, which showed reduced hepatic expression of *Srebp2* as well as several genes related to both *Srebp-1* and *Srebp-2* in *Mif-2^{-/-}Apoe^{-/-}* mice. Specifically, *Srebp-1*-regulated genes involved in fatty acid synthesis, such as *Acc* and *Fasn*, and *Srebp-2*-regulated genes involved in cholesterol uptake and synthesis, including *Hmgr*, *Hmgcs1*, and *Acat1*, were downregulated (**new Supplementary Figure S19**).

4. In the discussion, the authors should include a paragraph on development of MIF2-based therapeutic concepts in atherosclerosis and vascular inflammation.

Response: Thank you. Following the Reviewer's advice, we have added a **paragraph on MIF-2-based therapeutic concepts in the Discussion section (page 31-32 of the revised manuscript)**, discussing MIF-2-targeting antibodies, small molecule compounds and peptides available in the field and their potential therapeutic utility in atherosclerosis and vascular inflammation.

Reviewer #2:

General comments:

This is a comprehensive study providing evidence that MIF-2 is an atypical chemokine that regulates atherosclerosis and hepatic steatosis. Furthermore, they showed that MIF-2 acts in reciprocal fashion compared to MIF. Mechanistic studies suggest that MIF-2 interacts with its receptors, induces their interactions, and elicits signaling pathways that culminate in the suppression of Srebp1 activation. The main strength of the paper is about establishing the role of MIF-2 as an atypical chemokine.

Response: We thank the Reviewer for his/her positive feedback and nicely summarizing our work.

Major points:

The data about Srebp1 activation are convincing. However, its role in lipogenesis is mainly based on ORO that is a less sensitive assay. Authors can measure lipid levels in the liver to better define steatosis. Authors can measure lipogenesis by incubating hepatocytes with radiolabeled precursors such as acetate, glycerol or oleic acid.

Response: Thank you for your comment and the suggestions. These are important points. To address them, we have performed: i) an in vitro steatosis assay in Huh-7 hepatocytes; ii) a triglyceride synthesis assay in Huh-7 hepatocytes using glycerol as readout; iii) and, more relevant, regarding *in vivo* significance, a comprehensive lipidomics analysis from liver tissue of Mif-2^{-/-} Apoe^{-/-} versus Apoe^{-/-} mice. Regarding the *in vitro* lipogenesis assay, we opted to apply a commercial kit, i.e. the Triglyceride-Glo™ assay kit (Promega), due to limitations in work with radioactive material, since our institute is no longer equipped with an isotope lab. Setting up an isotope lab anew including applying for the necessary license would have taken >1 year and would have exceeded the revision time frame by far.

For the in vitro steatosis assay, we applied the “hepatic steatosis kit” from Sigma-Aldrich/Merck and visualized the lipid droplet formation in Huh-7 hepatocytes with ORO. Although this assay was also based on ORO staining of triglycerides, it provided clear evidence for a promoting effect of MIF-2 on hepatocyte lipogenesis and lipid accumulation. Of note, this effect was seen at concentrations of 4-8 nM MIF-2, i.e. the concentration range, at which also the SREBP activation effect of MIF-2 peaked.

To further explore effects of MIF-2 on lipogenesis, we opted to run the steatosis assay in the presence *versus* absence of (sub-maximum) concentrations of co-added oleic acid (OA). Again, we opted for a “cold” OA addition approach here, as our institute no longer has an isotope lab and as the regulatory hurdles for radioactive experiments are very high (see above). The observed effect of MIF-2 was similar to that of 1 mM OA, while no additive or synergistic effect between MIF-2 and OA was noted. This indicated that MIF-2 may promote both the *de novo* fatty acid synthesis step as well as fatty acid esterification to form TGs.

This notion was further confirmed by the Triglyceride-Glo™ assay, in which triglyceride levels are measured via glycerol that is released from an enzymatic reaction with a lipase: one mole of glycerol per mole of triglyceride. The assay determines both accumulated and *de novo* synthesized triglycerides. MIF-2 enhanced Huh-7 triglyceride levels in a dose-dependent manner, with a peak concentration of 2 nM.

Overall, these data add to the body of evidence suggesting that MIF-2 promotes lipogenesis in hepatocytes, in line with its effect on SREBPs, and are now shown in **new Supplementary Figure S17A-C** of the revised manuscript.

To further follow the Reviewer’s suggestion and to address the ORO sensitivity issue and comprehensively characterize any impact of MIF-2 on hepatic lipid levels in the *in vivo* mouse model, we teamed up with the lipidomics lab of Dr. Mark Haid from the core facility Metabolomics & Proteomics at the Helmholtz Center Munich. To quantify lipids in the liver samples, his team used a differential ion mobility MRM-based mass spectrometry approach (DMS-SLA¹⁰) that allows accurate quantification of up to 1500 lipid species from 20 lipid classes. The DMS-SLA assay is the successor of the Sciex Lipidizer™ with a broader spectrum of lipids including cholesterol esters (CE), ceramides (CER, HexCER, LacCER, DCER), diacylglycerols (DAG, DG), free fatty acids (FFA), sphingomyelins (SM), 9 glycerophospholipid classes, and, most importantly, triacylglycerols (TAG, TG). Comparing liver extracts from Mif-2^{-/-} Apoe^{-/-} and Apoe^{-/-} mice, the lipidomics revealed a marked and

highly significant reduction of TGs, DGs, and CEs in the *Mif-2*-deficient group, underscoring the hepatosteatotic effect of MIF-2 and delineating the lipid classes elevated by MIF-2 activity. The lipidomics data are represented in **new Figure 6** and **new Supplementary Figures S20 and S21** of the revised manuscript.

1. Another issue that was not addressed related to reductions in plasma lipids. The DKO mice have significantly lower plasma lipids. How can authors explain that the decreases in inflammatory response are not secondary to reduced atherogenic lipoproteins? How increased hepatic lipogenesis and reduced plasma lipids can be associated with reduced hepatic steatosis? If reduced plasma lipids are secondary reduced lipoprotein production, then increased hepatic lipogenesis and reduced lipoprotein production are expected to enhance hepatic steatosis.

Response: Thank you for bringing this up. There may have been a misunderstanding due to text describing the effects studied in hepatocytes *in vitro* upon stimulation with recombinant MIF-2 (“plus MIF-2”; effect of MIF-2 on hepatic lipogenesis) *versus* effects studied *in vivo* by investigating *Mif-2*-deficient mice (the DKO mice; effects of *Mif-2* deficiency; “minus MIF-2”). In fact, the Reviewer is correct in concluding that “decreases seen on vascular and systemic inflammation in the DKO mice could indeed be partly secondary to reductions in atherogenic lipoproteins, and thus be partly indirectly caused by the reduced hepatosteatosis phenotype of the liver”. This is in line with the effects of rMIF-2 on hepatocytes *in vitro* (“rMIF-2 enhances TG content and lipid accumulation in hepatocytes”) and with the *in vivo* lipidomics result showing that liver tissue of DKO mice contains considerably lower concentrations of TGs, DGs, and CEs (“*Mif-2* deficiency reduces hepatic TGs, DGs, and CEs *in vivo*”). In conjunction, we conclude from these data that: i) MIF-2 promotes lipid synthesis and accumulation in hepatocytes; ii) MIF-2 promotes hepatosteatosis; iii) MIF-2 promotes atherogenic lipoprotein production by the liver; iv) MIF-2 enhances vascular inflammation and leukocyte recruitment into the vessel wall.

We have prepared a new graphical abstract, which is now shown in the revised manuscript in **new Supplementary Figure S32**, which summarizes the pro-hepatosteatotic, pro-atherogenic, and pro-inflammatory effects of MIF-2 and highlights the inter-relationship between the liver and vascular phenotype (“feed-forward”, “auto-amplification” relationship between liver and vasculature).

Minor points:

1) Do they have information about changes in body weight? And other physiological parameters?

Response: Yes, we did record the body weights of the mice. Body weights of *Mif-2^{-/-}Apoe^{-/-}* mice were found to be significantly reduced compared to *Apoe^{-/-}* mice. These data are shown in **Figure 4I**, **Supplementary Figure S12** (formerly S9), and **Supplementary Table S2**.

In revision and to begin to address the Reviewer's question regarding "other physiological parameters", we teamed up with the laboratory of Alexander Bartelt (LMU Munich), an expert on lipid metabolism, and performed a preliminary metabolic cage/calorimetry experiment with 4x wildtype (regular C57Bl/6 background; WT) and 6x *Mif-2^{-/-}* (KO) mice. These preliminary data suggest that, at least independent of atherogenic conditions and HFD feeding, energy expenditure, food intake/consumption, and most other parameters measured in the metabolic cage set-up did not differ between KO and WT mice. Only few changes were noted. *Mif-2^{-/-}* mice appeared to consume more water and the respiratory exchange ratio (RER) was slightly lower. As these data are preliminary and so-far only based on a limited number of mice, we would prefer to not include them in the revised manuscript, but provide them as **Reviewer-only Figure R1** (as well as **Reviewer-only Figure R6** in response to Reviewer #3) in this point-by-point-response letter.

Reviewer-only Figure R1: Indirect calorimetry/metabolic cage experiment comparing wildtype versus *Mif-2^{-/-}* mice. Four wildtype (WT) and six *Mif-2^{-/-}* (16 weeks old) were studied in a Promethion metabolic cage device (Sable Systems Europe GmbH, Berlin, Germany) over a period of 5 days (120 h). **(A)** energy expenditure (Kcal/h); **(B)** total food consumption (Kcal). White and grey shades indicate diurnal periods. Data points in curves are means \pm SD.

2) Fig 4B: DKO mice aortic arch shows more ORO staining than *Apoe* KO mice. Please check whether pictures have been misplaced?

Response: Thank you for the hint. We have double-checked on those images in both Figure 4B (aortic root) and Figure 4D (aortic arch) and can confirm that the images as shown are correct. In line with the reduction depicted in the quantified bar graph of Figure 4B, the ORO-positive area in 4B left, while seemingly a bit more reddish, is considerably smaller in size. Similarly, in agreement with the reduction depicted in the quantified bar graph of Figure 4D, the HE-positive area in 4D left is considerably smaller in size (see dotted/circled areas indicating the actual lesions).

3) Did authors measure cholesterol and triglyceride in FPLC fractions? Such as analysis will provide better understanding about the changes in plasma lipids and lipoproteins?

Response: Thank you for this question. Indeed, the different subclasses of lipoproteins featuring different cholesterol and TG contents were analyzed by an FPLC methodology. The confusion arose due to **typographical errors in the results section**, specifically **on page 13, line 325**, and **page 79, line 2021**, where "HPLC" was mistakenly written instead of "FPLC". We apologize for this oversight. The FPLC method is described in the Materials & Methods section of the manuscript (**page 43**) and reads: "*Plasma very low-density lipoproteins (VLDL), low-density lipoproteins (LDL), and high-density lipoproteins (HDL) were separated by fast performance liquid chromatography (FPLC)-size exclusion chromatography analysis (gel filtration on a Superose 6 column (GE Healthcare) at a flow rate of 0.5 mL/min). The different lipoprotein fractions were collected according to their retention times as follows: VLDL between 40 and 50 min, LDL between 50 and 70 min, and HDL between 70 and 90 min. Results are presented as optical density (OD) profiles at 492 nm of the isolated lipoproteins*". A further sub-analysis of cholesterol and TGs from the lipoprotein fractions was not performed, but our method used is well established^{11, 12} and accurately delineates the VLDL, LDL, and HDL fractions, which in turn are known to contain the following lipid class contents: VLDL (55% TGs, 24% CEs, 20% phospholipids, 1% FFA); LDL (12% TGs, 59% CEs, 28% phospholipids, 1% FFA); HDL (12% TGs, 40% CEs, 47% phospholipids, 1% FFA). The results are shown in Figure 4L and Supplementary Figure S13.

4) Authors agreed that MIF-2 plays a role in lipogenesis. They highlighted that RNA-Seq analysis identified down regulation of apoA4, PCSK9 and Ces2. It is unclear how these proteins play a role in lipogenesis. It is also not known how purine metabolites play a role in lipogenesis.

Response: The Reviewer raises interesting questions. Indeed, some of the genes identified in the RNAseq analysis, including *ApoA4*, *Pcsk9*, *Ces2*, and purine metabolites are not directly involved in the mere lipogenesis process per se but rather in lipid metabolism in broader terms. In fact, *ApoA4*, *PCSK9*, and *Ces2* are involved in lipid metabolism, transport, and turnover, processes closely linked to lipogenesis. Purine metabolites play a critical role in supplying the energy and the cofactors required for these metabolic activities.

To further perform the newly identified role of MIF-2 in lipid metabolism, particularly in lipogenesis, we have meanwhile **performed a second – more comprehensive – bulk RNAseq analysis from whole liver lysates** of *Mif-2^{-/-}ApoE^{-/-}* versus *ApoE^{-/-}* mice, also sampling a larger mouse number per group (**new Supplementary Figure S18**). Overall, this second RNAseq confirmed the data obtained in the first one (previous Figure 5K-M), with overwhelming **evidence for an enrichment of DEGs and GO terms related to "fatty acid metabolic process" and "regulation of lipid metabolic process"**. While confirming many of the previously identified individual genes (e.g. *Ces2*, *Ces1*, or *Hsd17b*; see our further response below), the new RNAseq analysis also highlighted some additional DEGs such as several cytochrome P450 genes that were enriched in the *ApoE^{-/-}* group.

We also **performed RT-qPCR from liver tissue** of *ApoE^{-/-}* and *Mif-2^{-/-}ApoE^{-/-}* mice to confirm the expression of the genes identified in the RNAseq analyses, including *Pcsk9*, *ApoA4*, and the isoforms of *Ces2*, *Ces2a*, *Ces2c* and *Ces2e* (latter not shown) (**new Supplementary Figure S19**). The results showed a downregulation of *ApoA4*, *Ces2c*, and *Ces2e* mRNA levels in the livers of *Mif-2^{-/-}ApoE^{-/-}* mice, although the reduction did not reach significance. Interestingly, we observed an unexpected increase in the expression of *Pcsk9* in the livers of *Mif-2^{-/-}ApoE^{-/-}* mice. This upregulation was in line with the elevated transcript levels of *Pcsk9* seen in the *Mif-2^{-/-}ApoE^{-/-}* mice through RNAseq data. *PCSK9* primarily regulates cholesterol uptake and metabolism in the liver by modulating the LDL receptor, which leads to increased circulating cholesterol levels. However, in this study, we found that *Mif-2^{-/-}ApoE^{-/-}* mice had lower plasma cholesterol levels. Moreover, the RNAseq and RT-qPCR results indicated a significant reduction in the expression of key enzymes and factors involved in cholesterol biosynthesis and lipid uptake in the liver, including peroxisome *peroxisome proliferator-activated receptor gamma (Ppar-γ, Pparg)*, the scavenger receptor *Cluster of differentiation 36 (Cd36)*, *scavenger receptor class B type 1 (Sr-b1)*, *3-hydroxymethylglutaryl-coenzyme A*

synthase-1 (Hmgcs1), 3-hydroxy-3-methyl-glutaryl-coenzyme A reductase (Hmgcr), acetyl-CoA acetyltransferase 1 (Acat-1), when compared to control mice (Figure 5K-M and **new Supplementary Figure S19**). To this end, it is also important to note that *Srebp-2* was also among the significantly down-regulated genes in the livers of *Mif-2^{-/-}Apoe^{-/-}* mice, confirming the RNAseq data and highlighting the potential role of MIF-2 in cholesterol metabolism. Thus, beyond the regulatory effect of PCSK9 on circulating cholesterol, the downregulation of these crucial genes in the liver likely contributes to the observed decrease in circulating cholesterol levels. The increased expression of PCSK9 may be explained by a compensatory mechanism. We have added some **additional discussion** regarding these genes (and others) identified in both RNAseq analyses in the Discussion section of the revised manuscript (**page 27-29**).

5) *Authors have not discussed the regulation of ACC and FASN in the DKO livers; two critical enzymes in lipogenesis.*

Response: Thank you for alerting us to this. We have added discussion on both the ACC and FASN genes and ACC and FAS enzymes and their stimulation by MIF-2 in Huh-7 hepatocytes in the Discussion section of the revised manuscript (see **page 27**).

6) *Maybe they should de-emphasize lipid metabolism and mainly focus on chemokine studies.*

Response: Thank you for this suggestion. In revision, we have added substantially more discussion on the lipid phenotype as requested. This, we feel, also has led to a more balanced emphasis on the lipid *versus* chemokine parts of the MIF-2 activity spectrum. Moreover, the new graphical abstract, now provided as **new Supplementary Figure S32**, better links the two phenotypes and at the same time emphasizes that and how “the inflammatory/chemokine part of the coin” drives the liver/lipid phenotype and *vice versa*. The added discussion on “therapeutic concepts in atherosclerosis and vascular inflammation”, now included in the revised manuscript on **page 31-32** in response to a point raised by Reviewer #1, also lays additional emphasis on the chemokine focus of our study. Together, we think, the discussion now is more balanced and adequately reflects the focus on the chemokine aspect of the study. We hope that the Reviewer agrees.

Reviewer #3:

General comments:

This article presents new findings on how MIF2, d-dopachrome tautomerase (DDT) regulates atherosclerosis and hepatic lipogenesis. Chemokines affecting both vascular and metabolic pathologies are rare, but DDT/MIF2 promotes atherosclerotic plaque formation and hepatic lipid accumulation. MIF2 deficiency or blockade of MIF2 by the pharmacological inhibitor, 4-CPPC successfully protected lesion formation and vascular inflammation in early and advanced atherogenesis. In vitro experiments demonstrate that MIF2 promotes monocyte migration and arrest, and B cell recruitment, and foam cell formation which are important steps in the development of atherogenesis. MIF2 regulates these immune functions by binding to the CXCR4 receptor. In the liver, MIF2 also bind to CXCR4 and CD74 to regulate the Akt/SREBP signaling pathway. Finally, MIF2 staining was enhanced in unstable human carotid plaques from atherosclerotic patients undergoing CEA. Plasma MIF2 levels are high in CAD patients and correlated with disease severity. All these data suggest an important role for MIF2 in mediating atherosclerosis and affecting hepatic lipogenesis. Overall, the current study is very well organized and presents for the first-time novel findings regarding the role of DDT in modulating atherosclerosis. The reviewer has some thoughts and questions about this manuscript as follows.

Response: We very much appreciate this feedback and the excellent summary of our study.

Specific points:

1) *This study only compares Apoe^{-/-} and Mif2^{-/-}Apoe^{-/-} mice with HFD. It is unclear whether*

HFD upregulates plasma MIF2 compared to normal chow. If so, will MIF2 induce hepatic lipogenesis in 12-week HFD fed mice or this only happens in *Apoe*^{-/-} mice with HFD?

Response: Following the Reviewer's recommendation, we have determined plasma Mif-2 levels in *Apoe*^{-/-} mice fed an HFD or chow diet. As shown in **Reviewer-only-Figure R2**, Mif-2 levels did not differ between both feeding regimes. We did not compare the influence of HFD versus chow in non-*Apoe*^{-/-} wildtype mice, as our animal license currently does not cover HFD feeding to normal (i.e. non-*Apoe*^{-/-}) BL/6 wildtype mice; a new application and approval process plus the actual 12-week HFD feeding procedure in a new experiment would take 9-12 months, thus substantially extending our granted revision timeline. We hope, the Reviewer can agree and we think that the comparison of Mif-2 levels in HFD versus chow diet-fed *Apoe*^{-/-} mice overall addresses this important question. We conclude that HFD *per se* does not upregulate Mif-2.

Reviewer-only Figure R2: Plasma MIF-2 levels do not differ between high-fat diet (HFD) and chow feeding. Mif-2 levels were determined in plasma samples from *Apoe*^{-/-} mice fed an HFD or chow diet (6 mice per group) for 12 weeks by Mif-2-specific ELISA (see Materials & Methods part of the main manuscript).

2) Where is MIF2 derived from? If it is majorly from immune cells, whether transplantation of *Mif2*^{-/-} bone marrow cells into *Apoe*^{-/-} could reverse the atherosclerotic phenotypes?

Response: The Reviewer raises an important point. We suspect that immune cells are indeed a major source for Mif-2, but please see also our above discussion, answer, and revision experiments in response to a question from Reviewer #1, which together also indicate that VSMCs could be an additional contributing source of Mif-2 in the plaque microenvironment. Moreover, interestingly data bank entries (e.g. the Human Protein Atlas (HPA); <https://www.proteinatlas.org/ENSG0000099977-DDT/single+cell+type>; see **Reviewer-only Figure R3**) suggest a high abundance of MIF-2 expression in hepatocytes, a notion also supported by tissue Western blot experiments in earlier work on MIF-2¹³.

Reviewer-only Figure R3: RNA expression of D-dopa-chrome tautomerase (D-DT, aka MIF-2) at single cell level. Interrogation of the Human Protein Atlas (HPA) for D-DT and focus on RNA single cell query.

We agree with the Reviewer that – eventually – the causality of such potential contributions must be tested in cell-/tissue-specific knockout animal models.

Following up on the Reviewer’s suggestion to test for the contribution of myeloid/hematopoietic cells, we have **performed a bone marrow transplantation (BMT) experiment**, in which *Mif-2*-deficient vs wildtype (WT) donor BM was transplanted into atherogenic recipient mice, followed by a 12-week HFD period. For this experiment, we teamed up with the laboratory of Christian Schulz (formerly LMU Munich; now University of Heidelberg/Mannheim), which runs an established non-irradiation-based BMT mouse model that is based on BM-cell depletion by conditional gene deletion to ablate *c-Myb*¹⁴. *Mx1*^{Cre}:*c-Myb*^{fl/fl}:*CD45*^{2/2} mice were crossed with *Ldlr*^{-/-} mice to create *Mx1*^{Cre}:*c-Myb*^{fl/fl}:*CD45*^{2/2} *x Ldlr*^{-/-} atherogenic recipient mice. BM depletion is achieved by repetitive injections of poly(I:C). For BMT, bone marrow from *Mif-2*^{-/-} versus WT (*CD45*^{1/1}) was transplanted, and mice subjected to cholesterol-rich HFD. Due to breeding limitations, we have so far only been able to study 4 mice per group in this experiment and are providing these preliminary data as **Reviewer-only Figure R4**.

Reviewer-only Figure R4: Bone marrow transplantation experiment with *Mif-2*-deficient versus wildtype bone marrow to study the contribution of hematopoietic *Mif-2* to atherosclerotic plaque formation. *Mx1*^{Cre}:*c-Myb*^{fl/fl}:*CD45*^{2/2} *x Ldlr*^{-/-} atherogenic recipient mice were repetitively injected with poly(I:C) to achieve BM depletion (for detailed methods see¹⁴). Seven days later, BM was harvested from *Mif-2*^{-/-} versus WT (*CD45*^{1/1}) donor mice, and *Mx1*^{Cre}:*c-Myb*^{fl/fl}:*CD45*^{2/2} *x Ldlr*^{-/-} recipient mice received 1 × 10⁷ BM cells *via* tail vein injection. After a recovery period of 6 d, animals were fed a cholesterol-rich HFD for 12 weeks. **(A)** Representative oil red O (ORO)-stained aortic root sections from all 4 mice per group showing atherosclerotic plaque formation. Upper panel: BM from wildtype (WT) donor mice; lower panel: BM from *Mif-2*^{-/-} donor mice. **(B, C)** Quantification of lesion areas according to **(A)**; **(B)** mean of ORO-positive lesion areas; values shown are means ± SD and were analyzed by two-

tailed Student's t-test; ns, not significant. (C) Lesion areas plotted against distance from apex. WT, wildtype; KO, *Mif-2*^{-/-}; #1-4, individual mice from each group. Scale bar: 200 μm.

Although the data are very preliminary due to the low mouse numbers in both groups and the observed high variability of lesion areas (from sub-threshold to several hundred x10³ μm), we would carefully conclude that the determined lesion areas appear to be lower, when BM from *Mif-2*^{-/-} mice was transplanted compared to BM from WT mice, indicating a **potential role for hematopoietic Mif-2 in driving atherosclerotic lesion formation**.

Due to the moving process of Dr. Schulz' lab from Munich to Heidelberg/Mannheim, an expanded colony of the *Mx1*^{Cre}:*c-Myb*^{fl/fl}:*CD45*^{2/2} x *Ldlr*^{-/-} recipient mice is currently not available and will realistically not be available with the sufficient mouse numbers needed for a full-scale study until beginning/spring 2025. We thus hope that the Reviewer and Editor consent that we only provide these preliminary data as a Reviewer-only Figure in this response letter.

To nevertheless further address the Reviewer's point, we have **added a discussion and limitation statement in the revised manuscript**, submitting and hypothesizing that the main contributing source of MIF-2 (cell type and/or tissue) is currently unclear and needs to be elucidated in future studies, and that both myeloid/hematopoietic and hepatic MIF-2 may contribute to vascular inflammation/atherosclerotic plaque formation and hepatosteatosis" (see **Discussion section of revised manuscript, page 31-32**).

3) Was 4-CPPC also injected into *Mif2*^{-/-}*Apoe*^{-/-} as shown in Figure 11? But this experiment was not mentioned in the article?

Response: Thank you for bringing this up. *Mif-2*^{-/-}*Apoe*^{-/-} was an inadvertent copy/paste error. In Figure 11, only *Apoe*^{-/-} mice were used. We are providing a **revised Figure 11**.

4) Why do *Mif2*^{-/-} and MIF2 inhibition show reductions in different cytokine profiles? Does this suggest that MIF2 inhibitors may not be able to fully block the effects of MIF2?

Response: Sorry, we have probably not sufficiently elaborated on this point. Actually, while some differences can be noted between the groups, the overall cytokine profiles are quite similar, when comparing *Mif-2*-deficient *Apoe*^{-/-} mice at 4.5-week HFD with *Apoe*^{-/-} mice treated with 4-CPPC alongside the 4.5-week HFD. The following cytokines and chemokines were found to be reduced in a comparable manner in female or male *Mif-2*^{-/-}*Apoe*^{-/-} mice and male *Apoe*^{-/-} mice treated with 4-CPPC (comparison between three groups): IL-1α, IFN-γ, TNF-α, IL-23, CXCL11, and CCL2. Moreover, the reduction of numerous cytokines and chemokines is similar, when comparing male *Mif-2*^{-/-}*Apoe*^{-/-} mice and male *Apoe*^{-/-} treated with 4-CPPC (comparison between two groups). These are: IL-1α, IFN-γ, IL-5, CXCL5, IL-15, TNF-α, IL-4, IL-23, IL-1β, CCL5, CCL2, and IL-27. Few similarities were also observed, when comparing the profiles between female *Mif-2*^{-/-}*Apoe*^{-/-} mice at 4.5-week *versus* 12-week HFD (e.g. G-CSF, IL-27, TNF-α, IL-1α, IL-23). At the same, the Reviewer is right that the profiles are not identical. Differences observed between the profile in *Mif-2*-deficient *Apoe*^{-/-} mice and *Apoe*^{-/-} mice treated with 4-CPPC could be due to compensatory effects in global genetically *Mif-2*-deficient mice. Perhaps more relevant, 4-CPPC only targets extracellular MIF-2, whereas the genetic *Mif-2* knockout affects both the intra- and extracellular compartment of MIF-2. Also, the pharmacological MIF-2 blockade likely is less efficient than the genetic depletion due to tissue distribution, pharmacokinetic, and IC₅₀ considerations. We have **incorporated text in the Results and Discussion section of the revised manuscript** to address this point (**pages 8 and 26**).

5) 4.5 week HFD significantly regulates the formation of atherosclerotic plaques whether this short-term HFD feeding also affects liver lipogenesis or alteration of liver lipogenesis only occurs following 12 week HFD feeding?

Response: This is an interesting question. Unfortunately, unlike for the 12-week HFD (more chronic) model, in which effects on the liver were to be expected, we did not initially plan to study the liver in this model of early atherogenesis; so only a couple of livers were prepared and stored, making a reliable analysis of lipid levels difficult. We would assume that *Mif-2*-deficiency may already slightly affect liver lipogenesis after only a few weeks of HFD feeding, but this is speculative and an effect is probably going to be rather small, if present at all.

6) Does *Mif2*^{-/-} affect food intake and is accompanied by weight loss as shown in Figure S9D?

Response: Thank you. As similar question was also asked by Reviewer #2. Please also see our response there. To address this Reviewer's (as well as Reviewer #2's) question, we teamed up with the laboratory of Alexander Bartelt (LMU Munich), an expert on lipid metabolism, and performed a preliminary metabolic cage/calorimetry experiment with 4x wildtype (regular C57Bl/6 background; *WT*) and 6x *Mif-2*^{-/-} (KO) mice.

These preliminary data indicate that MIF-2 does not affect energy expenditure and food intake/consumption. Only very few changes were noted in the metabolic cage experiment. *Mif-2*^{-/-} mice appeared to consume more water and the respiratory exchange ratio (RER) was slightly lower. As these data are preliminary and so-far only based on a limited number of mice, we would prefer to not include them in the revised manuscript, but provide them as **Reviewer-only Figure R5** (as well as **Reviewer-only Figure R1** in response to Reviewer #2) in this point-by-point-response letter.

Reviewer-only Figure R5: Indirect calorimetry/metabolic cage experiment comparing wildtype versus *Mif-2*^{-/-} mice. Four wildtype (WT) and six *Mif-2*^{-/-} (16 weeks old) were studied in a Promethion metabolic cage device (Sable Systems Europe GmbH, Berlin, Germany) over a period of 5 days (120 h). (A) energy expenditure (Kcal/h); (B) total food consumption (Kcal). White and grey shades indicate diurnal periods. Data points in curves are means ± SD.

7) In Figure 5D, F and N and Supplemental Figure S14, MIF2 treatment inhibits AMPK but activates Akt. Akt may be a key upstream regulator of SREBP in hepatocytes. However, in fact, high fat diet may induce insulin resistance, resulting in reduced Akt in the liver. Just curious if MIF2 improves insulin sensitivity in the liver in vivo?

Response: Thank you for this interesting thought. Please note that due to the various revision experiments and modifications of the figures, the AMPK and AKT data were jointly moved to new **Supplementary Figure S22**, which now also contains further validation of the effect of MIF-2 on AKT by showing a Western blot comparison of pAKT/AKT levels in *Mif-2*^{-/-}*ApoE*^{-/-} versus *ApoE*^{-/-} liver tissue. We also examined the expression levels of hepatic AKT2, a key isoform of the protein kinase AKT, which activates SREBP-1 to promote fatty acid production

and triglyceride accumulation, contributing to hepatic steatosis¹⁵. Interestingly, although *Akt2* mRNA levels were downregulated in the livers of *Mif-2^{-/-}Apoe^{-/-}* mice, the decrease was not statistically significant (**Reviewer-only Figure R6**). Whether MIF-2 affects AKT2 activity in the liver requires further investigation.

Reviewer-only Figure R6: mRNA expression of *Akt2* in liver tissue of *Apoe^{-/-}* and *Mif-2^{-/-}Apoe^{-/-}* mice as determined by RT-qPCR. Total RNA was extracted from the livers of *Apoe^{-/-}* and *Mif-2^{-/-}Apoe^{-/-}* mice (n = 5 per group) after 12 weeks of HFD. The mRNA levels of *Akt2* were quantified by RT-qPCR and are expressed as relative mRNA expression levels. Data are presented as means ± SD and were analyzed by two-tailed Student's t-test; ns, not significant.

Overall, and based on these findings, we share the Reviewer's view that AKT may be an important MIF-2-driven upstream regulator of SREBP in hepatocytes. We also agree that HFD may induce insulin resistance (IR) – although IR was so far not shown to be clearly associated with Western diet feeding in an *Apoe^{-/-}* background in BL6 mice^{16, 17} - and that this may lead to attenuated AKT activity, which in turn would reduce SREBP activity and lipid synthesis pathways. Unfortunately, our animal experiment was not set to test/monitor IR.

We did perform an *in vitro* experiment, in which we incubated Huh-7 hepatocytes with different concentrations of insulin. Interestingly, insulin exhibited a dose-dependent stimulatory effect on the gene expression of *MIF-2*, but also of *MIF*, and the two MIF family receptors *CD74* and *CXCR4*. While very preliminary at this point, this *in vitro* experiment may indicate a complex impact of insulin (and potentially IR) on the ligands and receptors of the entire MIF family, making it difficult to predict what the net effect of MIF-2 on insulin resistance *versus* sensitivity may be.

Reviewer-only Figure R7: Preliminary data suggesting that insulin may induce MIF-2, MIF, CD74, and CXCR4 in Huh-7 hepatocytes. Huh-7 hepatocytes were stimulated with different concentrations of insulin as indicated for 16 h and MIF-2, MIF, CD74, and CXCR4 mRNA quantified by qPCR. Both MIF isoforms and both receptors are upregulated under these conditions. N = 3 independent experiments. Values shown are means ± SD; *, $P < 0.05$; **, $P < 0.01$; ****, $P < 0.0001$.

The early study by Iwata and colleagues¹⁸, who administered recombinant Mif-2 in *db/db* mice, a model to study the phases 1-3 of type 2 diabetes and obesity, observed improved glucose intolerance and decreased serum FFA levels. At first sight, these data, in terms of the overall effect on metabolic direction and impact on lipids, do not seem to be in line with our findings. However, the models are hugely different. Our Western diet/*ApoE*^{-/-} based atherosclerosis model combines vascular inflammation with hepatosteatosis, but lacks strong features of obesity; in contrast, the *db/db* mouse model is a classical diabetes/obesity model lacking major features on inflammation, in particular vascular inflammation, while exhibiting massive chronic hyperglycemia, pancreatic beta cell atrophy and propensity to be hypoinsulinemic.

Together, we thus cannot predict at this point, whether *Mif-2* deficiency may affect IR. To test this in an *in vivo* setting would be valuable and should be performed in the future.

We have added a **limitations and discussion statement** in the Discussion section to address the Reviewer's point and to discuss the mechanistic complexity for the reader (see **page 18 in the Discussion section** of the revised manuscript).

8) In Figure 5N, reduction of AMPK may attenuate fatty acid oxidation, further exacerbating lipid storage. Fatty acid oxidation related genes may need to be evaluated, such as *CPT1* which is also a downstream target of AMPK?

Response: Thank you for this excellent suggestion. In this context, we would like to highlight that we have performed a second independent bulk RNAseq analysis from liver lysates. This experiment on one hand served to validate the first RNAseq analysis, but also was done on whole liver lysates (rather than from liver sections) and encompassed a larger sample of mice in each group. This new RNAseq analysis is now shown as **new Supplemental Figure S18** in the revised manuscript. Of note, among the transcripts found to be highly upregulated in *ApoE*^{-/-} compared to *Mif-2*^{-/-}*ApoE*^{-/-} liver tissue were numerous cytochrome P450 type hydroxylases. We also followed the Reviewer's suggestion to specifically check for the expression of fatty acid oxidation genes including *Cpt1*, *Cpt2*, and *Acox-1* (*acyl-CoA oxidase 1*). RT-qPCR experiments performed in the course of the revision and now shown in **new Supplementary Figure S19** demonstrate a significant reduction of both *Cpt1* and *Acox-*

1 expression in *Mif-2^{-/-}ApoE^{-/-}* liver tissue (**Figure S19 panel O and P**). *Cpt2* levels were also reduced, although this decrease was not statistically significant (**Figure S19 panel Q**). Together, these analyses indicate that Mif-2 promotes the expression of hepatic *Cpt1*, *Acox-1* and several *Cyp* monooxygenases involved in lipid hydroxylation/oxidation. On the other hand, MIF-2 was found to attenuate AMPK levels in Huh-7 hepatocytes. Hepatic AMPK phosphorylates ACC, leading to inhibition of its activity, which in turn downregulates fatty acid biosynthesis and upregulates CPT1 to enhance β -oxidation¹⁹. In contrast, our RT-qPCR data reveal a significant downregulation of hepatic *Cpt1* and *Acox-1* expression, suggesting a potential decrease in β -oxidation. Moreover, lipidomics analysis shows a marked reduction in fatty acid concentrations in the livers of *Mif-2^{-/-}ApoE^{-/-}* mice. Therefore, we would conclude that the effects of MIF-2 on AMPK, CPT1, and the CYP enzymes are independent pathways. In turn, the indirect attenuating effect of MIF-2 on AMPK downstream enzymes such as CPT1 is thus probably overridden by stimulatory effects of MIF-2 on other pathways that may drive CPT1 and CYP enzymes. One such potential effect might be an indirect effect via CD36 and or SR-B1. *Mif-2^{-/-}ApoE^{-/-}* liver tissue also featured significantly reduced *Cd36* and *Sr-b1* levels, key scavenger receptors responsible for the uptake of FFAs and lipids. This may lead to a secondary down-regulatory effect on downstream enzymes such as CPT1²⁰.

9) Additionally, many experimental details are missing from the manuscript. For example, what is the duration of MIF2 treatment in Figure 2A to C?

Response: Thank you for this hint. In some cases, due to word count limitations of the figure legends, not all experimental details were in the figure legend; however, they were then meant to be present in the respective methods chapter. For example, the duration of MIF-2 treatment in Figure 2A was described in the Methods paragraph “flow adhesion assay”, but was not added again in the legend of Figure 2 for space reasons. Similar for Figure 2B and C. We have added the missing treatment time information to the Figure legend of 2A-C now in the revised version. Regarding other potentially missing experimental details, we have reviewed all methods paragraphs and figure legends again, and have added information in several places (see revised figure legends).

References cited in the point-by-point response

1. Misra, A. *et al.* Integrin beta3 regulates clonality and fate of smooth muscle-derived atherosclerotic plaque cells. *Nat Commun* **9**, 2073 (2018).
2. Wang, Y. *et al.* Smooth Muscle Cells Contribute the Majority of Foam Cells in ApoE (Apolipoprotein E)-Deficient Mouse Atherosclerosis. *Arterioscler Thromb Vasc Biol* **39**, 876-887 (2019).
3. Wirka, R.C. *et al.* Atheroprotective roles of smooth muscle cell phenotypic modulation and the TCF21 disease gene as revealed by single-cell analysis. *Nat Med* **25**, 1280-1289 (2019).
4. Pan, H. *et al.* Single-Cell Genomics Reveals a Novel Cell State During Smooth Muscle Cell Phenotypic Switching and Potential Therapeutic Targets for Atherosclerosis in Mouse and Human. *Circulation* **142**, 2060-2075 (2020).
5. Pekayvaz, K. *et al.* Mural cell-derived chemokines provide a protective niche to safeguard vascular macrophages and limit chronic inflammation. *Immunity* **56**, 2325-2341 e2315 (2023).
6. Li, Y. *et al.* AMPK phosphorylates and inhibits SREBP activity to attenuate hepatic steatosis and atherosclerosis in diet-induced insulin-resistant mice. *Cell Metab* **13**, 376-388 (2011).
7. Tang, J.J. *et al.* Inhibition of SREBP by a small molecule, betulin, improves hyperlipidemia and insulin resistance and reduces atherosclerotic plaques. *Cell Metab* **13**, 44-56 (2011).
8. Heinrichs, D. *et al.* Protective role of macrophage migration inhibitory factor in nonalcoholic steatohepatitis. *FASEB J* **28**, 5136-5147 (2014).
9. Miller, E.J. *et al.* Macrophage migration inhibitory factor stimulates AMP-activated protein kinase in the ischaemic heart. *Nature* **451**, 578-582 (2008).
10. Su, B. *et al.* A DMS Shotgun Lipidomics Workflow Application to Facilitate High-Throughput, Comprehensive Lipidomics. *J Am Soc Mass Spectrom* **32**, 2655-2663 (2021).

11. Doring, Y. *et al.* Identification of a non-canonical chemokine-receptor pathway suppressing regulatory T cells to drive atherosclerosis. *Nat Cardiovasc Res* **3**, 221-242 (2024).
12. Lehti, M. *et al.* High-density lipoprotein maintains skeletal muscle function by modulating cellular respiration in mice. *Circulation* **128**, 2364-2371 (2013).
13. Merk, M. *et al.* The D-dopachrome tautomerase (DDT) gene product is a cytokine and functional homolog of macrophage migration inhibitory factor (MIF). *Proc Natl Acad Sci U S A* **108**, E577-585 (2011).
14. Stremmel, C. *et al.* Inducible disruption of the c-myc gene allows allogeneic bone marrow transplantation without irradiation. *J Immunol Methods* **457**, 66-72 (2018).
15. Leavens, K.F., Easton, R.M., Shulman, G.I., Previs, S.F. & Birnbaum, M.J. Akt2 is required for hepatic lipid accumulation in models of insulin resistance. *Cell Metab* **10**, 405-418 (2009).
16. Li, J. *et al.* Hyperglycemia in apolipoprotein E-deficient mouse strains with different atherosclerosis susceptibility. *Cardiovasc Diabetol* **10**, 117 (2011).
17. Park, K. *et al.* Exogenous Insulin Infusion Can Decrease Atherosclerosis in Diabetic Rodents by Improving Lipids, Inflammation, and Endothelial Function. *Arterioscler Thromb Vasc Biol* **38**, 92-101 (2018).
18. Iwata, T. *et al.* The action of D-dopachrome tautomerase as an adipokine in adipocyte lipid metabolism. *PLoS One* **7**, e33402 (2012).
19. Srivastava, R.A. *et al.* AMP-activated protein kinase: an emerging drug target to regulate imbalances in lipid and carbohydrate metabolism to treat cardio-metabolic diseases. *J Lipid Res* **53**, 2490-2514 (2012).
20. Matsushita, Y., Nakagawa, H. & Koike, K. Lipid Metabolism in Oncology: Why It Matters, How to Research, and How to Treat. *Cancers (Basel)* **13** (2021).